# Ocean carbon and nitrogen isotopes in CSIRO Mk3L-COAL version 1.0: A tool for palaeoceanographic research

Pearse J Buchanan[1,2,3], Richard J Matear[2,3], Zanna Chase[1], Steven J Phipps[1], and Nathan L Bindoff[1,2,3,4]

[1]Institute for Marine and Antarctic Studies, University of Tasmania, Hobart, Tasmania, Australia.
[2]CSIRO Oceans and Atmosphere, CSIRO Marine Laboratories, G.P.O Box 1538, Hobart, Tasmania, Australia.
[3]ARC Centre of Excellence in Climate System Science, Hobart, Tasmania, Australia
[4]Antarctic Climate and Ecosystems Cooperative Research Centre, Hobart, Tasmania, Australia.

**Correspondence:** Pearse James Buchanan (pearse.buchanan@liverpool.ac.uk). Now at the Department of Earth, Ocean and Ecological Sciences, University of Liverpool, Liverpool, L69 3GP, United Kingdom.

**Abstract.** The isotopes of carbon ($\delta^{13}$C) and nitrogen ($\delta^{15}$N) are commonly used proxies for understanding the ocean. When used in tandem, they provide powerful insight into physical and biogeochemical processes. Here, we detail the implementation of $\delta^{13}$C and $\delta^{15}$N in the ocean component of an Earth system model. We evaluate our simulated $\delta^{13}$C and $\delta^{15}$N against contemporary measurements, place the model's performance alongside other isotope-enabled models, and document the response
of $\delta^{13}$C and $\delta^{15}$N to changes in ecosystem functioning. The model combines the Commonwealth Scientific and Industrial Research Organisation Mark 3L (CSIRO Mk3L) climate system model with the Carbon of the Ocean, Atmosphere and Land (COAL) biogeochemical model. The oceanic component of CSIRO Mk3L-COAL has a resolution of $1.6°$ latitude $\times$ $2.8°$ longitude and resolves multi-millennial timescales, running at a rate of $\sim$400 years per day. We show that this coarse resolution, computationally efficient model adequately reproduces water column and coretop $\delta^{13}$C and $\delta^{15}$N measurements, making it
a useful tool for palaeoceanographic research. Changes to ecosystem function involve varying phytoplankton stoichiometry, varying $CaCO_3$ production based on calcite saturation state, and varying $N_2$ fixation via iron limitation. We find that large changes in $CaCO_3$ production have little effect on $\delta^{13}$C and $\delta^{15}$N, while changes in $N_2$ fixation and phytoplankton stoichiometry have substantial and complex effects. Interpretations of palaeoceanographic records are therefore open to multiple lines of interpretation where multiple processes imprint on the isotopic signature, such as in the tropics where denitrification, $N_2$
fixation and nutrient utilisation influence $\delta^{15}$N. Hence, there is significant scope for isotope-enabled models to provide more robust interpretations of the proxy records.

## 1 Introduction

Elements that are involved in reactions of interest, such as exchanges of carbon and nutrients, experience isotopic fractionation. Typically, the heavier isotope will be enriched in the reactant during kinetic fractionation, in more oxidised compounds during equilibrium fractionation, and in the denser form during phase state fractionation (i.e. evaporation). Because fractionation
against one isotope relative to the other is minuscule, the isotopic content of a sample is conventionally expressed as a $\delta$ value ($\delta^h E$), where the ratio of the heavy to light element in solution ($^h$E:$^l$E) is compared to a standard ratio ($^h$E$_{std}$:$^l$E$_{std}$) in units

of per mille (‰).

$$\delta^h E = \left( \frac{{}^h E : {}^l E}{{}^h E_{std} : {}^l E_{std}} - 1 \right) \cdot 1000 \tag{1}$$

The strength of fractionation against the heavier isotope during a given reaction, $\epsilon$, is also expressed in per mille notation. Fractionation with an $\epsilon$ equal to 10 ‰, for example, will involve 990 units of ${}^h E$ for every 1000 units of ${}^l E$ at a hypothetical standard ratio (${}^h E_{std} : {}^l E_{std}$) of 1:1. At more realistic standard ratios $<<<$ 1:1, say 0.0112372:1 for a $\delta^{13}$C value of 0 ‰, a fractionation at 10 ‰ would involve $\sim 0.0111123 \left( 0.010 \cdot \frac{0.0112372}{1.0112372} \right)$ units of ${}^{13}$C per unit of ${}^{12}$C. Slightly greater preference of one isotope over another in this case involves a preference for the lighter carbon isotope (${}^{12}$C) over the heavier (${}^{13}$C), which enriches the remaining dissolved inorganic carbon (DIC) in ${}^{13}$C and depletes the product. Certain isotopic preferences, or strengths of fractionation, therefore allow certain reactions to be detected in the environment.

The measurement of the stable isotopes of carbon ($\delta^{13}$C) and nitrogen ($\delta^{15}$N) have been fundamental for understanding these important elements cycle within the ocean (e.g. Schmittner and Somes, 2016; Menviel et al., 2017; Rafter et al., 2017; Muglia et al., 2018). We will now briefly introduce each isotope in turn.

The distribution of $\delta^{13}$C is dependent on air-sea gas exchange, ocean circulation and organic matter cycling. These contributions make the $\delta^{13}$C signature difficult to interpret, and several modelling studies have attempted to elucidate their roles (Tagliabue and Bopp, 2008; Schmittner et al., 2013). These studies have shown that preferential uptake of ${}^{12}$C over ${}^{13}$C by biology in surface waters enforces strong horizontal and vertical gradients in $\delta^{13}$C of DIC ($\delta^{13}$C$_{DIC}$), greatly enriching surface waters, particularly in subtropical gyres where vertical exchange with deeper waters is restricted (Tagliabue and Bopp, 2008; Schmittner et al., 2013). Meanwhile, air-sea gas exchange and carbon speciation control the $\delta^{13}$C$_{DIC}$ reservoir over longer timescales (Schmittner et al., 2013). Because air-sea and speciation fractionation are temperature-dependent, such that cooler conditions tend to elevate the $\delta^{13}$C$_{DIC}$ of surface waters, they also tend to smooth the gradients produced by biology by working antagonistically to them. Despite this smoothing, biological fractionation drives strong gradients at the surface, which imparts unique $\delta^{13}$C signatures to the water masses that are carried into the interior. These insights have provided clear evidence of reduced ventilation rates in the deep ocean during glacial climates (Tagliabue et al., 2009; Menviel et al., 2017; Muglia et al., 2018).

$\delta^{15}$N is determined by biological processes that add or remove fixed forms of nitrogen. It therefore records the relative rates of sources and sinks within the marine nitrogen cycle (Brandes and Devol, 2002). Dinitrogen (N$_2$) fixation is the largest source of fixed nitrogen to the ocean, the bulk of which occurs in warm, sunlit surface waters and introduces nitrogen with a $\delta^{15}$N of approximately -1 ‰ (Sigman et al., 2009). Denitrification is the largest sink of fixed nitrogen and occurs in deoxygenated water columns and sediments. Denitrification fractionates strongly against ${}^{15}$N at $\sim$25 ‰ (Cline and Kaplan, 1975). Fractionation during denitrification is most strongly expressed in the water column where ample nitrate (NO$_3$) is available, making water column denitrification responsible for elevating global mean $\delta^{15}$N above the -1 ‰ of N$_2$ fixers (Brandes and Devol, 2002). Meanwhile, denitrification occurring in the sediments only weakly fractionates against ${}^{15}$N (Sigman et al., 2009), providing only a slight enrichment of $\delta^{15}$N above that introduced by N$_2$ fixation. Variations in $\delta^{15}$N can therefore tell us about global changes in the ratio of sedimentary to water column denitrification, with increases in $\delta^{15}$N associated with increases in the

proportion of denitrification occurring in the water column (Galbraith et al., 2013), but it can also reflect regional changes in $N_2$ fixation and denitrification (Ganeshram et al., 1995; Ren et al., 2009; Straub et al., 2013).

However, nitrogen isotopes are also subject to the effect of utilisation, which makes the interpretation of $\delta^{15}N$ more complicated. Basically, when nitrogen is abundant the preference for $^{14}N$ over $^{15}N$ increases but when nitrogen is limited this preference disappears (Altabet and Francois, 2001). Complete utilisation of nitrogen therefore reduces fractionation to 0 ‰. While this adds complexity, it also imbues $\delta^{15}N$ as a proxy of nutrient utilisation by phytoplankton. As nitrogen supply to phytoplankton is controlled by physical delivery from below, changes in $\delta^{15}N$ can be interpreted as changes in the physical supply (Studer et al., 2018). Phytoplankton fractionate against $^{15}N$ at ∼5 ‰ (Wada, 1980) when bioavailable nitrogen is abundant. If nitrogen is utilised to completion, which occurs in much of the low to mid latitude ocean, then no fractionation will occur and the $\delta^{15}N$ of organic matter will reflect the $\delta^{15}N$ of the nitrogen that was supplied (Sigman et al., 2009). However, in the case where nitrogen is not consumed towards completion, which occurs in zones of strong upwelling/mixing near coastlines, the equator and high latitudes, the bioavailable nitrogen pool will be enriched in $^{15}N$ as phytoplankton preferentially consume $^{14}N$. As the remaining bioavailable N is continually enriched in $^{15}N$ the organic matter that settles into sediments beneath a zone of incomplete nutrient utilisation will bear this enriched $\delta^{15}N$ signal. In combination with modelling (Schmittner and Somes, 2016), the $\delta^{15}N$ record is able to provide evidence for a more efficient utilisation of bioavailable nitrogen during glacial times (Martinez-Garcia et al., 2014; Kemeny et al., 2018) and a less efficient one during the Holocene (Studer et al., 2018).

Complimentary measurements of $\delta^{13}C$ and $\delta^{15}N$ provide powerful, multi-focal insights into oceanographic processes. $\delta^{13}C$ is largely a reflection of how water masses mix away the strong vertical and horizontal gradients enforced by biology, while $\delta^{15}N$ simultaneously reflects changes in the major sources and sinks of the marine nitrogen cycle and how effectively nutrients are consumed at the surface. However, the interpretation of these isotopes is often difficult. They are subject to considerable uncertainty because there are multiple processes that imprint on the measured values. Our goal is to equip version 1.0 of the CSIRO Mk3L-COAL Earth system model with oceanic $\delta^{13}C$ and $\delta^{15}N$ such that this model can be used for interpreting palaeoceanographic records. First, we introduce CSIRO Mk3L-COAL. Second, we detail the equations that govern the implementation of carbon and nitrogen isotopes. Third, we assess our simulated isotopes against contemporary measurements from both the water column and sediments and compare the model performance against other isotope-enabled models. Finally, as a first test of the model, we take the opportunity to document how changes in ecosystem functioning affect $\delta^{13}C$ and $\delta^{15}N$.

## 2 CSIRO Mk3L-COAL v1.0

The CSIRO Mk3L-COAL couples a computationally efficient climate system model (Phipps et al., 2013) with biogeochemical cycles in the ocean, atmosphere and land. The model is therefore based on the CSIRO Mk3L climate system model, where the "L" denotes that it is a low-resolution version of the CSIRO Mk3 model that contributed towards the Coupled Model Intercomparison Project Phase 3 (Meehl et al., 2007) and the Fourth Assessment Report of the Intergovernmental Panel on Climate Change (Solomon et al., 2007). See Smith (2007) for a complete discussion of the CSIRO family of climate models. The land biogeochemical component represents carbon, nitrogen and phosphorus cycles in the Community Atmosphere Biosphere Land

Exchange (CABLE) (Mao et al., 2011). The ocean component currently represents carbon, alkalinity, oxygen, nitrogen, phosphorus and iron cycles. The atmospheric component conserves carbon and alters its radiative properties according to changes in its carbon content. For this paper we focus on the ocean biogeochemical model (OBGCM).

Previous versions of the OBGCM have explored changes in oceanic properties under past (Buchanan et al., 2016), present (Buchanan et al., 2018) and future scenarios (Matear and Lenton, 2014, 2018). These studies demonstrate that the model can reproduce observed features of the global carbon cycle, nutrient cycling and organic matter cycling in the ocean. The OBGCM offers highly efficient simulations of these processes at computational speeds of $\sim$400 years per day when the ocean general circulation model (OGCM) is run offline (compared to $\sim$10 years per day in fully coupled mode). The ocean is made up of grid cells of $1.6°$ in latitude by $2.8°$ in longitude, with 21 vertical depth levels spaced by 25 metres at the surface and 450 metres in the deep ocean (Table 1). The OGCM timestep is one hour, while the OBGCM timestep is 1 day. The ability of the OBGCM to reproduce large-scale dynamical and biogeochemical properties of the ocean coupled with its fast computational speed makes the OBGCM useful as a tool for palaeoceanographic research.

## 2.1 Ocean biogeochemical model (OBGCM)

The OBGCM is equipped with 13 prognostic tracers (Figure 1). These can be grouped into carbon chemistry fields, oxygen fields, nutrient fields, age tracers and nitrous oxide ($N_2O$). Carbon chemistry fields include dissolved inorganic carbon (DIC), alkalinity (ALK), $DI^{13}C$ and radiocarbon ($^{14}C$). Radiocarbon is simulated according to Toggweiler et al. (1989). Oxygen fields include dissolved oxygen ($O_2$) and abiotic dissolved oxygen ($O_2^{abio}$), a purely physical tracer from which true oxygen utilisation (TOU) can be calculated (Duteil et al., 2013). Nutrient fields include phosphate ($PO_4$), dissolved bioavailable iron (Fe), nitrate ($NO_3$) and $^{15}NO_3$. Although we define the phosphorus and nitrogen tracers as their dominant species, being $PO_4$ and $NO_3$, these tracers can also be thought of as total dissolved inorganic phosphorus and nitrogen pools. Remineralisation, for instance, implicitly accounts for the process of nitrification from ammonium ($NH_4$) to $NO_3$ (Paulmier et al., 2009) and therefore implicitly includes $NH_4$ and nitrite ($NO_2$) within the $NO_3$ tracer. Age tracer fields include years since subduction from the surface ($Age_{gbl}$), and years since entering a suboxic zone where $O_2$ concentrations are less than 10 mmol m$^{-3}$ ($Age_{omz}$). Finally, $N_2O$ in $\mu$mol m$^{-3}$ is produced via nitrification and denitrification according to the temperature-dependent equations of Freing et al. (2012). All air-sea gas exchanges ($CO_2$, $^{13}CO_2$ $O_2$ and $N_2O$) and carbon speciation reactions are computed according to the Ocean Modelling Intercomparison Project phase 6 protocol (Orr et al., 2017).

Because the isotopes of carbon and nitrogen are influenced by biological processes and there is as yet no accepted standard for ecosystem model parameterisation in the community (see Hülse et al., 2017, for a more detailed discussion), we provide a thorough description of the ecosystem component of the OBGCM in appendix A. Default parameters for the OBGCM are further provided in appendix B. Briefly, the ecosystem model simulates the production, remineralisation and stoichiometry (elemental composition) of three types of primary producers: a general phytoplankton group, diazotrophs ($N_2$ fixers) and calcifiers.

**Figure 1.** A conceptual representation of the ocean biogeochemical model (OBGCM). The bottom panel shows organic matter cycling involving the isotopes of carbon and nitrogen. (1) Carbon chemistry reactions. (2) Air-sea gas exchange. (3) Biological uptake of nutrients and production of organic and inorganic matter. Particulate organic carbon (POC) is produced by the general phytoplankton group and $N_2$ fixers (diazotrophs), while particulate inorganic carbon (PIC) as calcium carbonate ($CaCO_3$) is produced by calcifiers. Export of POC by the general (G) phytoplankton group and $N_2$ fixers (D; diazotrophs) are herein referred to as $C_{org}^G$ and $C_{org}^D$ (see appendix A1), respectively. (4) Remineralisation of sinking organic matter under oxic and suboxic conditions. (5) Sedimentary oxic and suboxic remineralisation. (6) Nitrous oxide production and consumption.

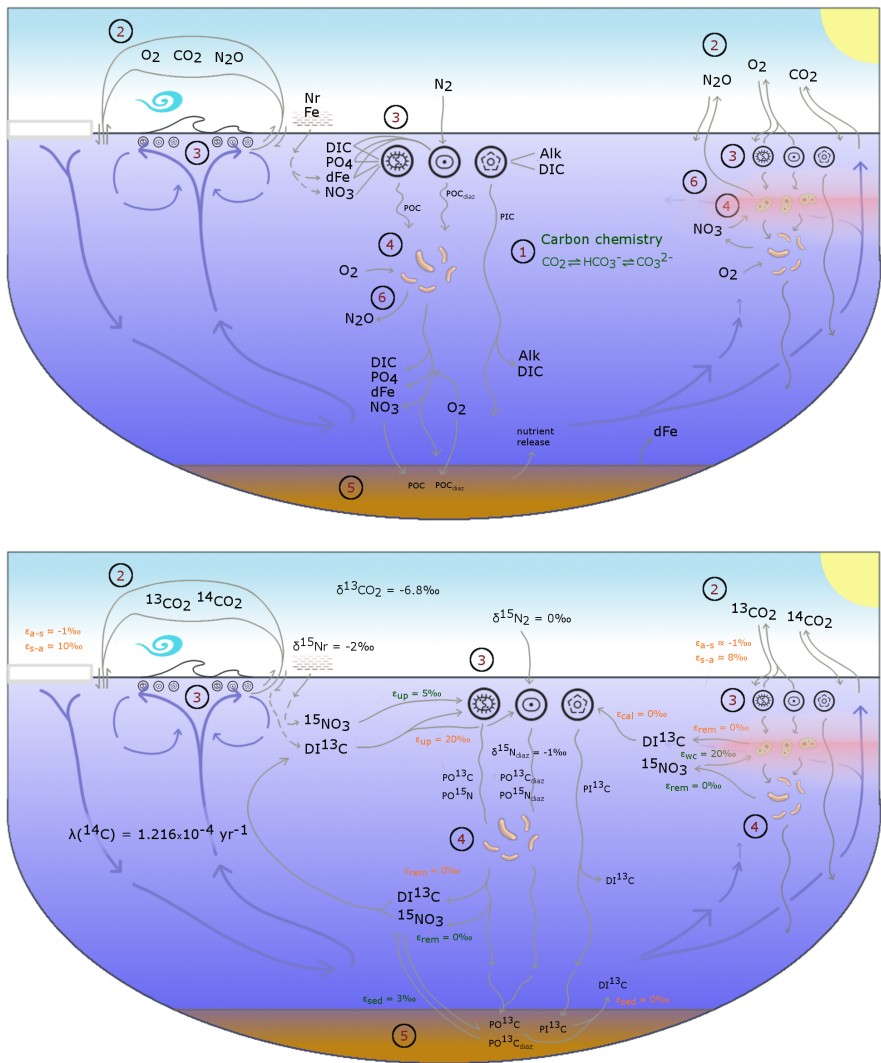

## 3 Carbon and nitrogen isotope equations

### 3.1 $\delta^{13}C$

The OBGCM explicitly simulates the fractionation of $^{13}C$ from the total DIC pool, where for simplicity we make the assumption that the total DIC pool represents the light isotope of carbon and is therefore $DI^{12}C$. Fractionation occurs during air-sea gas exchange, equilibrium reactions and biological consumption in the euphotic zone.

The **air-sea gas exchange** of $^{13}CO_2$ is calculated as the exchange of $CO_2$ with additional fractionation factors applied to the sea-air and air-sea components (Zhang et al., 1995; Orr et al., 2017). The flux of $^{13}CO_2$ across the air-sea interface, $F(^{13}CO_2)$, therefore takes the form of $CO_2$ with additional terms that convert to units of $^{13}C$ in both environments. Without any isotopic fractionation, the equation requires the gas piston velocity of carbon dioxide in m s$^{-1}$ ($k_{CO_2}$), the concentration of aqueous $CO_2$ in both mediums at the air-sea interface in mmol m$^{-3}$ ($CO_2^{air}$ and $CO_2^{sea}$), and the ratios of $^{13}C{:}^{12}C$ in both mediums ($R_{atm}$ and $R_{sea}$):

$$F(^{13}CO_2) = k_{CO_2} \cdot \left( CO_2^{air} \cdot R_{atm} - CO_2^{sea} \cdot R_{DIC} \right) \tag{2}$$

where,

$$R_{DIC} = \frac{DI^{13}C}{DI^{13}C + DI^{12}C}$$

$$R_{atm} = \frac{^{13}CO_2}{^{13}CO_2 + ^{12}CO_2} = 0.011164381$$

A transfer of $^{13}C$ into the ocean is therefore positive and an outgassing is negative. The $R_{atm}$ is set to a preindustrial atmospheric $\delta^{13}C$ of -6.48 ‰ (Friedli et al., 1986).

The fractionation of carbon isotopes during air-sea exchange involves three components. These are ($\epsilon_k^C$) a kinetic fractionation that occurs during transfer of gaseous $CO_2$ into or out of the ocean, ($\epsilon_{aq \leftarrow g}^C$) a fractionation that occurs as gaseous $CO_2$ becomes aqueous $CO_2$ (is dissolved in solution), and ($\epsilon_{DIC \leftarrow g}^C$) an equilibrium isotopic fractionation as carbon speciates into dissolved inorganic carbon (DIC) constituents ($H_2CO_2 \Leftrightarrow HCO_3^- \Leftrightarrow CO_3^{2-}$). The kinetic fractionation during transfer, $\epsilon_k^C$, is constant at 0.99912, thus reducing the $\delta^{13}C$ of carbon entering the ocean by 0.88 ‰. Conversely, carbon outgassing increases the $\delta^{13}C$ of the ocean. The fractionation during dissolution ($\epsilon_{aq \leftarrow g}^C$) and speciation ($\epsilon_{DIC \leftarrow g}^C$) are both dependent on temperature. Fractionation during speciation is also dependent on the fraction of $CO_3^{2-}$ relative to total DIC ($f_{CO_3^{2-}}$). These fractionation factors are parameterised as:

$$\epsilon_{aq \leftarrow g}^C = \frac{0.0049 \cdot T - 1.31}{1000} + 1 \tag{3}$$

$$\epsilon_{DIC \leftarrow g}^C = \frac{0.0144 \cdot T \cdot f_{CO_3^{2-}} - 0.107 \cdot T + 10.53}{1000} + 1 \tag{4}$$

Dissolution of $CO_2$ into seawater ($\epsilon_{aq \leftarrow g}^C$) therefore preferences the lighter isotope and lowers $\delta^{13}C$ by between 1.32 and 1.14 ‰, while the speciation of gaseous $CO_2$ into DIC instead preferences the heavier isotope and raises $\delta^{13}C$ by between 10.7 and 6.8 ‰ for temperatures between -2 and 35 °C.

These fractionation factors are applied to the gaseous exchange of $CO_2$ (Eq. (2)) to calculate carbon isotopic fractionation.

$$F(^{13}CO_2) = k \cdot \epsilon_k^C \cdot \epsilon_{aq \leftarrow g}^C \cdot \left( CO_2^{air} \cdot R_{atm} - \frac{CO_2^{sea} \cdot R_{DIC}}{\epsilon_{DIC \leftarrow g}^C} \right) \tag{5}$$

Because fractionation to aqueous $CO_2$ from DIC ($\epsilon_{aq \leftarrow DIC}^{C}$) is equal to $\frac{\epsilon_{aq \leftarrow g}^{C}}{\epsilon_{DIC \leftarrow g}^{C}}$, a strong preference to hold the heavy isotope in solution exists ($\epsilon_{aq \leftarrow DIC}^{C}$ = -11.9 to -7.9 ‰ between -2 and 35 °C). Aqueous carbon that is transferred to the atmosphere is hence depleted in $^{13}$C. It is therefore the equilibrium fractionation associated with carbon speciation that is largely responsible for bolstering the oceanic $\delta^{13}$C signature above the atmospheric signature, as it tends to shift $^{13}$C towards the oxidised species

($CO_3^{2-}$), a tendency that strengthens under cooler conditions.

In the default version of CSIRO Mk3L-COAL v1.0, the fractionation of carbon during **biological uptake** ($\epsilon_{bio}^{^{13}C}$) is set at 21 ‰ for general phytoplankton, 12 ‰ for diazotrophs (e.g. Carpenter et al., 1997) and at 2 ‰ for the formation of calcite (Ortiz et al., 1996). However, a variable fractionation rate for the general phytoplankton group may be activated and depends on the aqueous $CO_2$ concentration ($CO_2(aq)$ in mmol m$^{-3}$) and the growth rate ($\mu$ in day$^{-1}$, as a function of temperature and

limiting resources) following Tagliabue and Bopp (2008):

$$\epsilon_{bio}^{^{13}C} = \left(0.371 - \frac{\mu}{CO_2(aq)}\right)/0.015 \tag{6}$$

An upper bound of 25 ‰ exists within Eq. (6) when $\frac{\mu}{CO_2(aq)}$ approaches zero, but a lower bound does not. We chose to limit $\epsilon_{bio}^{^{13}C}$ to a minimum of 15 ‰ given the reported variations of $\epsilon_{bio}^{^{13}C}$ from culture studies (e.g. Laws et al., 1995).

$$\epsilon_{bio}^{^{13}C} = \max\left(15, \epsilon_{bio}^{^{13}C}\right) \tag{7}$$

Biological fractionation of $^{13}$C is then applied to the uptake and release of organic carbon.

$$\Delta DI^{13}C = R_{DIC} \cdot C_{org} \cdot \left(1 - \frac{\epsilon_{bio}^{^{13}C}}{1000}\right) \tag{8}$$

Because biological fractionation is strong for the general phytoplankton group, which dominates export production throughout most of the ocean, this imparts a negative $\delta^{13}$C signature to the deep ocean. Subsequent remineralisation releases DIC with no fractionation. Finally, the concentration of DI$^{13}$C is converted into a $\delta^{13}$C via:

$$\delta^{13}C = \left(\frac{DI^{13}C}{DIC} \cdot \frac{1}{0.0112372} - 1\right) \cdot 1000 \tag{9}$$

where 0.0112372 is the Pee Dee Belemnite standard (Craig, 1957).

### 3.2   $\delta^{15}$N

The OBGCM explicitly simulates the fractionation of $^{15}$N from the pool of bioavailable nitrogen. For simplicity we treat this bioavailable pool as nitrate ($NO_3$), where total $NO_3$ is the sum of $^{15}NO_3$ and $^{14}NO_3$. We therefore chose to ignore fractionation

during reactions involving ammonium, nitrite and dissolved organic nitrogen, which can vary in their isotopic composition independent of $NO_3$ but represent a small fraction of the bioavailable pool of nitrogen.

The isotopic signatures of $N_2$ fixation and atmospheric deposition, and the fractionation during water column denitrification ($\epsilon_{wc}^{^{15}N}$) and sedimentary denitrification ($\epsilon_{sed}^{^{15}N}$) determine the global $\delta^{15}$N of $NO_3$ (Brandes and Devol, 2002). Biological assimilation ($\epsilon_{bio}^{^{15}N}$) and remineralisation are internal exchanges of the oceanic nitrogen cycle and affect the distribution of $\delta^{15}NO_3$.

$N_2$ fixation and atmospheric deposition introduce $^{15}NO_3$ to the ocean with $\delta^{15}N$ values of -1 ‰ and -2 ‰, respectively, while biological assimilation, water column denitrification and sedimentary denitrification fractionate against $^{15}NO_3$ at 5 ‰, 20 ‰ and 3 ‰, respectively (Sigman et al., 2009, Fig. 1).

The accepted standard $^{15}N$:$^{14}N$ ratio used to measure variations in nature is the average atmospheric $^{15}N$:$^{14}N$ ratio of 0.0036765. To minimise numerical errors caused by the OGCM, we set the atmospheric standard to 1. This scales up the $^{15}NO_3$ such that a $\delta^{15}N$ value of 0 ‰ was equivalent to an $^{15}N$:$^{14}N$ ratio of 1:1.

Because we simulate $NO_3$ and $^{15}NO_3$ as tracers, our calculations require solving for an implicit pool of $^{14}NO_3$ during each reaction involving $^{15}NO_3$. The introduction of $NO_3$ at a fixed $\delta^{15}N_{NO_3}$ of -1 ‰ due to remineralisation of $N_2$ fixer biomass provides a simple example with which we can begin to describe our equations. Setting the isotopic value of newly fixed $NO_3$ to -1 ‰ is simple because it removes any complications associated with fractionation. We note, however, that in reality the nitrogenase enzyme does fractionate during its conversion of aqueous $N_2$ (+0.7 ‰) to ammonium and the biomass that is subsequently produced can vary substantially depending of the type of nitrogenase enzyme used (vanadium versus molybdenum based) (McRose et al., 2019). However, we choose to implicitly account for these transformations and considerably simplify them by setting the $\delta^{15}N$ of $N_2$ fixer biomass equal to -1 ‰, which reflects the biomass of $N_2$ fixers associated with the more common Mo-nitrogenase.

A $\delta^{15}N_{NO_3}$ of -1 ‰ is equivalent to a $^{15}N$:$^{14}N$ ratio of 0.999 in our approach where 0 ‰ equals a 1:1 ratio of $^{15}N$:$^{14}N$. If the amount of $NO_3$ being added is known alongside its $^{15}N$:$^{14}N$ ratio, in this case 0.999 for $N_2$ fixation, we are able to calculate how much $^{15}NO_3$ is added. We begin with two equations that describe the system.

$$NO_3 = {}^{15}NO_3 + {}^{14}NO_3 \tag{10}$$

$$\delta^{15}N_{NO_3} = \left( \frac{{}^{15}NO_3/{}^{14}NO_3}{{}^{15}N_{std}/{}^{14}N_{std}} - 1 \right) \cdot 1000 \tag{11}$$

Ultimately, we need to solve for the change in $^{15}NO_3$ associated with an introduction of $NO_3$ by **$N_2$ fixation**. Our knowns are the change in $NO_3$, the $\delta^{15}N$ of that $NO_3$, and the $^{15}N_{std}/^{14}N_{std}$. Our two unknowns are $^{15}NO_3$ and $^{14}NO_3$. We must solve for $^{14}NO_3$ implicitly by describing it according to $^{15}NO_3$ by rearranging Eq. (11).

$$^{14}NO_3 = {}^{15}NO_3 / \left( \left( \frac{\delta^{15}N_{NO_3}}{1000} + 1 \right) \cdot {}^{15}N_{std}/{}^{14}N_{std} \right) \tag{12}$$

This allows us to replace the $^{14}NO_3$ term in Eq. (10), such that

$$NO_3 = {}^{15}NO_3 + {}^{15}NO_3 / \left( \left( \frac{\delta^{15}N_{NO_3}}{1000} + 1 \right) \cdot {}^{15}N_{std}/{}^{14}N_{std} \right) \tag{13}$$

In our example of $N_2$ fixation we know the $\delta^{15}N$ of the newly added $NO_3$ as being -1 ‰. We also know $^{15}N_{std}/^{14}N_{std}$ as equal to 1:1, or 1. Our equation is simplified.

$$NO_3 = {}^{15}NO_3 + {}^{15}NO_3 / 0.999 \tag{14}$$

We can now solve for $^{15}NO_3$ by rearranging the equation.

$$^{15}NO_3 = \frac{0.999 \cdot NO_3}{1 + 0.999}. \tag{15}$$

The same calculation is applied to $NO_3$ addition via **atmospheric deposition** except at a constant fraction of 0.998 ($\delta^{15}N = -2$ ‰), and can be applied to any addition or subtraction of $^{15}NO_3$ relative to $NO_3$ where the isotopic signature is known.

Fractionating against $^{15}NO_3$ during **biological assimilation** ($\epsilon_{bio}^{^{15}N}$), **water column denitrification** ($\epsilon_{wc}^{^{15}N}$) and **sedimentary denitrification** ($\epsilon_{sed}^{^{15}N}$) involves more considerations because we must account for the preference of $^{14}NO_3$ over $^{15}NO_3$. We begin with an $\epsilon$ of 5 ‰ for biological assimilation. This is equivalent to an $^{15}NO_3:^{14}NO_3$ ratio of 0.995 when our atmospheric standard is equal to 1:1 using the following equation.

$$\epsilon = \left( \frac{^{15}N/^{14}N}{^{15}N_{std}/^{14}N_{std}} - 1 \right) \cdot 1000 \tag{16}$$

Note that a positive $\epsilon$ value returns an $^{15}NO_3:^{14}NO_3$ ratio < 1, while a negative $\delta^{15}N$ in the previous example with $N_2$ fixation also returned an $^{15}NO_3:^{14}NO_3$ ratio < 1. This works because the reactions are in opposite directions. $N_2$ fixation adds $NO_3$, while assimilation removes $NO_3$. This means that 0.995 units of $^{15}NO_3$ are assimilated per unit of $^{14}NO_3$. As we have seen, a more useful way to quantify this is per unit of $NO_3$ assimilated into organic matter. Using Eq. (15), we find that ∼0.4987 units of $^{15}NO_3$ and ∼0.5013 units of $^{14}NO_3$ are assimilated per unit (1.0) of $NO_3$ when $\epsilon$ equals 5 ‰. Biological assimilation therefore leaves slightly more $^{15}N$ in the unused $NO_3$ pool relative to $^{14}N$, which increases the $\delta^{15}N$ of $NO_3$ while creating more $^{15}N$-deplete organic matter ($\delta^{15}N_{org}$).

However, we must also account for the effect that $NO_3$ availability has on fractionation. The preference of $^{14}NO_3$ over $^{15}NO_3$ strongly depends on the availability of $NO_3$, such that when $NO_3$ is abundant the preference for the lighter isotope will be strongest. This preference (fractionation) becomes weaker as $NO_3$ is depleted because cells will absorb any $NO_3$ that is available irrespective of its isotopic composition (Mariotti et al., 1981). Thus, as $NO_3$ is utilised, $u$, towards 100 % of its availability ($u = 1$), the fractionation against $^{15}NO_3$ decreases to an $\epsilon$ of 0 ‰. This means that when $u$ is equal to 1, no fractionation occurs and equal parts $^{15}N$ and $^{14}N$ (0.5:0.5 per unit $NO_3$) are assimilated. As we are interested in long timescales, we chose the accumulated product equations (Altabet and Francois, 2001) to approximate this process, where:

$$u = \min\left( 0.999, \max\left( 0.001, \frac{N_{org}}{NO_3} \right) \right) \tag{17}$$

$$\epsilon_u = \epsilon \cdot \frac{1-u}{u} \cdot \ln(1-u) \tag{18}$$

For numerical reasons, we limited the domain of $u$ to (0.001,0.999) rather than (0,1), such that the utilisation-affected $\epsilon_u$ has a range of -4.997 to 0.035 ‰ for an $\epsilon$ of 5 ‰. $\epsilon_u$ is then converted into ratio units by dividing by 1000, and added to the ambient $^{15}N:^{14}N$ of $NO_3$ in the reactant pool to determine the $^{15}N:^{14}N$ of the product. In this case, it is the $^{15}N:^{14}N$ of newly created organic matter, but could also be unused $NO_3$ effluxed from denitrifying cells in the case for denitrification.

$$^{15}N_{org}:^{14}N_{org} = {}^{15}NO_3:^{14}NO_3 + \epsilon_u \tag{19}$$

We then solve for how much $^{15}NO_3$ is assimilated into organic matter using Eq. (15) because we now know the change in $NO_3$ ($\Delta NO_3$) and the $^{15}N:^{14}N$ of the product, which is $^{15}N_{org}/^{14}N_{org}$ in our example of biological assimilation.

$$\Delta^{15}NO_3 = \frac{^{15}N_{org}/^{14}N_{org} \cdot \Delta NO_3}{1 + {}^{15}N_{org}/^{14}N_{org}} \tag{20}$$

Here, the change in $^{15}NO_3$ is equivalent to that assimilated into organic matter. Following assimilation into organic matter, the release of $^{15}NO_3$ through the water column during remineralisation occurs with no fractionation, such that the same $\delta^{15}N$ signature is released throughout the water column.

We apply these calculations to each reaction in the nitrogen cycle that involves fractionation (assimilation, water column denitrification and sedimentary denitrification). They could be applied to any form of fractionation process with knowledge of $\epsilon$, the isotopic ratio of the reactant, the amount of reactant that is used, and the total amount of reactant available.

## 4   Model performance

CSIRO Mk3L-COAL adequately reproduces the large-scale thermohaline properties and circulation of the ocean under pre-industrial conditions in numerous prior studies (Phipps et al., 2013; Matear and Lenton, 2014; Buchanan et al., 2016, 2018). Rather than reproduce these studies, we concentrate here on how the biogeochemical model performs relative to measurements of $\delta^{13}C$ and $\delta^{15}N$ in the water column (Eide et al., 2017, $\delta^{15}N_{NO_3}$ data courtesy of The Sigman Lab at Princeton University) and in the sediments (Tesdal et al., 2013; Schmittner et al., 2017). We make these model-data comparisons alongside other isotope-enabled ocean general circulation models (Table 1).

All analyses of model performance were undertaken using the default parameterisation of the biogeochemical model, which is summarised in the tables of appendix B. Each experiment was run towards steady-state under preindustrial atmospheric conditions over many thousands of years. All results presented in this paper therefore reflect tracers that have achieved an equilibrium solution. We present annual averages of the equilibrium state in the following analysis.

**Table 1.** Models assessed against isotope data. The University of Victoria - Model of Ocean Biogeochemistry and Isotopes (UVic-MOBI) fields taken from Schmittner and Somes (2016). Pelagic Interactions Scheme for Carbon and Ecosystem Studies (PISCES) fields provided by Laurent Bopp (Bopp et al., *in prep for Geoscientific Model Development*). LOch–Vecode-Ecbilt-CLio-agIsm Model (LOVECLIM) fields taken from Menviel et al. (2017). The isotope-enabled Community Earth System Model (iCESM) fields for $\delta^{13}C$ (low resolution) provided by Alexandra Jahn (Jahn et al., 2015) and those for $\delta^{15}N$ (high resolution) provided by Simon Yang (Yang and Gruber, 2016). PISCES and CESM model resolutions have a range of longitude/latitude spacings to reflect regions of finer resolution, including the equator and polar regions.

| Model | Group | Lon × Lat | Vertical levels |
|---|---|---|---|
| CSIRO Mk3L-COAL | Commonwealth Scientific and Industrial Research Organisation | $2.8125° \times {\sim}1.6°$ | 21 |
| UVic-MOBI | Oregon State University / GEOMAR Kiel | $3.6° \times 1.8°$ | 19 |
| PISCES | Nucleus for European Modelling of the Ocean | $1° \times {\sim}0.05\text{-}0.95°$ | 75 |
| LOVECLIM | Université catholique de Louvain | $3° \times 3°$ | 20 |
| iCESM-low | National Center for Atmospheric Research | $\leq 3.4° \times {\sim}3.6°$ | 60 |
| iCESM-high | National Center for Atmospheric Research | $\leq 1.1° \times \leq 0.6°$ | 60 |

## 4.1 $\delta^{13}$C of dissolved inorganic carbon ($\delta^{13}$C$_{DIC}$)

The recent reconstruction of preindustrial $\delta^{13}$C$_{DIC}$ by Eide et al. (2017) provides a large dataset for comparison. We chose this dataset over the compilation of point location water column data of Schmittner et al. (2017) because it offers a gridded product where short-term and small-scale variability are smoothed, making for more appropriate comparison with model output.

Predicted values of $\delta^{13}$C$_{DIC}$ from CSIRO Mk3L-COAL broadly replicated the preindustrial distribution. The predicted global mean of 0.41 ‰ reflected that of the reconstructed mean of 0.42 ‰ (Table 2). Spatial agreement was acceptable with a global correlation of 0.80 ($G$ marker in Fig. 2). Regionally, the Southern Ocean performed well with the lowest RMS error of 0.42 ‰, while a greater degree of disagreement in the values of $\delta^{13}$C$_{DIC}$ existed in the middle and lower latitudes of each major basin, particularly in the Atlantic where model-data agreement (correlation, RMS error and normalised standard

deviation) was poorest. Subsurface $\delta^{13}$C$_{DIC}$ was too low in the tropics of the major basins by $\sim$0.2 ‰, and too high in the North Pacific and North Atlantic by 0.4 to 0.6 ‰ (Fig. 3).

     These inconsistencies were likely related to physical and biological limitations of CSIRO Mk3L-COAL. $\delta^{13}$C$_{DIC}$ in subsurface tropical waters was too low because restricted horizontal mixing and high carbon export drove very negative $\delta^{13}$C$_{DIC}$ values. The very negative $\delta^{13}$C values were associated with very large oxygen minimum zones and were thus a product of poorly

represented, fine-scale equatorial dynamics. Coarse resolution OGCMs are known to have weak equatorial undercurrents that lead to oxygen minimum zones that are too large (Matear and Holloway, 1995; Oschlies, 2000) and CSIRO Mk3L-COAL is no exception. Alternatively, the large oxygen minimum zones could be due to our conservative treatment of organic matter remineralisation (appendix A), where remineralisation is prevented when $O_2$ and $NO_3$ are unavailable. Organic matter therefore falls deeper into the interior through oxygen-deficient zones, leading to their vertically expansion. Almost certainly, however, it

was the poorly represented dynamics within the Pacific basin that were responsible for high $\delta^{13}$C$_{DIC}$ in the subsurface North Pacific, which contains low $O_2$, low $\delta^{13}$C$_{DIC}$ water due to northward transport from the tropics.

     Another inconsistency was a positive bias in the upper 200-500 metres, with values exceeding 2 ‰ in many areas of the lower latitudes. However, values as high as 2 ‰ have been measured in the upper 500 metres of the Indo-Pacific (Schmittner et al., 2017). Given the difficulties associated with accounting for the Suess Effect (invasion of isotopically light fossil fuel

$CO_2$) it is possible that the upper ocean values of Eide et al. (2017) underestimate the preindustrial $\delta^{13}$C$_{DIC}$ surface field.

     It is also equally possible that a fixed biological fractionation ($\epsilon^{13C}_{bio}$) of 21 ‰ may have driven unrealistic enrichment in the simulated field. High growth rates, such as occurs in the tropical regions, are thought to lower the strength of fractionation during carbon fixation (Laws et al., 1995). To explore the possibility of model-data mismatch caused by our choice to fix $\epsilon^{13C}_{bio}$ at 21 ‰, we implemented biological fractionation that is dependent on phytoplankton growth rate and aqueous $CO_2$ concentration

(Eq. 6). We found the implementation of a variable $\epsilon^{13}_{bio}$ reduced high values in the upper part of the low latitude ocean, but that this reduction was small (Fig. 4). The overwhelming effect was an increase in $\delta^{13}$C$_{DIC}$ throughout the interior, itself caused by weaker fractionation in the tropical ocean. Global mean $\delta^{13}$C$_{DIC}$ subsequently increased by 0.25 ‰. Meanwhile, model skill was unaffected (see CSIRO Mk3L-COAL (vary-$\epsilon^{13C}_{bio}$) in Fig. 2). Neither fixed nor variable biological fractionation could reproduce the low upper ocean values of the data.

**Figure 2.** Global and regional fits between data (Eide et al., 2017) and simulated $\delta^{13}C$ of dissolved inorganic carbon displayed as Taylor Diagrams (Taylor, 2001). Shading of the markers represent normalised bias. G = Global; S = Southern Ocean (90°S - 40°S); A = Atlantic (40°S - 70°N); P = Pacific (40°S - 70°N); I = Indian (40°S - 70°N). Measures of fit do not include the Arctic nor the upper 200 metres of the water column. All data was regridded onto the CSIRO Mk3L-COAL gridspace before comparison.

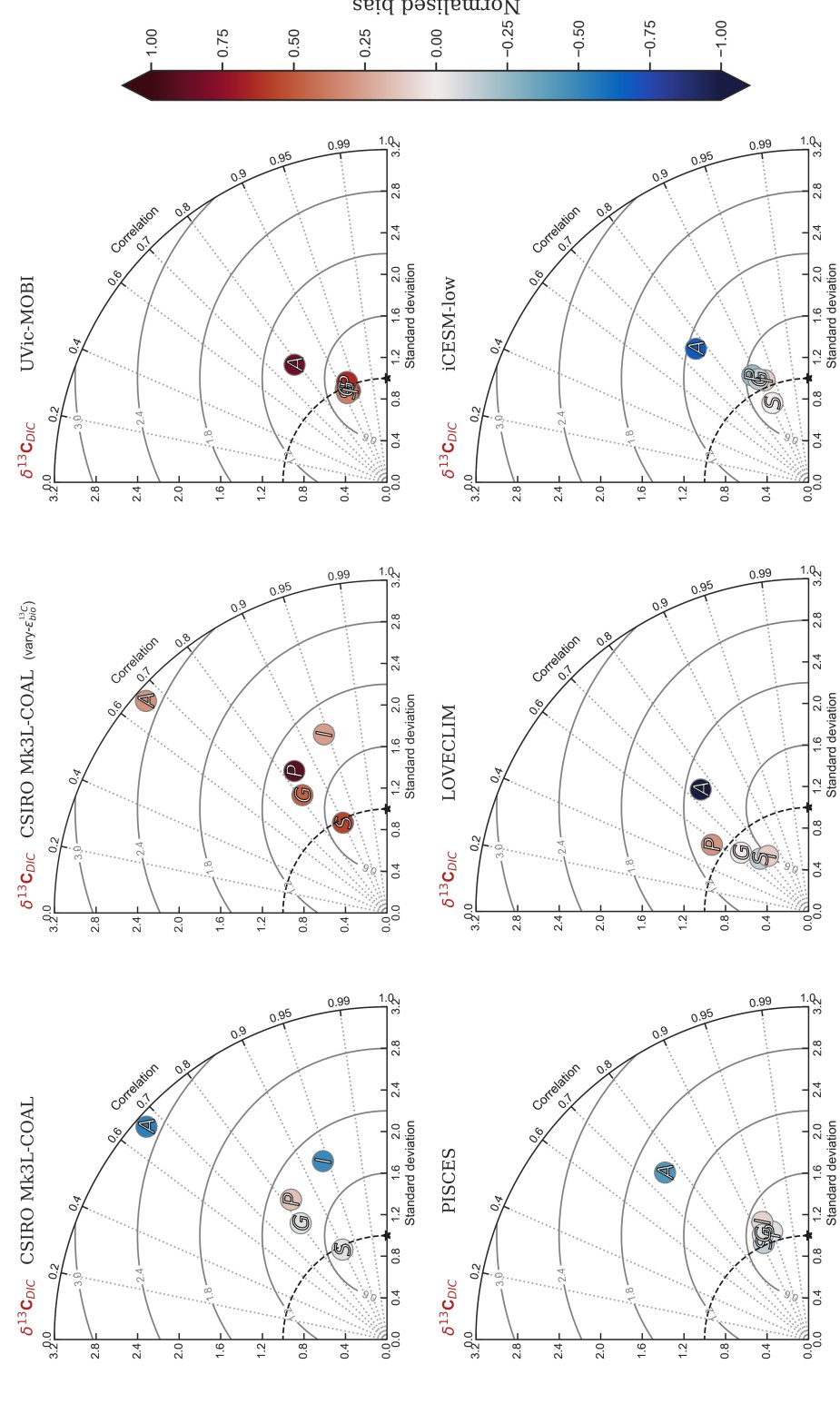

**Table 2.** Comparison of global and region mean $\delta^{13}C_{DIC}$ between observations (Eide et al., 2017) and model simulations. Means are annual averages and do not include the Arctic nor the upper 200 metres of the water column. All data was regridded onto the CSIRO Mk3L-COAL gridspace.

| | Global | Southern Ocean | Atlantic | Pacific | Indian |
|---|---|---|---|---|---|
| **Eide et al. (2017)** | **0.44 ‰** | **0.61 ‰** | **0.97 ‰** | **0.11 ‰** | **0.39 ‰** |
| CSIRO Mk3L-COAL | 0.41 ‰ | 0.61 ‰ | 0.87 ‰ | 0.17 ‰ | 0.21 ‰ |
| CSIRO Mk3L-COAL (vary-$\epsilon_{bio}^{13C}$) | 0.67 ‰ | 0.81 ‰ | 1.04 ‰ | 0.47 ‰ | 0.53 ‰ |
| LOVECLIM | 0.44 ‰ | 0.57 ‰ | 0.74 ‰ | 0.23 ‰ | 0.45 ‰ |
| UVic-MOBI | 0.65 ‰ | 0.74 ‰ | 1.15 ‰ | 0.37 ‰ | 0.66 ‰ |
| PISCES | 0.40 ‰ | 0.57 ‰ | 0.89 ‰ | 0.09 ‰ | 0.44 ‰ |
| iCESM-low | 0.37 ‰ | 0.61 ‰ | 0.84 ‰ | 0.01 ‰ | 0.45 ‰ |

**Figure 3.** Zonal mean observed (top) and modelled (bottom) $\delta^{13}C$ of DIC produced by CSIRO Mk3L-COAL for each major basin. The red dashed line marks the upper 175 metres and is used for comparison between observed and modelled distributions. Replicate figures for the other models are available in the supplement.

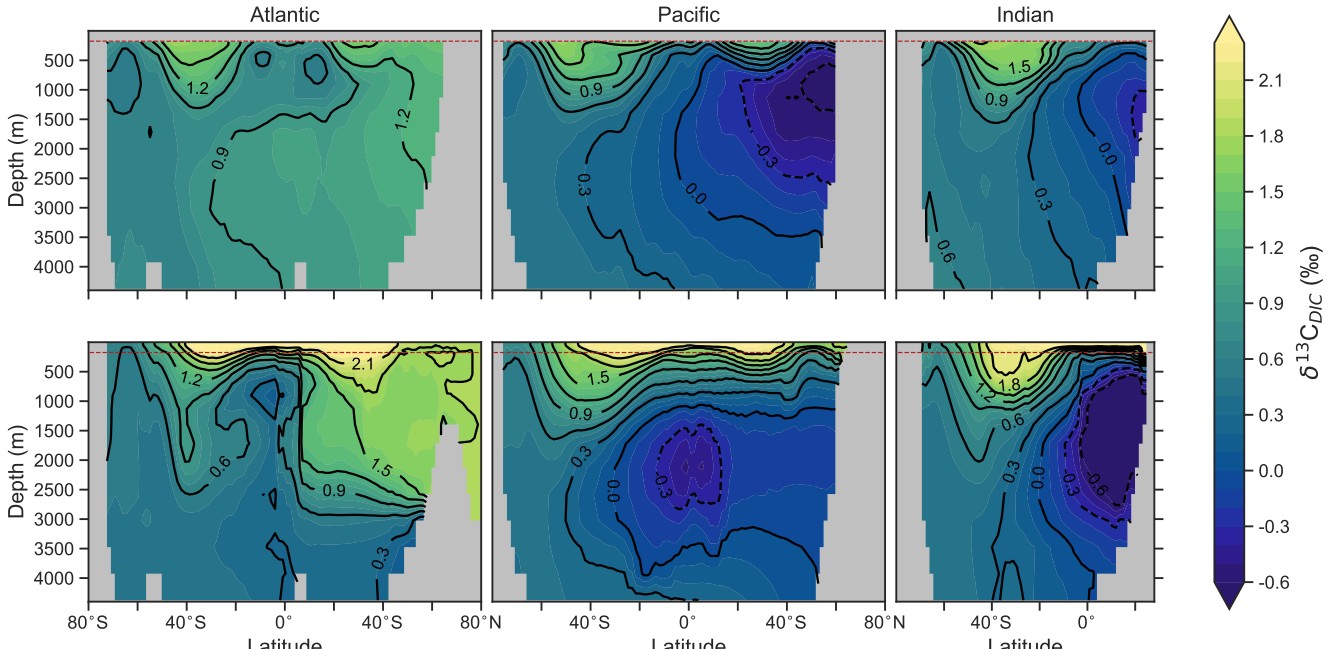

**Figure 4.** The introduction of variable carbon fractionation by phytoplankton (top) and the consequent change in $\delta^{13}C_{DIC}$ represented as a zonal mean (bottom) relative to a case where $\epsilon_{bio}^{13C}$ is fixed at 21 ‰.

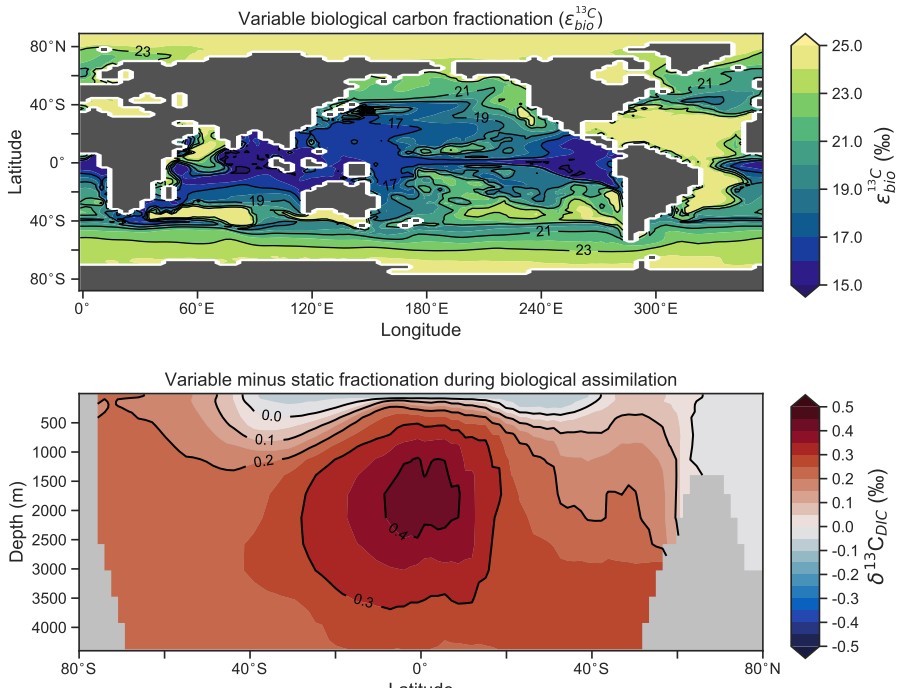

It is helpful to place our predicted $\delta^{13}C_{DIC}$ alongside those of other global ocean models (Fig. 2; Table 2), both for skill assessment and to further understand the cause of the positive bias in the upper ocean. We take annually averaged, preindustrial $\delta^{13}C_{DIC}$ distributions from the UVic-MOBI, PISCES, LOVECLIM and iCESM-low biogeochemical models, most of which have been used in significant palaeoceanographic modelling studies (Menviel et al., 2017; Tagliabue et al., 2009; Schmittner and Somes, 2016). Predicted $\delta^{13}C_{DIC}$ performs adequately in CSIRO Mk3L-COAL relative to these state-of-the-art models. LOVECLIM showed good fit in terms of global and regional means (Table 2), but had lower correlations (Fig. 2), suggesting that its values were accurate but its distribution biased. UVic-MOBI had high correlations, but it consistently overestimated the preindustrial field by ~0.2 ‰. Interestingly, the bias of UVic-MOBI, which treats biological fractionation as a function of growth rate and aqueous $CO_2$, is similar to CSIRO Mk3L-COAL when this form of fractionation was activated (vary-$\epsilon_{bio}^{13C}$). PISCES and iCESM-low were the best performing models, equally demonstrating high correlations, low biases, accurate regional and global means and the lowest RMS errors. This is perhaps not surprising considering the significantly finer vertical resolutions of these OGCMs and their more complex horizontal grid structure that enables an improved representation of ocean dynamics (Table 1). However, all models performed most poorly in the Atlantic Ocean, with poor correlations, high variability and greater biases.

Returning to the consistent positive bias in the upper ocean, most models (except iCESM-low) predicted upper ocean $\delta^{13}C_{DIC} \geq 2.0$ ‰ (Supplementary Figures S1, S2, S3 and S4) similar to CSIRO Mk3L-COAL. As each model has a unique representation of the marine ecosystem and consequently a unique treatment of biological fractionation, the common prediction of high upper ocean $\delta^{13}C$ once more suggests that the upper ocean values between 200 and 500 metres of Eide et al. (2017) may be too low. The underestimation of $\delta^{13}C_{DIC}$ may be due to a neglect of biology introducing anthropogenic, isotopically-depleted carbon to surface and subsurface layers via remineralisation (the biological Suess effect). This would in turn suggest that a higher global mean of 0.73 ‰ generated from a global compilation of foraminiferal $\delta^{13}C$ (Schmittner et al., 2017) is perhaps a more accurate representation of preindustrial $\delta^{13}C$ values.

Overall, CSIRO Mk3L-COAL performed acceptably in terms of its mean values and correlations, but had consistently greater RMS errors in major basins outside of the Southern Ocean. This indicates that CSIRO Mk3L-COAL exaggerated regional minima and maxima as discussed. Despite the regional biases of CSIRO Mk3L-COAL, the comparison demonstrates that all models have strengths and weaknesses. Given its low resolution and computational efficiency, CSIRO Mk3L-COAL performs adequately among other biogeochemical models in its simulation of $\delta^{13}C_{DIC}$.

## 4.2 $\delta^{13}C$ of *Cibicides* foraminifera ($\delta^{13}C_{Cib}$)

We extended our assessment of modelled $\delta^{13}C_{DIC}$ by comparing it to a compilation of benthic $\delta^{13}C$ measured within the calcite of foraminifera from the genus *Cibicides* (Schmittner et al., 2017), a genus on which much of the palaeoceanographic $\delta^{13}C$ records are based. For this comparison, we adjusted our predicted $\delta^{13}C_{DIC}$ to predicted $\delta^{13}C_{Cib}$ using the linear dependence on carbonate ion concentration and depth suggested by Schmittner et al. (2017):

$$\delta^{13}C_{Cib} = 0.45 + \delta^{13}C_{DIC} - 2.2 \times 10^{-3} \cdot CO_3 - 6.6 \times 10^{-5} \cdot z \tag{21}$$

This adjustment accounts for slight fractionation during incorporation of DIC into foraminiferal calcite and is found to be partly explained by the concentration of $CO_3^{2-}$ ions and pressure. A one to one comparison between $\delta^{13}C_{DIC}$ and $\delta^{13}C_{Cib}$ hence introduces some degree of error since this fractionation is not accounted for. Because we are interested in applying simulated $\delta^{13}C_{DIC}$ to a palaeoceanographic context, we must first be able to convert our simulated $\delta^{13}C_{DIC}$ to $\delta^{13}C_{Cib}$ in an effort to make better comparisons, particularly as the distribution of $CO_3^{2-}$ is subject to change. By adjusting our three-dimensional $\delta^{13}C_{DIC}$ output using Eq. (21), we attain predicted $\delta^{13}C_{Cib}$ (see inset entitled "Calibration" in Fig. 5). For good measure, we also computed measures of statistical fit for a traditional one to one comparison between $\delta^{13}C_{DIC}$ and $\delta^{13}C_{Cib}$ to assess the benefit of the calibration.

Measured $\delta^{13}C_{Cib}$ data from Schmittner et al. (2017) were binned into model grid boxes and averaged for the comparison. Those measurements that fell within the OGCM's land mask were excluded. Transfer and averaging onto the coarse resolution OGCM grid reduced the number of points for comparison from 1,763 to 690, lowered the mean of measured $\delta^{13}C_{Cib}$ from 0.76 ‰ to 0.52 ‰ and reduced the absolute range from -0.9→2.1 to -0.7→2.1.

Adjusted $\delta^{13}C_{Cib}$ using Eq. (21) showed good fit to measured $\delta^{13}C_{Cib}$ given the sparsity of data, with a global correlation of 0.64, a mean of 0.57 ‰ and an RMS error of 0.63 ‰ (Table 3). If a one to one relationship between $\delta^{13}C_{DIC}$ and

$\delta^{13}C_{Cib}$ was used, the global correlation was not affected and only slightly worse skill was detected in mean, RMS error and standard deviation. Accounting for the regional influence of carbonate ion concentration and depth was therefore beneficial, likely because very negative and positive values were slightly adjusted towards the mid-range (inset in Fig. 5), but this was not necessary for an adequate comparison. This conclusion was also reached by Schmittner et al. (2017). Likewise, implementing variable fractionation by phytoplankton ($\epsilon_{bio}^{13C}$) had little effect except to increase values and slightly improve measures of skill (Table 3). Of the 690 data points used in the comparison, 419 fell within the error around what could be considered a good fit (Shaded red area in Fig. 5). The error was taken as 0.29 ‰, and represents the standard deviation associated with the relationship between $\delta^{13}C_{DIC}$ with $\delta^{13}C_{Cib}$ measurements (Schmittner et al., 2017).

**Table 3.** Statistical comparison of coretop $\delta^{13}C_{Cib}$ with predicted values produced by the CSIRO Mk3L-COAL ocean model.

| | Global (N=690) | | | |
|---|---|---|---|---|
| | Average | SD | RSME | $r^2$ |
| Observations | 0.52 ‰ | 0.50 ‰ | 0.00 ‰ | 1.0 |
| Raw comparison | 0.44 ‰ | 0.82 ‰ | 0.67 ‰ | 0.64 |
| Calibration | 0.49 ‰ | 0.76 ‰ | 0.63 ‰ | 0.64 |
| Calibration (vary-$\epsilon_{bio}^{13C}$) | 0.73 ‰ | 0.60 ‰ | 0.61 ‰ | 0.65 |

Some notable over and underestimation occurred in the adjusted $\delta^{13}C_{Cib}$ output that more or less mirrored those inconsistencies previously discussed for $\delta^{13}C_{DIC}$. Values as low as -1.9 ‰, well below measured $\delta^{13}C_{Cib}$ minima of -0.7 ‰, existed in the equatorial subsurface Pacific and Indian Oceans (i.e. where the oxygen minimum zones existed). This can be seen in figure 5, where some values in the equatorial band are well below the shaded region of good fit. Meanwhile, very high values of $\delta^{13}C_{Cib}$ were predicted in Arctic surface waters. The exaggeration of these local minima and maxima reflect those found in the modelled $\delta^{13}C_{DIC}$ distribution. Despite these local inconsistencies, CSIRO Mk3L-COAL shows good potential for direct comparisons to palaeoceanographic data sets of foraminiferal $\delta^{13}C$ with or without calibration.

## 4.3 $\delta^{15}N$ of nitrate ($\delta^{15}N_{NO_3}$)

We produced univariate measures of fit by comparing measurements of $\delta^{15}N_{NO_3}$ with equivalent values from CSIRO Mk3L-COAL at the nearest point (Fig. 6; Table 4). Measured $\delta^{15}N_{NO_3}$ were collected over a 30 year period using a variety of collection and measurement methods with a distinct bias towards the Atlantic Ocean. To try and remove some temporal and spatial bias, we binned and averaged measurements into equivalent model grids.

CSIRO Mk3L-COAL adequately reproduced the global patterns of $\delta^{15}N_{NO_3}$. We found excellent agreement in the volume-weighted means of $\delta^{15}N_{NO_3}$ (Table 4). Tight agreement in the means was a consequence of reproducing similar values where the majority of observed data existed. Most $\delta^{15}N_{NO_3}$ measurements have been taken from the upper 1,000 meters in the North Atlantic where values cluster at just under 5 ‰ (see lefthand panels in Fig. 7). Closer inspection of the Atlantic using depth and zonally averaged sections (Figs. 8 and 9) revealed that the model adequately reproduced the low $\delta^{15}N$ signature of $\sim$4 ‰

**Figure 5.** Measured versus modelled $\delta^{13}C_{Cib}$ (N = 690) of CSIRO Mk3L-COAL coloured by latitude. Red shading about the 1:1 line is an estimate of the variability implicit in the relationship between $\delta^{13}C_{Cib}$ and $\delta^{13}C_{DIC}$ of Schmittner et al. (2017). The inset at the bottom right shows the effect of the calibration of Eq. (21).

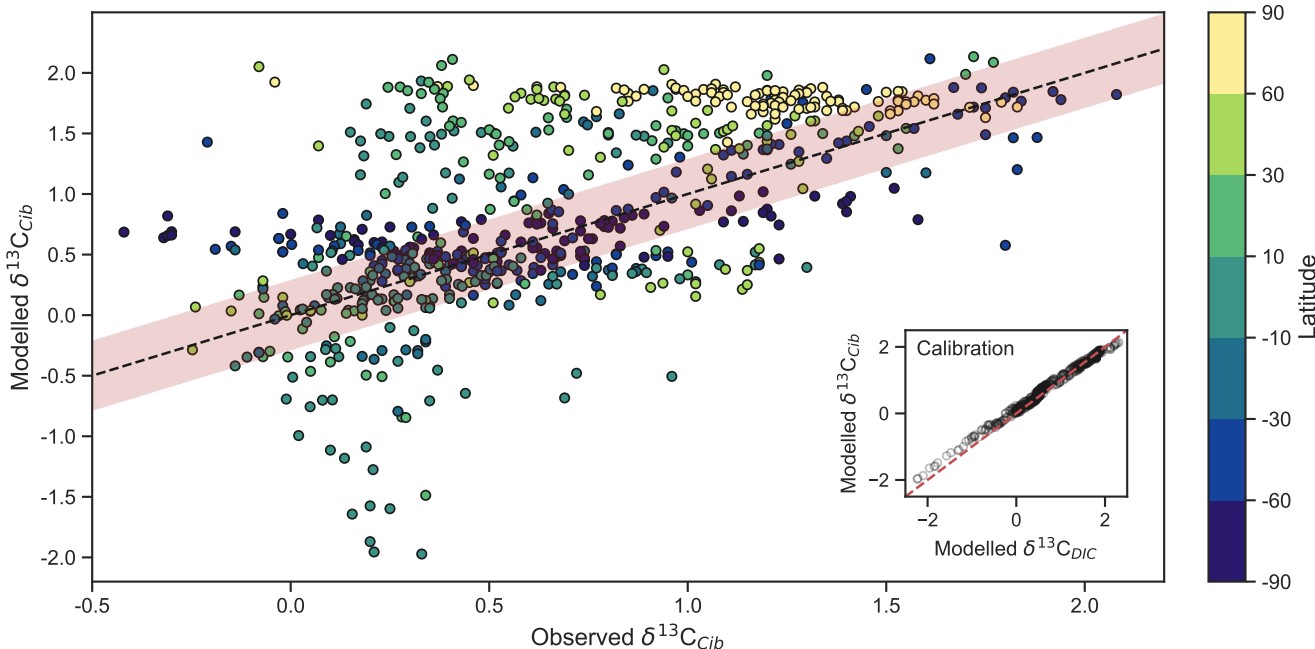

**Table 4.** Comparison of global and region mean $\delta^{15}N_{NO_3}$ between observations and model simulations. Model means are annual averages. All data was regridded onto the CSIRO Mk3L-COAL gridspace. The $\delta^{15}N$ data (5,330 measurements courtesy of The Sigman Lab, Princeton University) was binned into corresponding grid boxes and averaged for direct comparison, which reduced the data to 2,532 points. More than one data point of $\delta^{15}N$ may therefore contribute to each simulated value.

|  | Global | Southern Ocean | Atlantic | Pacific | Indian |
|---|---|---|---|---|---|
| **data** | **5.4 ‰** | **5.3 ‰** | **4.8 ‰** | **6.8 ‰** | **6.7 ‰** |
| CSIRO Mk3L-COAL | 5.5 ‰ | 5.4 ‰ | 4.7 ‰ | 7.8 ‰ | 5.2 ‰ |
| UVic-MOBI | 6.6 ‰ | 5.5 ‰ | 6.2 ‰ | 7.6 ‰ | 7.4 ‰ |
| PISCES | 4.3 ‰ | 4.6 ‰ | 3.7 ‰ | 5.6 ‰ | 5.1 ‰ |
| iCESM-high | 6.2 ‰ | 5.3 ‰ | 5.2 ‰ | 8.6 ‰ | 6.2 ‰ |

caused by $N_2$ fixation occurring in the tropical Atlantic (Marconi et al., 2017). A basin-wide rate of Atlantic $N_2$ fixation equal to $\sim$33 Tg N yr$^{-1}$ lowered Atlantic values below 5 ‰ and was fundamental for reproducing the observations. Outside the Atlantic

**Figure 6.** Global and regional fits between observations and simulated $\delta^{15}N_{NO_3}$ displayed as Taylor Diagrams (Taylor, 2001). Shading of the markers represent normalised bias. G = Global; S = Southern Ocean (90°S - 40°S); A = Atlantic (40°S - 70°N); P = Pacific (40°S - 70°N); I = Indian (40°S - 70°N). The $\delta^{15}N$ data (5,330 measurements courtesy of The Sigman Lab, Princeton University) was binned into corresponding grid boxes and averaged for direct comparison, which reduced the data to 2,532 points. More than one data point of $\delta^{15}N$ may therefore contribute to each simulated value.

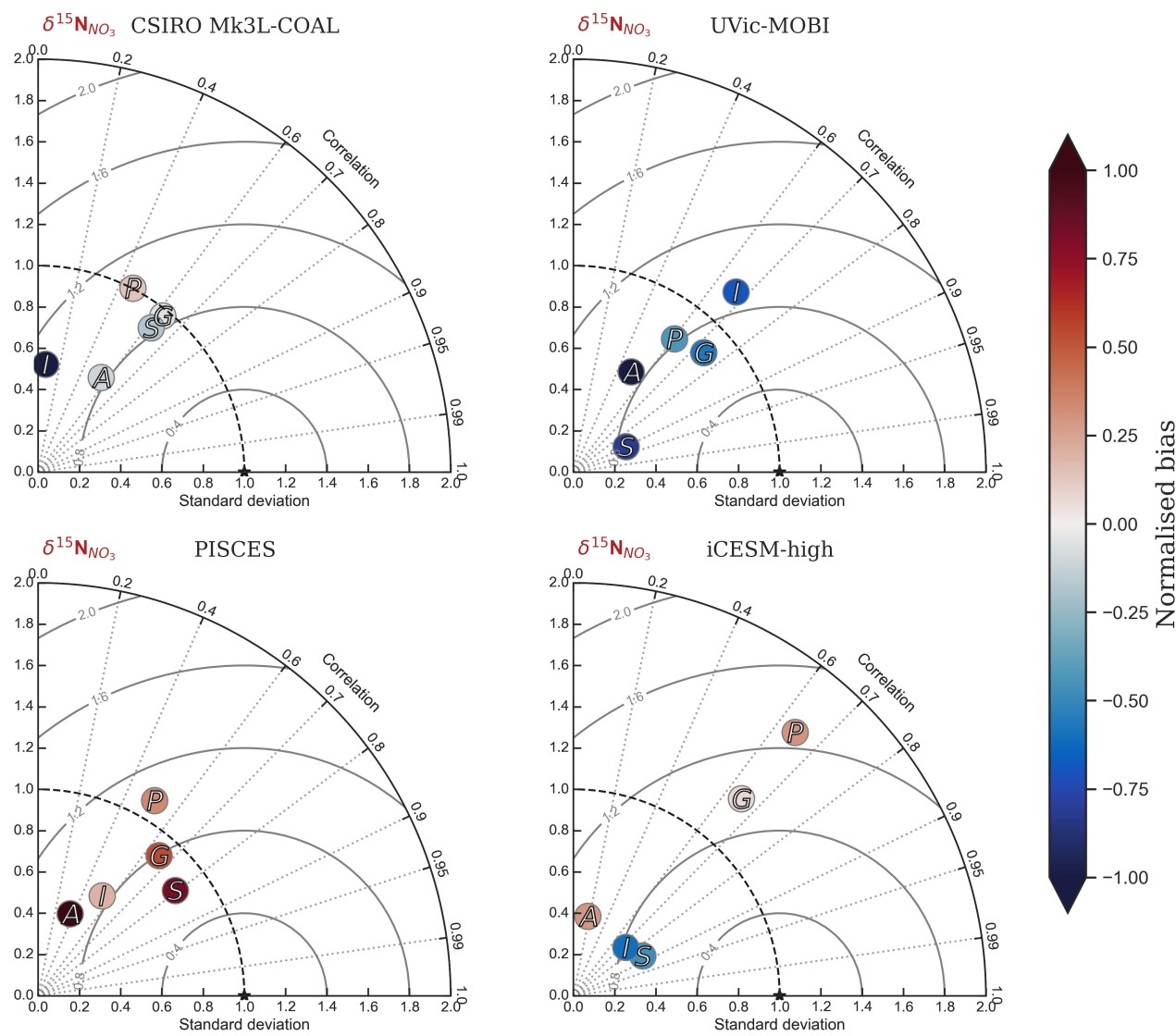

**Figure 7.** Observed (left) and modelled (right) $\delta^{15}$N of $NO_3$ data (N = 5,004) plotted against depth (a and b), latitude (c and d) and longitude (e and f). Colour shading represents the density of data, such that the darker a mass of data points is the more data is represented there.

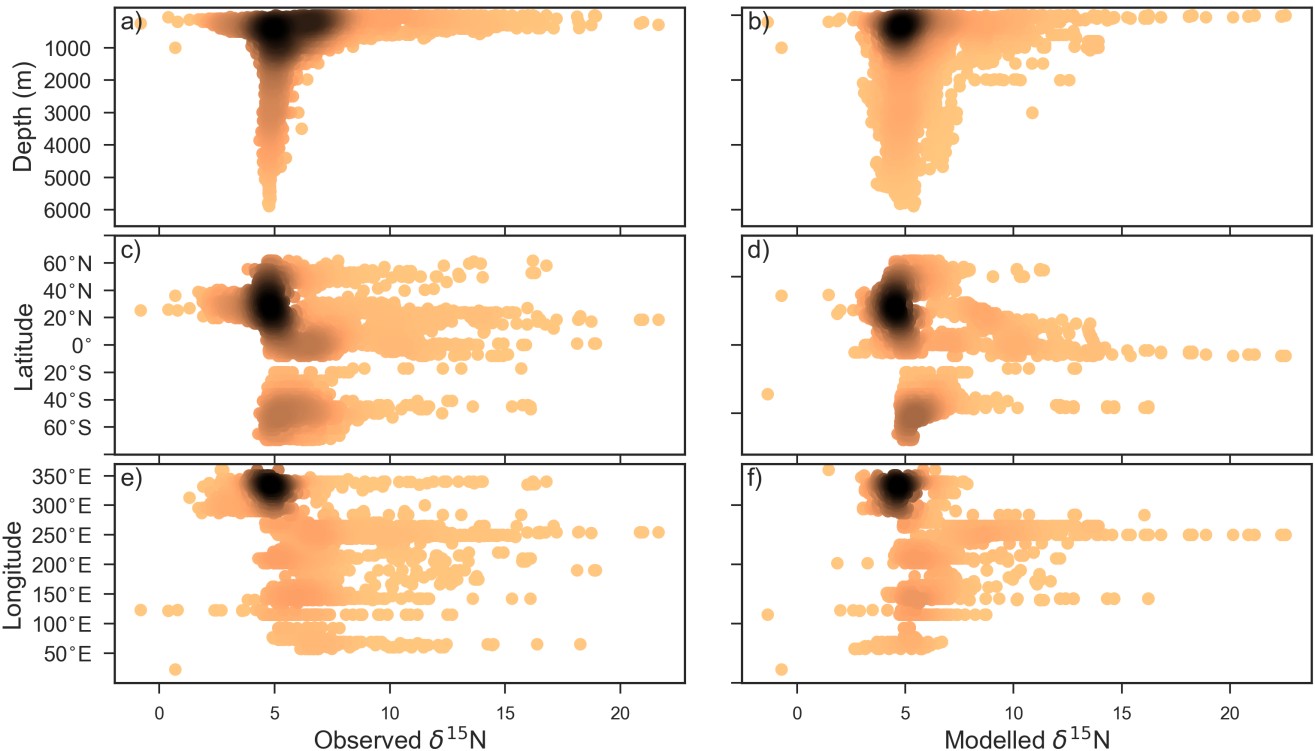

where data is more sparse, the model successfully reproduced the meridional gradients across the Antarctic, Subantarctic and Subtropical zones, the subsurface $\delta^{15}$N$_{NO_3}$ maxima in the tropics of all major basins, and the tongues of high and low values in surface waters of the Pacific consistent with changes in nitrate utilisation (Figs. 8 and 9).

Some important regional inconsistencies between the simulated and measured values did exist (refer to Figs. 8 and 9) and degraded the correlation. Much like the high values of $\delta^{13}$C$_{DIC}$ that were transported too deeply into the North Atlantic interior, a low $\delta^{15}$N$_{NO_3}$ signature was transported too far into the deep North Atlantic. CSIRO Mk3L-COAL therefore underestimated deep $\delta^{15}$N$_{NO_3}$ before mixing through to the South Atlantic restored values towards the measurements. Subsurface values in the North Pacific were also underestimated, which can be attributed to the inability of the coarse resolution OGCM to transport low $O_2$, high $\delta^{15}$N$_{NO_3}$ water northwards from the Eastern Tropical Pacific. Simulated values in the Indian Ocean, specifically near to the Arabian Sea, significantly underestimated the data because the suboxic zone was misrepresented in the Bay of Bengal. Misrepresentation of the North Indian seas was responsible for very poor model-data fit in the Indian Ocean (Fig. 6). Meanwhile, the deep (> 1,500 metres) Eastern Tropical Pacific tended to overestimate the data, owing to a large,

**Figure 8.** Depth averaged sections of modelled (colour contours) and observed (overlaid markers) $\delta^{15}N_{NO_3}$.

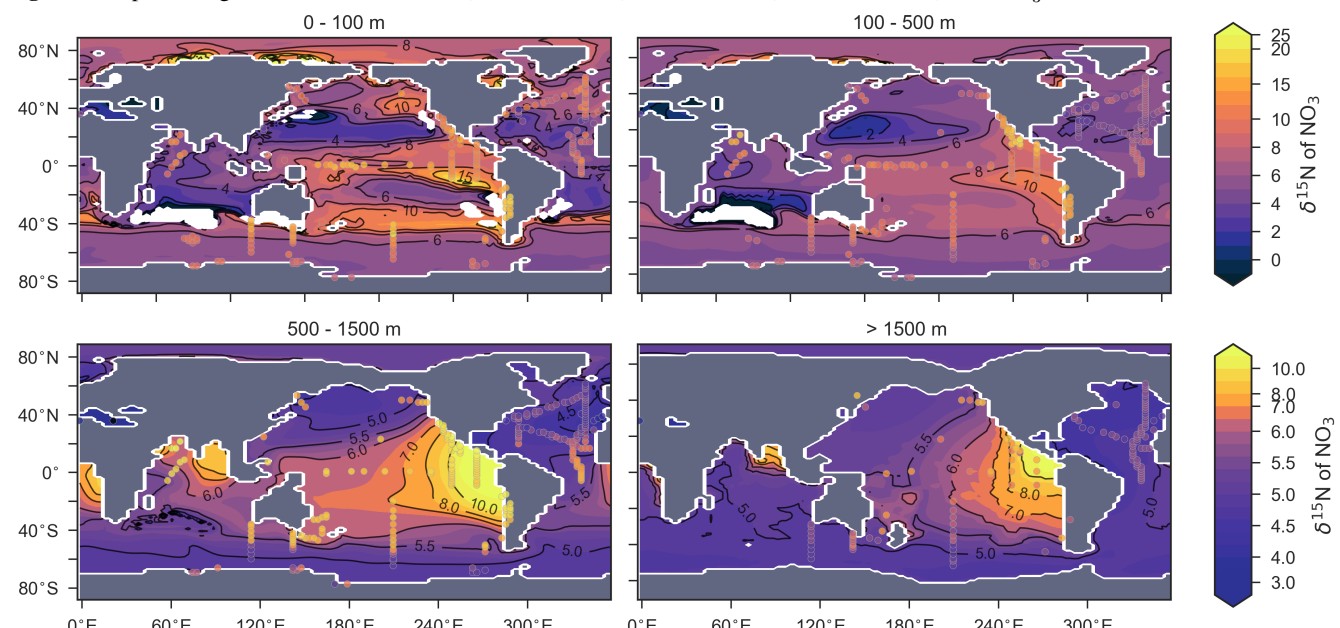

deep, unimodal suboxic zone. These physically-driven inconsistencies in the oxygen field are common to other coarse resolution models (Oschlies et al., 2008; Schmittner et al., 2008), and like the $\delta^{13}C$ distribution, were the main cause of the misfit between simulated and observed $\delta^{15}N_{NO_3}$. The correlations reflected these regional under and overestimations, particularly in the Indian Ocean.

Finally, we placed CSIRO Mk3L-COAL in the context of other isotope-enabled global models: UVic-MOBI, PISCES and iCESM-high (Table 1). This comparison demonstrated that the modelled distribution of $\delta^{15}N_{NO_3}$ was adequately placed among the current generation of models. The global and regional means were more accurately reproduced by CSIRO Mk3L-COAL than for UVic-MOBI, PISCES and iCESM-high (Table 4; also see shading in Figure 6). Atlantic $\delta^{15}N_{NO_3}$ was best reproduced by CSIRO Mk3L-COAL. Meanwhile, correlations tended to be slightly lower for CSIRO Mk3L-COAL than UVic-MOBI and iCESM-high, and consistently lower than PISCES (Figure 6). UVic-MOBI underestimated the data, but produced high

correlations in the Southern Ocean and globally. Regionally, PISCES was best correlated to the measurements of $\delta^{15}N_{NO_3}$ of the three models, although it had a consistent positive bias. iCESM-high was acceptably correlated to the data in the global sense, but was highest in RMS errors, particularly in the Pacific. CSIRO Mk3L-COAL therefore showed an acceptable measure of fit to the noisy and sparse $\delta^{15}N_{NO_3}$ data and reproduced most regional patterns, albeit with misrepresentation in the Indian

Ocean and some exaggerations of local minima/maxima as discussed. Future model-data comparisons with CSIRO Mk3L-COAL should therefore take these limitations into account. Overall, however, we find that CSIRO Mk3L-COAL broadly

**Figure 9.** Zonally averaged sections of modelled (colour contours) and observed (overlaid markers) $\delta^{15}N_{NO_3}$. The global zonal average encompasses all basins.

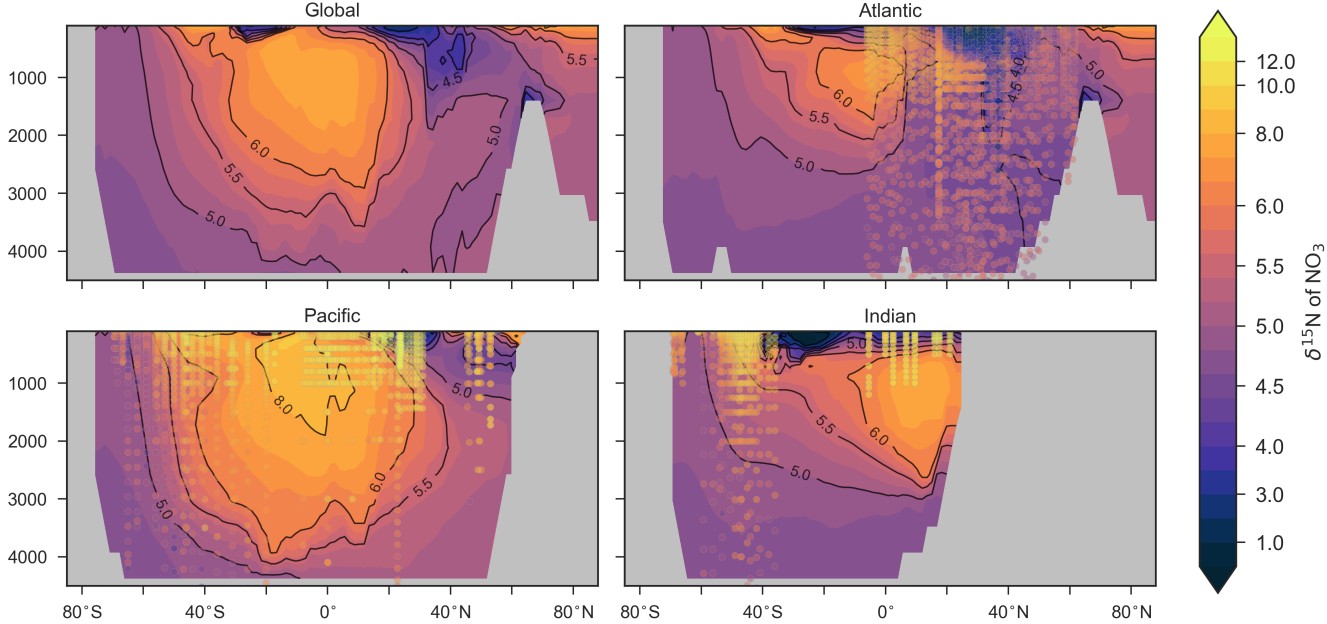

reproduced the $\delta^{15}N_{NO_3}$ data. Annual rates of $N_2$ fixation, water column denitrification and sedimentary denitrification at roughly 122, 52 and 78 Tg N yr$^{-1}$, respectively, produced this agreement.

An important caveat to the $\delta^{15}N_{NO_3}$ routines of CSIRO Mk3L-COAL should be noted. CSIRO Mk3L-COAL underwent significant tuning of water column and sedimentary denitrification parameterisations in order to reproduce known values of
5    $\delta^{15}N_{NO_3}$ during development. One important parameter is the lower threshold of $NO_3$ concentration at which point water column denitrification is shut off (section A2.3). In CSIRO Mk3L-COAL this is set at 30 mmol m$^{-3}$, which is an arbitrary limit that was implemented to prevent water column denitrification from reducing $NO_3$ to zero in the large suboxic zones. Hence, a caveat of the current model is an inability for water column and sedimentary denitrification to realistically adjust as suboxia changes. However, the parameterisation does allow for targeted experiments where the ratio of water column to
10    sedimentary denitrification can be controlled if, for instance, it is unclear how water column and sedimentary denitrification respond to certain conditions. This is currently the case during the Last Glacial Maximum, where expansive suboxic zones in the Pacific (Hoogakker et al., 2018) were counter-intuitively associated with reduced rates of water column denitrification (Ganeshram et al., 1995). We have, in this version, chosen to keep this parameterisation and note that future developments will focus on dynamic responses to variations in suboxia.

## 4.4 $\delta^{15}$N of organic matter ($\delta^{15}$N$_{org}$)

CSIRO Mk3L-COAL tracks the $\delta^{15}$N signature of organic matter ($\delta^{15}$N$_{org}$) that is deposited in the sediments. We compared the simulated $\delta^{15}$N$_{org}$ to the coretop compilation of Tesdal et al. (2013) with 2,176 records of $\delta^{15}$N$_{org}$. These records were binned and averaged onto the CSIRO Mk3L-COAL ocean grid, such that the 2,176 records became 592. When comparing sediment coretop measurements of $\delta^{15}$N to that of the model, it is necessary to consider how $\delta^{15}$N$_{org}$ is altered by early burial. As records in the compilation of Tesdal et al. (2013) are from bulk nitrogen, we can assume that the "diagenetic offset" as described by Robinson et al. (2012) is active. The diagenetic offset involves an increase in the $\delta^{15}$N of sedimentary nitrogen of between 0.5 and 4.1 ‰ relative to that of sinking particulate organic matter and appears to be related to pressure (Robinson et al., 2012), although the reasoning behind this relationship remains to be defined.

In light of the diagenetic offset, we make three comparisons with the compilation of Tesdal et al. (2013). A raw comparison is made, alongside an attempt to account for the diagenetic offset using two depth-dependent corrections (Table 5 and Fig. 10):

$$\delta^{15}N_{org}^{cor:1} = \begin{cases} \delta^{15}N_{org}, & \text{if } z(km) < 1km \\ \delta^{15}N_{org} + \left(1 \cdot z(km) + 1\right), & \text{if } z(km) \geq 1km \end{cases} \tag{22}$$

$$\delta^{15}N_{org}^{cor:2} = \delta^{15}N_{org} + 0.9 \cdot z(km) \tag{23}$$

The first correction ($\delta^{15}$N$_{org}^{cor:1}$) is taken from Robinson et al. (2012), while the second ($\delta^{15}$N$_{org}^{cor:2}$) originates from how Schmittner and Somes (2016) treated sedimentary nitrogen isotope data in their study of the Last Glacial Maximum. Both are based on the observation that the diagenetic offset increases with pressure, in this case represented by depth ($z$) in kilometres (km).

Following binning and averaging onto the model grid, the raw comparison immediately showed a consistent underestimation of the coretop data, with a predicted mean of 2.7 ‰ well below the observed mean of 4.7 ‰. Our correlation was 0.27, which indicates a limited ability to replicate regional patterns. This underestimation and low correlation is easily seen when predicted values are compared directly to the coretop data in Fig. 10. Like the nitrogen isotope model of Somes et al. (2010), we find that the offset between simulated and observed coretop bulk $\delta^{15}$N$_{org}$ is roughly equivalent to the observed average diagenetic offset of $\sim$2.3 $\pm$ 1.8 ‰. This indicates that diagenetic alteration of $\delta^{15}$N$_{org}$ is active during early burial in the coretop data.

Including a diagenetic offset therefore improved agreement between our predicted $\delta^{15}$N$_{org}$ and the coretop data considerably (Table 5 and Fig. 10). Both corrections accounted for the enrichment of $\delta^{15}$N in deeper regions and the minor diagenetic alteration in areas of high sedimentation that typically occur in shallower sediments. The average $\delta^{15}$N$_{org}$ increased to 4.5 ‰ for $\delta^{15}$N$_{org}^{cor:1}$ and 5.2 ‰ for $\delta^{15}$N$_{org}^{cor:2}$. Correlations increased from 0.27 to 0.47 and 0.53, respectively. The improvement was clearly observed in the Southern Ocean, where both the magnitude and spatial pattern of $\delta^{15}$N$_{org}$ were well replicated by the model. Changes in the Southern Ocean over glacial-interglacial cycles reflect shifts in the global marine nitrogen cycle and nutrient utilisation (Martinez-Garcia et al., 2014; Studer et al., 2018), and the ability of CSIRO Mk3L-COAL to account for these patterns in the coretop data is encouraging for future study. We suggest that future palaeoceanographic model-data comparisons of $\delta^{15}$N$_{org}$ use the depth-correction of Schmittner and Somes (2016) as it provided the best correlations and reproduced Southern Ocean $\delta^{15}$N$_{org}$ at 0.5 ‰ greater than the global mean (see Table 5).

**Figure 10.** Direct comparison of observed versus modelled $\delta^{15}\text{N}_{org}$ incident on the sediments. Left-side panels show spatial distribution of simulated $\delta^{15}\text{N}_{org}$ overlain by coretop data from the compilation of Tesdal et al. (2013). Right-side panels compare all coretop data against simulated $\delta^{15}\text{N}_{org}$. Top panels depicts raw output of the model, while the middle and bottom panels depict the predicted values of the model following two depth-dependent offsets (Eqs. (22) and (23)) that account for diagenetic alteration.

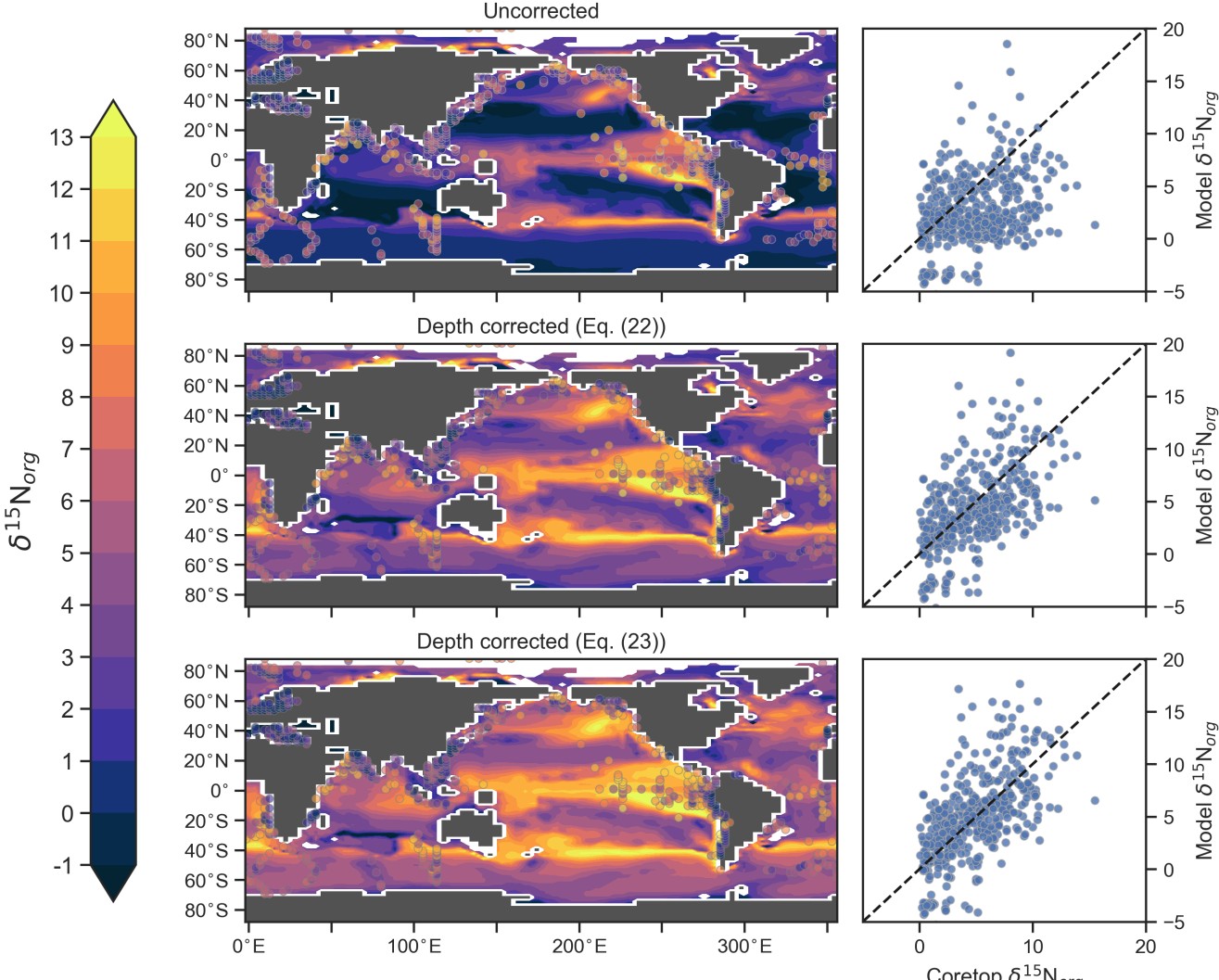

**Table 5.** Statistical comparison of coretop $\delta^{15}N_{org}$ with predicted values of the CSIRO Mk3L-COAL ocean model. The corrected vales ($\delta^{15}N_{org}^{cor:1}$ and $\delta^{15}N_{org}^{cor:2}$) account for alteration during early diageneis following burial.

| | Global (N=592) | | | Southern Ocean (N=81) | | |
|---|---|---|---|---|---|---|
| | Average | SD | $r^2$ | Average | SD | $r^2$ |
| Observations | 4.7 ‰ | 3.1 ‰ | 1.0 | 5.2 ‰ | 1.7 ‰ | 1.0 |
| Raw comparison | 2.7 ‰ | 3.2 ‰ | 0.27 | 1.1 ‰ | 1.6 ‰ | 0.13 |
| $\delta^{15}N_{org}^{cor:1}$ | 4.5 ‰ | 3.8 ‰ | 0.47 | 4.3 ‰ | 1.8 ‰ | 0.45 |
| $\delta^{15}N_{org}^{cor:2}$ | 5.2 ‰ | 4.2 ‰ | 0.53 | 5.7 ‰ | 1.9 ‰ | 0.47 |

## 5   Ecosystem effects

As a first test of the isotope-enabled ocean model, we undertook simple ecosystem experiments to assess the effect on $\delta^{13}C$ and $\delta^{15}N$. For reference, the assessment of model performance described above used model output with variable stoichiometry activated, a fixed 8% rain ratio of $CaCO_3$ to organic carbon, and a strong iron limitation of $N_2$ fixers that enforced a low degree of spatial coupling between $N_2$ fixers and denitrification zones. A summary of the biogeochemical effects of the different experiments is provided in Table 6.

### 5.1   Variable versus Redfieldian stoichiometry

Enabling variable stoichiometry (see appendix A3) of the general phytoplankton group ($P_{org}^G$) over a Redfieldian ratio (C:N:P:$O_2^{rem}$:$NO_3^{rem}$ = 106:16:1:-138:-94.4) altered the rate and distribution of organic matter export. Organic matter had more carbon and nitrogen per unit phosphorus in regions with low $PO_4$, such as the Atlantic Ocean (Fig. 11a), which elevated $O_2$ and $NO_3$ demand during oxic and suboxic remineralisation (denitrification), respectively. Lower ratios were produced in eutrophic regions such as the subarctic Pacific, Southern Ocean and tropical zones of upwelling. Overall, global mean C:P increased from the Redfieldian 106:1 to 117:1 and caused an increase in carbon export from 7.6 to 8.0 Pg C yr$^{-1}$. Approximately 0.1 Pg C yr$^{-1}$, or 25 % of the increase, was attributed purely to organic carbon export from $N_2$ fixation, which increased from 107 to 122 Tg N yr$^{-1}$ as higher N:P ratios in the tropics broadened their competitive niche. The total contribution of $N_2$ fixation to the increase in carbon export was likely greater than 25 %, as $NO_3$ also became more available to $NO_3$-limited ecosystems in the lower latitudes (Moore et al., 2013). The increase in carbon export under variable stoichiometry as compared to a Redfieldian ocean was therefore felt largely in the lower latitudes between 40°S and 40°N (Fig. 11b). Export production decreased poleward of 40°, particularly in the Southern Ocean, because C:P ratios were lower than the 106:1 Redfield ratio (Fig. 11a).

Distributions of both isotopes were affected by the change in carbon export and the marine nitrogen cycle. Global mean $\delta^{13}C_{DIC}$ increased from 0.52 to 0.54 ‰, and $\delta^{15}N_{NO_3}$ increased from 5.1 to 5.6 ‰. These are not great changes on the global scale and they had little influence on model-data measures of fit. However, the spatial distribution of these isotopes was significantly altered. Intermediate waters leaving the Southern Ocean were depleted in $\delta^{13}C_{DIC}$ by up to 0.1 ‰ and $\delta^{15}N_{NO_3}$

**Figure 11.** Simulated difference in the C:P ratio of exported organic matter due to variable stoichiometry as compared to Redfield stoichiometry (top) and the resulting change in carbon export out of the euphotic zone (bottom).

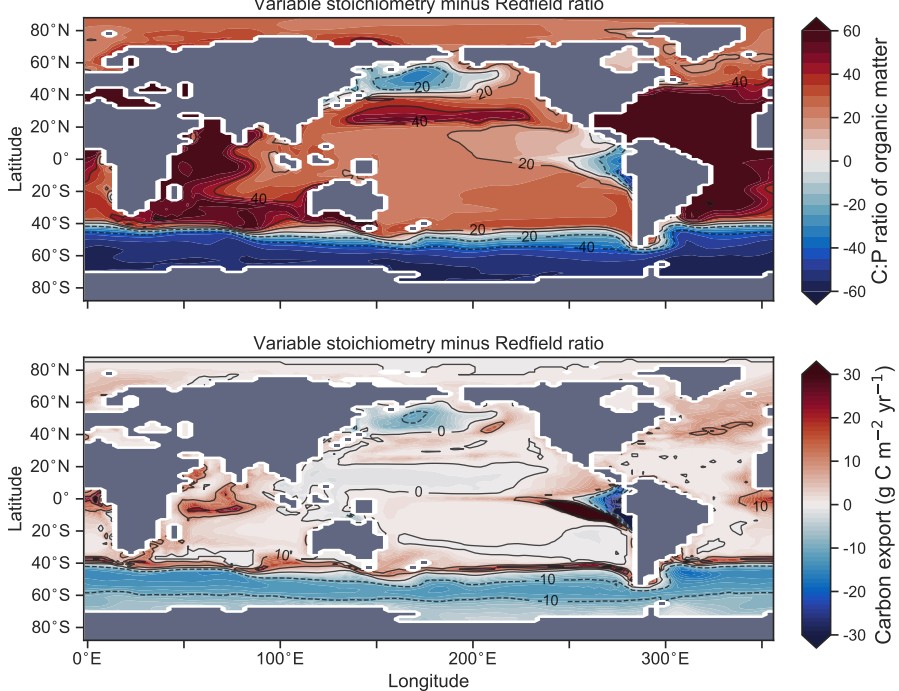

by up to 1 ‰, while the deep ocean, particularly the Pacific, was enriched in both isotope to a similar degree (Fig. 12). Depletion of both isotope in waters subducted between 40°S and 60°S reflected the local loss in export production as a result of lower C:P and N:P ratios, such that biological fractionation was unable to enrich DIC and $NO_3$ in the heavier isotope to the same degree as surface waters travelled north. Enrichment of $\delta^{13}$C in the deep ocean was the result of reduced carbon export in the Antarctic

5   zone due to low C:P ratios, while enrichment of $\delta^{15}$N in the deep ocean was the result of increased tropical production that increased water column denitrification ($\epsilon^{15N}_{wc} = 20$ ‰). Lower C:P and N:P ratios in both the Antarctic and Subantarctic zones therefore elicited divergent isotope effects in deep and intermediate waters leaving the Southern Ocean.

Meanwhile, each isotope showed a different response in the suboxic zones of the tropics where variable stoichiometry increased the volume of suboxia ($O_2 < 10$ mmol m$^{-3}$) by 0.5 %. The increase in water column denitrification caused by the

10   expansion of suboxia increased $\delta^{15}$N$_{NO_3}$, while the local increase in carbon export that drove the increase in water column denitrification reduced $\delta^{13}$C$_{DIC}$ in the same waters (Fig. 12). Overall, the increase in low latitude carbon export caused an expansion of water column suboxia and elicited diverging behaviours in the isotopes, whereby $\delta^{15}$N$_{NO_3}$ increased and $\delta^{13}$C$_{DIC}$ decreased.

**Figure 12.** Differences in $\delta^{13}C_{DIC}$ (top) and $\delta^{15}N_{NO_3}$ (bottom) as a result of variable stoichiometry as compared to Redfield stoichiometry. Values are zonal means.

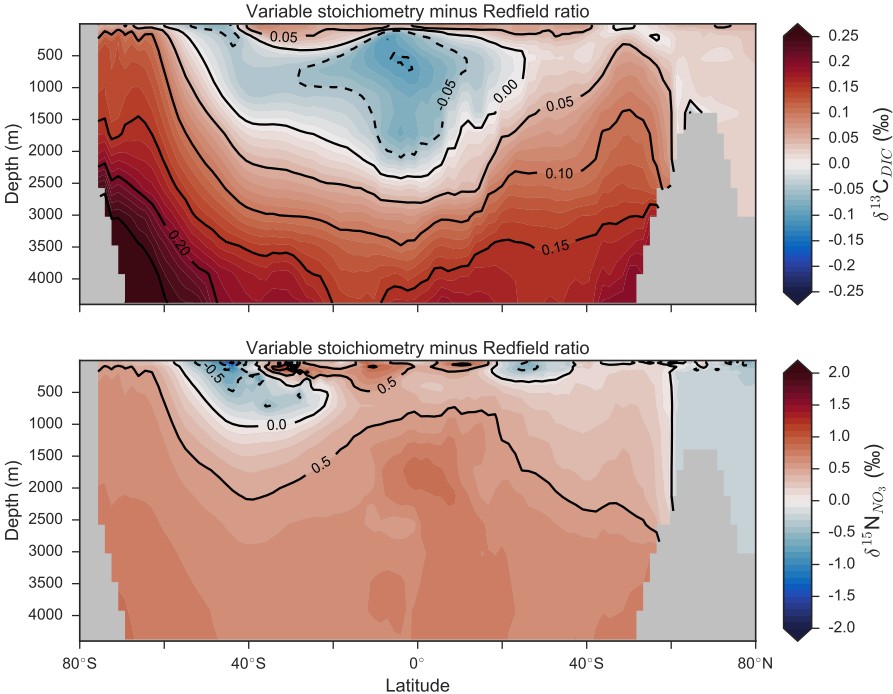

## 5.2 Calcifier dependence on calcite saturation state

The rate of calcification of planktonic foraminifera and coccolithophores is dependent on the calcite saturation state (Zondervan et al., 2001). In previous experiments, the production of $CaCO_3$ was fixed at a rate of 8 % per unit of organic carbon produced in accordance with the modelling study of Yamanaka and Tajika (1996) and produced 0.54 Pg $CaCO_3$ yr$^{-1}$. Now we investigate
5  how spatial variations in the $CaCO_3$:$C_{org}$ ratio ($R_{CaCO_3}$ in Eq. (A17)) affected $\delta^{13}C_{DIC}$ and $\delta^{13}C_{Cib}$ (see appendix A1.3). We applied three different values of $\eta$ to Eq. (A18) to alter the quantity of $CaCO_3$ produced per unit of organic carbon ($C_{org}^{G}$) given the calcite saturation state ($\Omega_{ca}$). The $\eta$ coefficients were 0.53, 0.81 and 1.09. These numbers are equivalent to those in the experiments of Zhang and Cao (2016).

    Mean $R_{CaCO_3}$ was 4.5, 6.6 and 9.5 % and annual $CaCO_3$ production was 0.32, 0.47 and 0.68 Pg $CaCO_3$ yr$^{-1}$ in the
10  three experiments. Although different in total $CaCO_3$ production, the three experiments shared the same spatial patterns. Low latitude waters were high in $R_{CaCO_3}$, particularly the oligotrophic subtropical gyres, while high latitudes were low, particularly the Antarctic zone where mixing of old waters into the surface depressed the calcite saturation state (Fig. 13). These regional patterns in $R_{CaCO_3}$ therefore had the largest effect in areas of high export production. Productive, high latitude areas like the Southern Ocean, subpolar Pacific and North Atlantic waters all produced less $CaCO_3$ when compared to an enforced 8 % rain

**Table 6.** Summary of the biogeochemical effects of the different treatments of the ecosystem in CSIRO Mk3L-COAL. $C_{org}$ is the total organic carbon exported from the euphotic zone composed of both general and diazotrophic phytoplankton groups ($C_{org}^G + C_{org}^D$; see appendix A1), while $C_{CaCO_3}$ is the total export of $CaCO_3$ out of the euphotic zone. The sum of $C_{org}$ and $C_{CaCO_3}$ equal the global rate of carbon export referred to in the text. Sed:WC refers to the sedimentary to water column denitrification ratio. Note that the global mean $\delta^{13}C_{DIC}$ is higher than reported in Table 2 because it includes the upper 200 metres and the Arctic.

| | $C_{org}$ | $C_{CaCO_3}$ | $N_2$ fix | Sed:WC | $O_2$ | Suboxia | DIC | $\delta^{13}C_{DIC}$ | $\delta^{15}N_{NO_3}$ |
|---|---|---|---|---|---|---|---|---|---|
| | Pg C yr$^{-1}$ | | Tg N yr$^{-1}$ | ratio | mmol m$^{-3}$ | % ocean | Pg C | ‰ | |
| **Variable versus Redfieldian stoichiometry (section 5.1)** | | | | | | | | | |
| Redfield | 7.08 | 0.52 | 107 | 1.5 | 187 | 1.5 | 33908 | 0.47 | 5.1 |
| Variable | 7.42 | 0.54 | 122 | 1.5 | 193 | 2.1 | 33870 | 0.51 | 5.6 |
| **Calcifier dependence on calcite saturation state (section 5.2)** | | | | | | | | | |
| Fixed (8% of $C_{org}^G$) | 7.42 | 0.54 | 122 | 1.5 | 193 | 2.1 | 33870 | 0.51 | 5.6 |
| Variable ($\eta = 0.53$) | 7.41 | 0.32 | 122 | 1.5 | 193 | 2.1 | 34010 | 0.52 | 5.6 |
| Variable ($\eta = 0.81$) | 7.41 | 0.47 | 122 | 1.5 | 193 | 2.1 | 33916 | 0.50 | 5.6 |
| Variable ($\eta = 1.09$) | 7.42 | 0.68 | 122 | 1.5 | 193 | 2.1 | 33783 | 0.48 | 5.6 |
| **Strength of coupling between $N_2$ fixation and denitrification(section 5.3)** | | | | | | | | | |
| Weak | 7.42 | 0.54 | 122 | 1.5 | 193 | 2.1 | 33870 | 0.51 | 5.6 |
| Moderate | 7.72 | 0.48 | 144 | 1.9 | 188 | 2.5 | 34079 | 0.45 | 5.2 |
| Strong | 7. | 0.46 | 154 | 2.1 | 187 | 2.7 | 34182 | 0.42 | 5.0 |

ratio, while $CaCO_3$ production between 40°S and 40°N relative to a fixed $R_{CaCO_3}$ of 8 % was dependent on $\eta$. The highest $\eta$ coefficient of 1.09 achieved greater export of $CaCO_3$ in the mid to lower latitude regions of high export production (Fig. 13). The consequence of increasing $CaCO_3$ production in the mid-lower latitudes was a loss of upper ocean alkalinity, subsequent outgassing of $CO_2$ and losses in the DIC inventory. Losses in global DIC were 95 and 130 Pg C as $R_{CaCO_3}$ increased from

5  $4.6 \to 6.6 \to 9.5$ % (Table 6), equivalent to $\frac{1}{5}^{th}$ of the glacial increase in oceanic carbon (Ciais et al., 2011).

Despite the significant changes associated with the implementation of $\Omega_{ca}$-dependent $CaCO_3$ production, effects were negligible on both $\delta^{13}C_{DIC}$ and $\delta^{13}C_{Cib}$. Global mean $\delta^{13}C_{DIC}$ was 0.51 ‰, when $R_{CaCO_3}$ was fixed at 8 %, and this changed to 0.52, 0.50 and 0.48 ‰ under $\eta$ coefficients of 0.53, 0.81 and 1.09 (Table 6). Likewise, global mean $\delta^{13}C_{Cib}$ was 0.59 ‰, when $R_{CaCO_3}$ was fixed at 8 %, and this changed to 0.60, 0.58 and 0.55 ‰. Minimal change in $\delta^{13}C_{Cib}$ indicated minimal change in

10  the $CO_3^{2-}$ concentration (see Eq. (21)), which varied by $\leq 2$ mmol m$^{-3}$ between experiments. Visual inspection of the change in $\delta^{13}C_{DIC}$ and $\delta^{13}C_{Cib}$ distributions showed an enrichment of these isotopes in the upper ocean north of 40°S. Subsequent increases in $\eta$, which increased low latitude $CaCO_3$ production, magnified the enrichment. Enrichment of $\delta^{13}C_{DIC}$ and $\delta^{13}C_{Cib}$ was caused by outgassing of $CO_2$ as surface alkalinity decreased in response to greater $CaCO_3$ production (Fig. 14). The change, however, was at most 0.1 ‰, which lies well within one standard deviation of variability known in the proxy data

**Figure 13.** Global distribution of CaCO$_3$ export as a percentage of organic carbon (C$_{org}$) export (top), and the change in the CaCO$_3$ production field as a result of making CaCO$_3$ production dependent on calcite saturation state ($\eta = 1.09$) compared to when it was a fixed 8 % of C$_{org}$ (bottom). Areas where export production does not occur due to severely nutrient limited conditions are masked out.

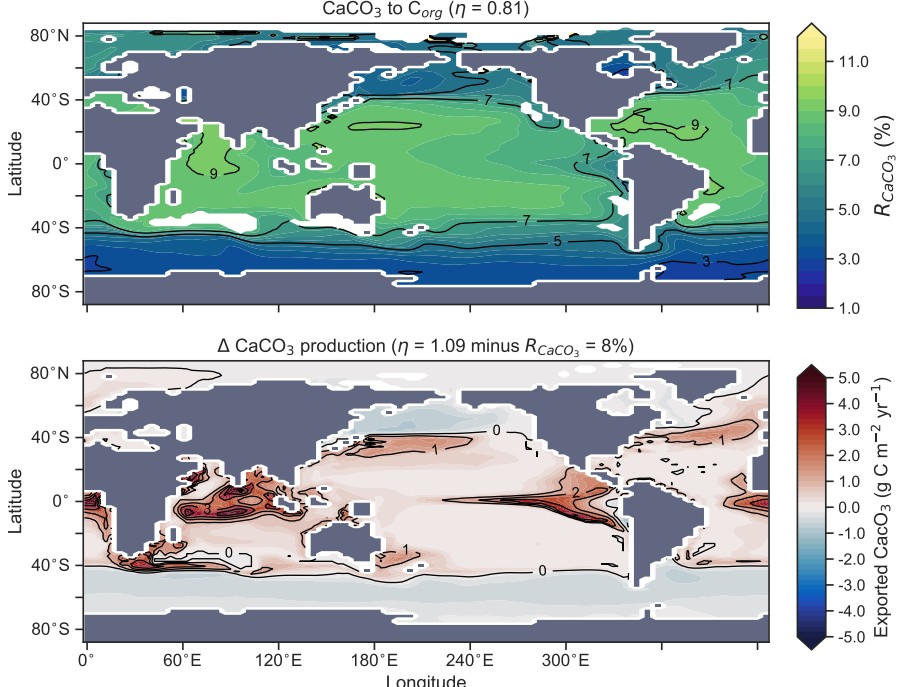

(Schmittner et al., 2017). We therefore find little scope for recognising even large variations in global CaCO$_3$ production (0.32 to 0.68 Pg CaCO$_3$ yr$^{-1}$) in the signature of carbon isotopes despite considerable effects on the oceanic inventory of DIC.

However, we stress that version 1.0 of CSIRO Mk3L-COAL does not include CaCO$_3$ burial or dissolution from the sediments according the calcite saturation state of overlying water (Boudreau, 2013). To neglect ocean-sediment CaCO$_3$ cycling is to
5   neglect of an important aspect of the global carbon cycle active on millennial timescales (Sigman et al., 2010). Changes in CaCO$_3$ burial and dissolution could have a non-negligible effect on $\delta^{13}$C through altering whole ocean alkalinity and thereby air-sea gas exchange of CO$_2$, which would in turn affect surface $\delta^{13}$C as we have seen. While we do not address these effects here, we aim to do so in upcoming versions of the model equipped with carbon compensation dynamics.

### 5.3   Strength of coupling between N$_2$ fixation and denitrification

10   The degree to which N$_2$ fixers are spatially coupled to the tropical denitrification zones is controlled by altering the degree to which N$_2$ fixers are limited by iron ($K_{Fe}^D$) in Eq. (A12) (see appendix A1.2). Decreasing $K_{Fe}^D$ ensures that N$_2$ fixation becomes less dependent on iron supply, and as such is released from regions of high aeolian deposition, such as the North Atlantic, to inhabit areas of low NO$_3$:PO$_4$ ratios. Areas of low NO$_3$:PO$_4$ exist in the tropics proximal to water column denitrification

**Figure 14.** Changes in the distribution of carbon isotopes ($\delta^{13}C_{DIC}$ and $\delta^{13}C_{Cib}$; top) and carbon chemistry (dissolved inorganic carbon and alkalinity; bottom) as a result of increasing $CaCO_3$ production in surface waters between 40°S and 40°N.

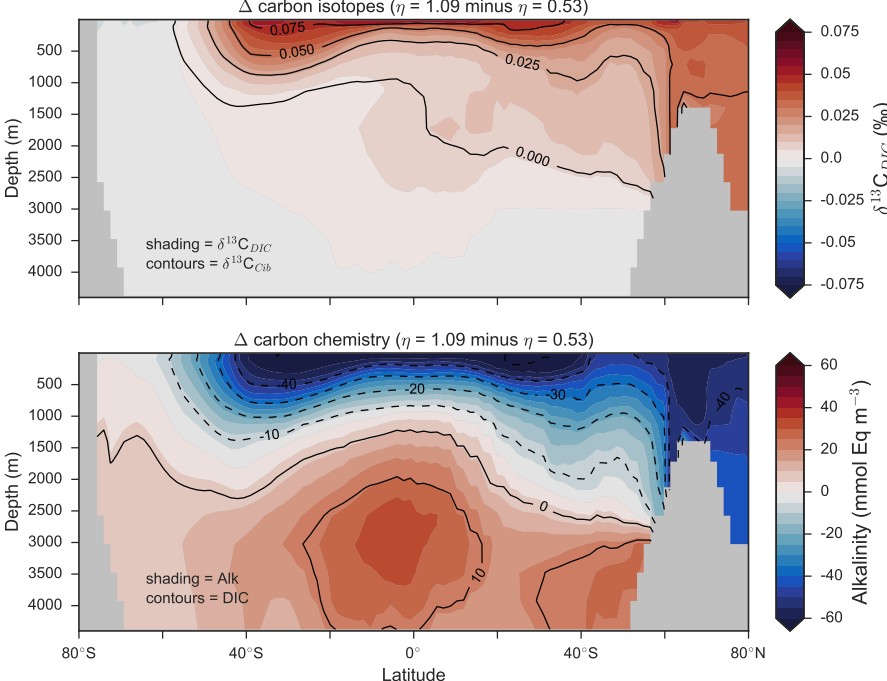

zones. Releasing $N_2$ fixers from Fe limitation therefore increases the spatial coupling between $N_2$ fixation and water column denitrification and increases the global rate of $N_2$ fixation.

We steadily decreased iron limitation ($K_{Fe}^D$) to increase the strength of spatial coupling between $N_2$ fixers and the tropical denitrification zones (Fig. 15). As $N_2$ fixers coupled more strongly to regions of low $NO_3:PO_4$, the rate of $N_2$ fixation increased from 122 to 144 to 154 Tg N yr$^{-1}$ (Table 6). An expansion of suboxia from 2.1 to 2.5 to 2.7 % of global ocean volume in the tropics accompanied the increase in $N_2$ fixation, as did a decrease in global mean $\delta^{13}C_{DIC}$ of 0.06 and 0.1 ‰, since greater rates of $N_2$ fixation stimulated tropical export production. Due to the expansion of the already large suboxic zones, which occurred in both horizontal and vertical directions, the amount of organic carbon that reached tropical sediments (20°S to 20°N) increased from 0.35 to 0.46 to 0.51 Pg C yr$^{-1}$.

The overarching consequence for $\delta^{15}N_{NO_3}$ due to an expansion of the suboxic zones was an increase in the sedimentary to water column denitrification ratio from 1.5 to 1.9 to 2.2, which decreased mean $\delta^{15}N_{NO_3}$ from 5.6 to 5.2 to 5.0 ‰ (Table 6). The increase in $N_2$ fixation ($\delta^{15}N_{org}$ = -1 ‰) and sedimentary denitrification ($\epsilon_{sed}^{15N}$ = 3 ‰) in the tropics was felt globally for $\delta^{15}N_{NO_3}$ (Fig. 16). Lower $\delta^{15}N_{NO_3}$ by 0.5 and 0.9 ‰ permeated water columns in the Southern Ocean and tropics, respectively. Meanwhile, $\delta^{15}N_{NO_3}$ was up to 10 ‰ lower in surface waters of the tropical and subtropical Pacific, which is

where the greatest increase in $N_2$ fixation and sedimentary denitrification occurred. The dramatic reduction in surface $\delta^{15}N_{NO_3}$ was subsequently conveyed to the sediments as $\delta^{15}N_{org} \pm 1\text{-}2 \text{ ‰}$.

These simple experiments demonstrate that the insights garnered from sedimentary records of $\delta^{15}N$ are open to multiple lines of interpretation. An expansion of the suboxic zones, normally associated with an increase in $\delta^{15}N_{NO_3}$ (Galbraith et al., 2013), could instead cause a decrease in $\delta^{15}N_{NO_3}$ if more organic matter reached the sediments to stimulate sedimentary denitrification. There is good evidence that the suboxic zones might have undergone a vertical expansion (Hoogakker et al., 2018) and that more organic matter reached the tropical sediments under glacial conditions (Cartapanis et al., 2016). The glacial decrease in bulk $\delta^{15}N_{org}$ recorded in the eastern tropical Pacific (Ganeshram et al., 1995; Liu et al., 2008) therefore does not necessarily mean a decrease in suboxia. Rather, our experiments show that lower $\delta^{15}N_{org}$ might also be caused by an increase in local $N_2$ fixation and sedimentary denitrification. The decrease in $\delta^{15}N_{NO_3}$ associated with more sedimentary denitrification and local $N_2$ fixation demonstrates the complexity of interpreting sedimentary $\delta^{15}N_{org}$ records in the lower latitudes.

## 6  Conclusions

The stable isotopes of carbon ($\delta^{13}C$) and nitrogen ($\delta^{15}N$) are proxies that have been fundamental for understanding the ocean. We have included both isotopes into the ocean component of an Earth System Model, CSIRO Mk3L-COAL, to enable future studies with the capability for direct model-proxy data comparisons. We detailed how these isotopes are simulated, how to conduct model-data comparisons using both water column and sedimentary data, and some basic assessment of changes caused by altered ecosystem functioning. We made three overall findings. First, CSIRO Mk3L-COAL performs well alongside a number of isotope-enabled global ocean GCMs. Second, alteration of $\delta^{13}C$ during formation of foraminiferal calcite does not jeopardise simple one to one comparisons with simulated $\delta^{13}C$ of DIC, while diagenetic alteration of bulk organic $\delta^{15}N$ during early burial must be accounted for in model-data comparisons. Third, changes in how marine ecosystems function can have significant and complex effects on $\delta^{13}C$ and $\delta^{15}N$. Our idealised experiments hence showed that the interpretation of palaeoceanographic records may suffer from multiple lines of interpretation, particularly records from the lower latitudes where multiple processes imprint on the isotopic signatures laid down in sediments. Future work will involve palaeoceanographic simulations of CSIRO Mk3L-COAL that seek to understand how the oceanic carbon and nitrogen cycles respond to and influence important climate transitions.

*Data availability.* All model output is provided for download on Australia's National Computing Infrastructure (NCI) at https://geonetwork.nci.org.au/geor and is citable with doi:10.25914/5c6643f64446c. Nitrogen isotope data are available by request to Dario M. Marconi and Daniel M. Sigman at Princeton University. LOVECLIM data is freely available for download at https://researchdata.ands.org.au/loveclim-glacial-maximum-d13c-d14c/792249. UVic-MOBI data was provided by Christopher Somes, PISCES data by Laurent Bopp, iCESM-high data from Simon Yang and iCESM-low data by Alexandra Jahn.

**Figure 15.** Changes in the distribution of marine $N_2$ fixation caused by altering how limiting iron is to the growth of $N_2$ fixers via the coefficient $K_{Fe}^D$ in Eq. (A12). Iron limitation is sequentially relaxed from top to bottom.

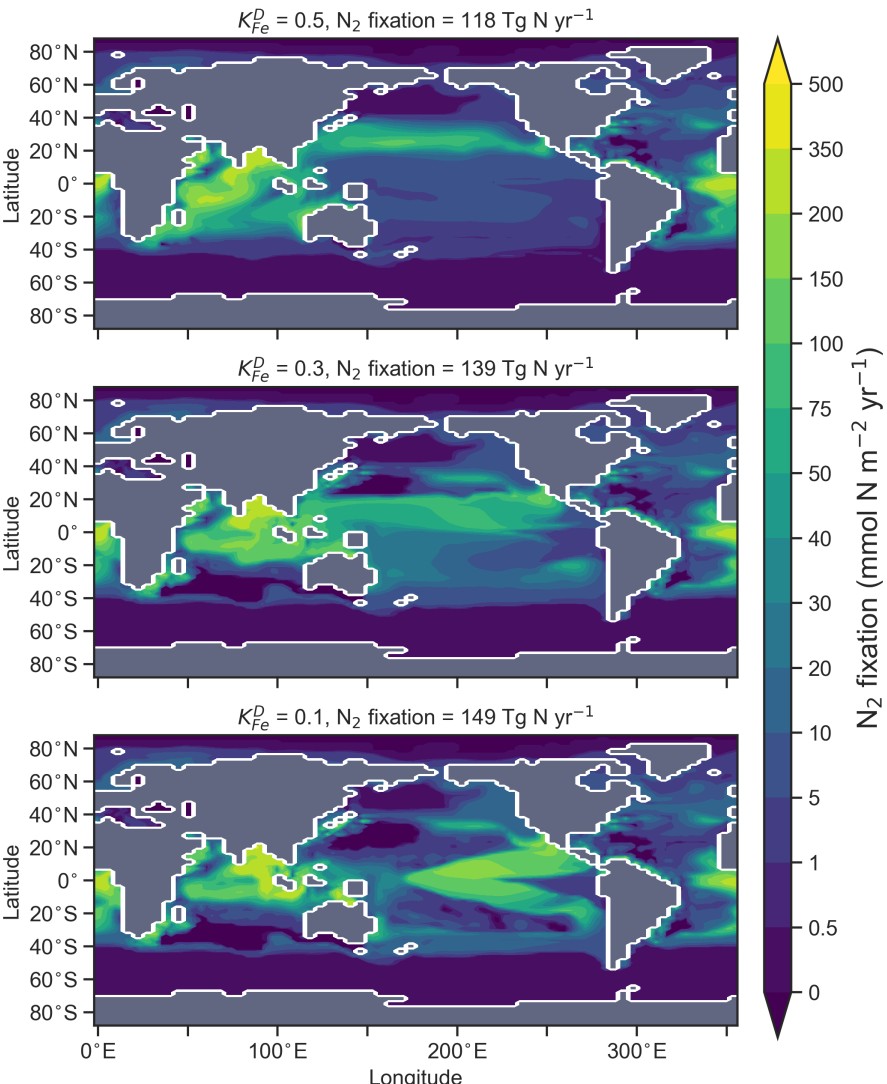

**Figure 16.** Change in $\delta^{15}N_{NO_3}$ caused by a stronger coupling between $N_2$ fixation and tropical regions of low $NO_3{:}PO_4$ concentrations (i.e. tropical upwelling zones with active water column denitrification). The top panel shows the global zonal mean change, while the bottom panel shows the average change in the euphotic zone, here defined as the top 100 metres. Areas with very low $NO_3$ ($< 0.1$ mmol m$^{-3}$) are masked out.

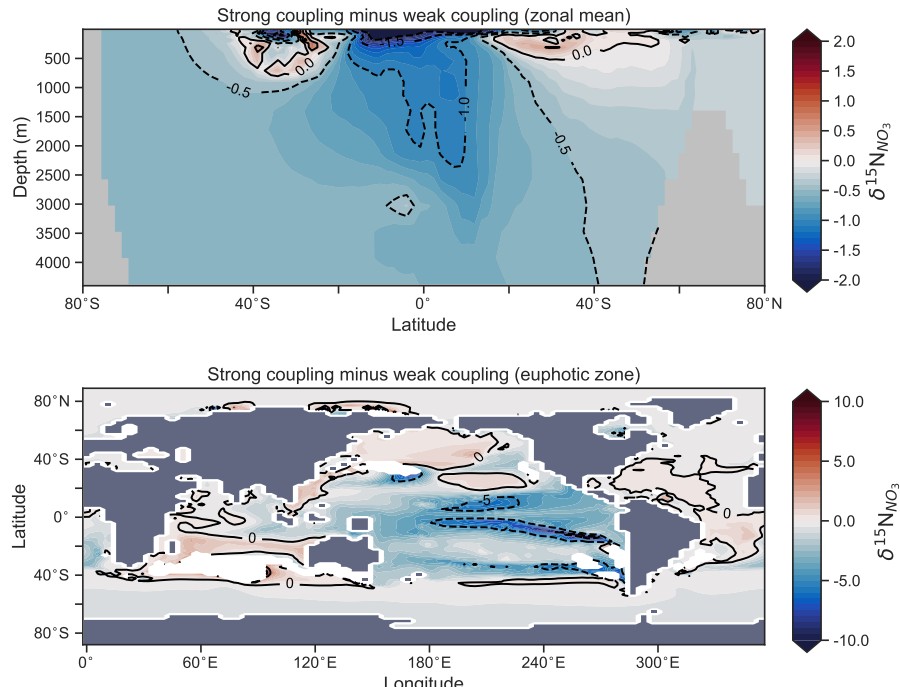

*Code availability.* The source code for CSIRO Mk3L-COAL is shared via a repository located at http://svn.tpac.org.au/repos/CSIRO_Mk3L/branches/CSI COAL/. Access to the repository may be obtained by following the instructions at https://www.tpac.org.au/csiro-mk3l-access-request/. Access to the source code is subject to a bespoke license that does not permit commercial usage, but is otherwise unrestricted. An "out-of-the-box" run directory is also available for download with all files required to run the model in the configuration used in this study,

although users will need to modify the *runscript* according to their computing infrastructure.

## Appendix A:  Ecosystem component of the OBGCM

### A1    Export production

#### A1.1    General phytoplankton group ($^{G}$)

The production of organic matter by the general phytoplankton group ($P_{org}^{G}$) is measured in units of mmol phosphorus (P) m$^{-3}$
5    day$^{-1}$, and is dependent on temperature (T), nutrients (PO$_4$, NO$_3$, and Fe) and irradiance (I):

$$P_{org}^{G} = S_{E:P}^{G} \cdot \mu(T)^{G} \cdot \min \left( P_{lim}^{G}, N_{lim}^{G}, Fe_{lim}^{G}, F(I) \right) \tag{A1}$$

where,

$$S_{E:P}^{G} = 0.005 \, \text{mmol PO}_4 \, \text{m}^3$$

$$\mu(T)^{G} = 0.59 \cdot 1.0635^{T} \tag{A2}$$

$$F(I) = 1 - e^{G(I)} \tag{A3}$$

$$G(I) = \frac{I \cdot \alpha \cdot PAR}{\mu(T)} \tag{A4}$$

In the above, $S_{E:P}$ converts growth rates in units of day$^{-1}$ to mmol PO$_4$ m$^{-3}$ day$^{-1}$. $S_{E:P}$ conceptually represents the
export to production ratio, and for simplicity we assume it does not change. $\mu(T)$ is the temperature-dependent maximum
daily growth rate of phytoplankton (doublings day$^{-1}$), as defined by Eppley (1972). The light limitation term ($F(I)$) is the
productivity versus irradiance equation used to describe phytoplankton growth defined by Clementson et al. (1998), and is
dependent on $I$, the daily averaged shortwave incident radiation (W m$^{-2}$), $\alpha$, the initial slope of the productivity versus
radiance curve (day$^{-1}$/(W m$^{-2}$)), and $PAR$, the fraction of shortwave radiation that is photosynthetically active.

The nutrient limitation terms ($P_{lim}^{G}$, $N_{lim}^{G}$, and $Fe_{lim}^{G}$) may be calculated in two ways.

If the option for **static nutrient limitation** is true, then Michaelis-Menten kinetics (Dugdale, 1967) is used:

$$P_{lim}^{G} = \frac{PO_4}{PO_4 + K_{PO_4}^{G}} \tag{A5}$$

$$N_{lim}^{G} = \frac{NO_3}{NO_3 + K_{NO_3}^{G}} \tag{A6}$$

$$Fe_{lim}^{G} = \frac{Fe}{Fe + K_{Fe}^{G}} \tag{A7}$$

Half-saturation coefficients ($K_{nutrient}^{G}$) show a large range across phytoplankton species (e.g. Timmermans et al., 2004), and
so for simplicity, we set $K_{PO_4}^{G}$ = 0.1 mmol PO$_4$ m$^{-3}$ (Smith, 1982), $K_{NO_3}^{G}$ = 0.75 mmol NO$_3$ m$^{-3}$ (Eppley et al., 1969;
Carpenter and Guillard, 1971) and $K_{Fe}^{G}$ = 0.1 $\mu$mol Fe m$^{-3}$ (Timmermans et al., 2001).

If the option for **variable nutrient limitation** is true (default), then Optimal Uptake kinetics (Smith et al., 2009) is used:

$$P_{lim}^G = PO_4 / \left( \frac{PO_4}{1-f_A} + \frac{V/A}{f_A \cdot \text{N:P}} \right) \tag{A8}$$

$$N_{lim}^G = NO_3 / \left( \frac{NO_3}{1-f_A} + \frac{V/A}{f_A} \right) \tag{A9}$$

$$Fe_{lim}^G = \frac{Fe}{Fe+K_{Fe}} \tag{A10}$$

where,

$$f_A = \max \left[ \left( 1 + \sqrt{\frac{[NO_3]}{V/A}} \right)^{-1}, \left( 1 + \sqrt{\frac{[PO_4] \cdot \text{N:P}}{V/A}} \right)^{-1} \right] \tag{A11}$$

Optimal uptake kinetics varies the two terms in the denominator of the Michaelis-Menten form according to the availability of nutrients. It therefore accounts for different phytoplankton communities with different abilities for nutrient uptake, and does so using the $f_A$ term. The $V/A$ term represents the maximum potential nutrient uptake, $V$, over the cellular affinity for that nutrient, $A$, and is set at 0.1.

## A1.2 Diazotrophs ($^D$; N$_2$ fixers)

Organic matter produced by diazotrophs ($P_{org}^D$) is also measured in units of mmol phosphorus (P) m$^{-3}$ day$^{-1}$, and is calculated in the same form of Eq. (A1), but using the maximum growth rate $\mu(T)^D$ of Kriest and Oschlies (2015), notable changes in the limitation terms, and minimum thresholds that ensure the nitrogen fixation occurs everywhere in the ocean, except under sea ice. $P_{org}^D$ is calculated via:

$$P_{org}^D = S_{E:P}^D \cdot \mu(T)^D \cdot \max \left( 0.01, \min \left( N_{lim}^D, P_{lim}, Fe_{lim}^D \right) \right) \cdot \left( 1 - ico \right) \tag{A12}$$

where,

$$\mu(T)^D = \max \left( 0.01, -0.0042T^2 + 0.2253T - 2.7819 \right) \tag{A13}$$

$$N_{lim}^D = e^{-NO_3} \tag{A14}$$

$$P_{lim}^D = \frac{PO_4}{PO_4 + K_{PO_4}^D} \tag{A15}$$

$$Fe_{lim}^D = \max \left( 0.0, \tanh \left( 2Fe - K_{Fe}^D \right) \right) \tag{A16}$$

The half saturation values for PO$_4$ and Fe limitation are set at 0.1 mmol m$^{-1}$ and 0.5 $\mu$mol m$^{-1}$, respectively, in the default parameterisation. The motivation for making N$_2$ fixers strongly limited by Fe was the high cellular requirements of Fe for diazotrophy (see Sohm et al., 2011, and references therein). A dependency on light is omitted from the limitation term when $P_{org}^D$ is produced. The omission of light is justified by its strong correlation with sea surface temperature (Luo et al., 2014) and its negligible effect on nitrogen fixation in the Atlantic Ocean (McGillicuddy, 2014). Finally, the fractional area coverage of sea ice ($ico$) is included to ensure that cold water N$_2$ fixation (Sipler et al., 2017) does not occur under ice, since a light dependency is omitted.

### A1.3 Calcifiers

The calcifying group produces calcium carbonate ($CaCO_3$) in units of mmol carbon (C) m$^{-3}$ day$^{-1}$. The production of $CaCO_3$ is always a proportion of the organic carbon export of the general phytoplankton group ($C_{org}^G$), according to:

$$CaCO_3 = C_{org}^G \cdot R_{CaCO_3} \tag{A17}$$

The ratio of $CaCO_3$ to $C_{org}^G$ ($R_{CaCO_3}$) can be calculated in two ways.

If the option for **fixed** $R_{CaCO_3}$ is true (default), then $R_{CaCO_3}$ is set to 0.08 as informed by the experiments of Yamanaka and Tajika (1996). The production of $CaCO_3$ is thus 8 % of $C_{org}^G$ everywhere.

If the option for **variable** $R_{CaCO_3}$ is true, then $R_{CaCO_3}$ varies as a function of the saturation state of calcite ($\Omega_{ca}$) according to Ridgwell et al. (2007), where:

$$R_{CaCO_3} = 0.022 \cdot \left(\Omega_{ca} - 1\right)^{\eta} \tag{A18}$$

The exponent ($\eta$) is easily modified consistent with the parameterisations of Zhang and Cao (2016) and controls the rate of
$CaCO_3$ production at a given value of $\Omega_{ca}$.

### A2 Remineralisation

### A2.1 General phytoplankton group ($^G$)

Organic matter produced by the general phytoplankton group (in units of phosphorus: $P_{org}^G$) at the surface is instantaneously remineralised each timestep at depth levels beneath the euphotic zone using a power law scaled to depth (Martin et al., 1987).
This power law defines the concentration of organic matter remaining at a given depth ($P_{org}^{G,z}$) as a function of organic matter at the surface ($P_{org}^{G,0}$) and depth itself ($z$). Its form is as follows:

$$P_{org}^{G,z} = P_{org}^{G,0} \cdot \left(\frac{z}{z_{rem}}\right)^b \tag{A19}$$

Where $z_{rem}$ in the denominator represents the depth at which remineralisation begins and is set to be 100 metres everywhere. The OBGCM therefore does not consider sinking speeds, nor an interaction between organic matter and physical mixing. However, variations in the $b$ exponent affect the steepness of the curve, thereby emulating sinking speeds and affecting the
transfer and release of nutrients from the surface to the deep ocean.

Remineralisation of $P_{org}^G$ through the water column is therefore dependent on the exponent $b$ value in Eq. (A19). The $b$ exponent is calculated in two ways.

If the option for **static remineralisation** is true, then $b$ is set to -0.858 according to Martin et al. (1987).

If the option for **variable remineralisation** is true (default), then $b$ is dependent on the component fraction of picoplankton
($F_{pico}$) in the ecosystem. The $F_{pico}$ shows a strong inverse relationship to the transfer efficiency ($T_{eff}$) of organic matter from beneath the euphotic zone to 1,000 metres depth (Weber et al., 2016). Because $F_{pico}$ is not explicitly simulated in OBGCM, we estimate $F_{pico}$ from the export production field in units of carbon ($C_{org}^G$), calculate $T_{eff}$ using the parameterisation of Weber

et al. (2016), and subsequently calculate the $b$ exponent:

$$F_{pico} = 0.51 - 0.26 \cdot \frac{C_{org}^{G} \text{ (mg C m}^{-2}\text{ hour}^{-1})}{C_{org}^{G,max} \text{ (mg C m}^{-2}\text{ hour}^{-1})} \tag{A20}$$

$$T_{eff} = 0.47 - 0.81 \cdot F_{pico} \tag{A21}$$

$$b = \frac{\log(T_{eff})}{\log(\frac{1000}{100})} = \log(T_{eff}) \tag{A22}$$

## A2.2 Diazotrophs ($^D$)

Remineralisation of diazotrophs ($P_{org}^{D}$) is calculated in the same way as the general phytoplankton group ($P_{org}^{G}$), with the exception that the depth at which remineralisation occurs is raised from 100 to 25 metres in Eq. (A19). This alteration emulates

the release of $NO_3$ from $N_2$ fixers well within the euphotic zone, which in some cases can exceed the physical supply from below (Capone et al., 2005). Release of their N and C-rich organic matter (see Stoichiometry section A3.2) therefore occurs higher in the water column than the general phytoplankton group.

## A2.3 Suboxic environments

The remineralisation of $P_{org}^{G}$ and $P_{org}^{D}$ will typically require $O_2$ to be removed, except for in regions where oxygen concen-

10 trations are less than a particular threshold ($Den_{lim}^{O_2}$), which is set to 7.5 mmol $O_2$ m$^{-3}$ and represents the onset of suboxia. In these regions, the remineralisation of organic matter begins to consume $NO_3$ via the process of denitrification. We calculate the fraction of organic matter that is remineralised by denitrification ($F_{den}$) via:

$$F_{den} = \left(1 - e^{-0.5 \cdot Den_{lim}^{O_2}} + e^{O_2 - 0.5 \cdot Den_{lim}^{O_2}}\right)^{-1} \tag{A23}$$

Such that $F_{den}$ rises and plateaus at 100 % in a sigmoidal function as $O_2$ is depleted from 7.5 to 0 mmol m$^{-3}$.

Following this, the strength of denitrification is reduced if the ambient concentration of $NO_3$ is deemed to be limiting.

Denitrification within the modern oxygen minimum zones only depletes $NO_3$ towards concentrations between 15 and 40 mmol m$^{-3}$ (Codispoti and Richards, 1976; Voss et al., 2001). Without an additional constraint that weakens denitrification as $NO_3$ is drawn down, here defined as $r_{den}$, $NO_3$ concentrations quickly go to zero in simulated suboxic zones (Schmittner et al., 2008). We weaken denitrification by prescribing a lower bound at which $NO_3$ can no longer be consumed via denitrification, $Den_{lim}^{NO_3}$, which is set at 30 mmol $NO_3$ m$^{-3}$.

$$r_{den} = 0.5 + 0.5 \cdot \tanh\left(0.25 \cdot NO_3 - 0.25 \cdot Den_{lim}^{NO_3} - 2.5\right) \tag{A24}$$

$$\text{if} \quad F_{den} > r_{den}, \quad \text{then} \quad F_{den} = r_{den} \tag{A25}$$

$F_{den}$ is therefore reduced if $NO_3$ is deemed to be limiting, and subsequently applied against both $P_{org}^{G}$ and $P_{org}^{D}$ to get the proportion of organic matter to be remineralised by $O_2$ and $NO_3$.

If the availability of $O_2$ and $NO_3$ is insufficient to remineralise all the organic matter at a given depth level, $z$, then the unremineralised organic matter will pass into the next depth level. Unremineralised organic matter will continue to pass into lower depth levels until the final depth level is reached, at which point all organic matter is remineralised by either water

column or sedimentary processes. This version of CSIRO Mk3L-COAL does not consider burial of organic matter.

## A2.4 Calcifiers

The dissolution of $CaCO_3$ is calculated using an $e$-folding depth-dependent decay, where the amount of $CaCO_3$ at a given depth $z$ is defined by:

$$CaCO_3^z = CaCO_3^0 \cdot e^{\frac{-z}{z_{dis}}} \tag{A26}$$

Where $z_{dis}$ represents the depth at which $e^{-1}$ of $CaCO_3$ ($\sim 0.37$) produced at the surface remains undissolved.

Calcifiers are not susceptible to oxygen-limited re-mineralisation nor the concentration of carbonate ion because the dissolution of $CaCO_3$ depends solely on the this depth-dependent decay. All $CaCO_3$ reaching the final depth level is remineralised without considering burial. Future work will include a full representation of carbonate compensation.

## A3 Stoichiometry

The elemental constitution, or stoichiometry, of organic matter affects the biogeochemistry of the water column through uptake
(production) and release (remineralisation). The general phytoplankton group and diazotrophs both affect carbon chemistry, $O_2$, and nutrients ($PO_4$, $NO_3$ and Fe), while the calcifiers only affect carbon chemistry tracers (DIC, DI[13]C and ALK).

Alkalinity ratios for both the general and nitrogen fixing groups are the negative of the N:P ratio, such that for a loss of 1 mmol of $NO_3$, alkalinity will increase at 1 mmol Eq m$^{-3}$ (Wolf-Gladrow et al., 2007).

### A3.1 General phytoplankton group ($^G$)

The stoichiometry of the general phytoplankton group is calculated in two ways.

If the option for **static stoichiometry** is true, then the C:N:Fe:P ratio is set according to the Redfield ratio of 106:16:0.00032:1 (Redfield et al., 1937).

If the option for **variable stoichiometry** is true (default), then the C:N:P ratio of $P_{org}^G$ is made dependent on the ambient nutrient concentration according to Galbraith and Martiny (2015):

$$\text{C:P} = \left( \frac{6.9 \cdot [PO_4] + 6}{1000} \right)^{-1} \tag{A27}$$

$$\text{N:C} = 0.125 + \frac{0.03 \cdot [NO_3]}{0.32 + [NO_3]} \tag{A28}$$

$$\text{N:P} = \text{C:P} \cdot \text{N:C} \tag{A29}$$

Thus, the stoichiometry of $P_{org}^G$ varies across the ocean according to the nutrient concentration, and the uptake and release of carbon, nutrients and oxygen (see section A3.4) is dependent on the concentration of surface $PO_4$ and $NO_3$. The ratio of iron to phosphorus (Fe:P) remains fixed at 0.00032, such that 0.32 $\mu$mol of Fe is consumed per mmol of $PO_4$. We chose to maintain a fixed Fe:P ratio because phytoplankton communities from subtropical to Antarctic waters appear to show similar iron contents (Boyd et al., 2015), despite changes in C:N:P. However, the ratio of C:N:Fe does change as a result of varying C:N:P ratios,
with higher C:Fe in oligotrophic environments and lower C:Fe in eutrophic regions.

## A3.2 Diazotrophs ($^D$)

The stoichiometry of diazotrophs is fixed at a C:N:P:Fe ratio of 331:50:1:0.00064, which represents values reported in the literature (Kustka et al., 2003; Karl and Letelier, 2008; Mills and Arrigo, 2010). Diazotrophs do not consume $NO_3$, rather they consume $N_2$, which is assumed to be of unlimited supply, and release $NO_3$ during remineralisation.

## A3.3 Calcifiers

Calcifying organisms produce $CaCO_3$, which includes DIC, $DI^{13}C$ and ALK, and these tracers are consumed and released at a ratio of 1:0.998:2, respectively, relative to organic carbon. Thus, the ratio of C:$DI^{13}$C:Alk relative to each unit of phosphorus consumed by the general phytoplankton group is equal to the rain ratio of $CaCO_3$ to organic phosphorus multiplied by 106:105.8:212. This group has no effect on nutrient tracers or oxygen values.

## A3.4 Stoichiometry of remineralisation

The requirements for oxygen ($O_2^{rem}$:P) and nitrate ($NO_3^{rem}$:P) during oxic and suboxic remineralisation, respectively, are calculated from the C:N:P ratios of organic matter via the equations of Paulmier et al. (2009). Additional knowledge of the hydrogen and oxygen content of the organic matter is also required to calculate $O_2^{rem}$:P and $NO_3^{rem}$:P. However, the hydrogen and oxygen content of phytoplankton depends strongly on the proportions of lipids, carbohydrates and proteins that constitute the cell. As there is no empirical model for predicting these physiological components based on environmental variables, we continue Redfield's legacy by assuming that all organic matter is a carbohydrate of the form $CH_2O$. Future work, however, should address this obvious bias.

To calculate $O_2^{rem}$:P and $NO_3^{rem}$:P, we therefore need to first calculate the amount of hydrogen and oxygen in organic matter via:

$$H:P = 2C:P + 3N:P + 3 \tag{A30}$$

$$O:P = C:P + 4 \tag{A31}$$

Once a C:N:P:H:O ratio for organic matter is known, we calculate $O_2^{rem}$:P and $NO_3^{rem}$:P in units of mmol m$^{-3}$ P$^{-1}$ using the equations of Paulmier et al. (2009):

$$O_2^{rem}:P = -(C:P + 0.25H:P - 0.5O:P - 0.75N:P + 1.25) - 2N:P \tag{A32}$$

$$NO_3^{rem}:P = -(0.8C:P + 0.25H:P - 0.5O:P - 0.75N:P + 1.25) + 0.6N:P \tag{A33}$$

The calculation of $O_2^{rem}$:P accounts for the oxygen that is also needed to oxidise ammonium to nitrate.

From these calculations we find the following requirements of oxic and suboxic remineralisation, assuming the static stoichiometry option for the general phytoplankton group:

$$O_2^{rem}{:}P_{org}^G = 138$$

$$NO_3^{rem}{:}P_{org}^G = 94.4$$

$$O_2 rem{:}P_{org}^D = 431$$

$$NO_3^{rem}{:}P_{org}^D = 294.8$$

These numbers change dynamically alongside C:N:P ratios when the stoichiometry of organic matter is allowed the vary.

## A4  Sedimentary processes

The remineralisation of organic matter within the sediments is provided as an option in the OBGCM. Sedimentary denitrification, and its slight preference for the light isotopes of fixed nitrogen ($\epsilon_{sed}^{15}{}^N = 3$ ‰), is an important component of the marine nitrogen cycle and its isotopes. It acts as an additional sink of $NO_3$, and reduces the $\delta^{15}N$ value of the global ocean by offsetting the strong fractionation of water column denitrification ($\epsilon_{wc}^{15}{}^N = 20$ ‰).

If sedimentary processes are active, the empirical model of Bohlen et al. (2012) is used to estimate the rate of sedimentary denitrification, where the removal of $NO_3$ is dependent on the rate of particulate organic carbon ($C_{org}^G + C_{org}^D$) arriving at the sediments and the ambient concentrations of oxygen and nitrate. In the following, we assume that the concentrations of $NO_3$ and $O_2$ that are available in the sediments are $\frac{2}{3}$ of the concentration in overlying water column based on observations of transport across the diffusive boundary layer (Gundersen and Jorgensen, 1990).

$$\Delta NO_3(sed) = \left(\alpha + \beta \cdot 0.98^{\left(O_2 - NO_3\right)}\right) \cdot \left(C_{org}^G + C_{org}^D\right) \tag{A34}$$

$$\text{where,} \quad \alpha = 0.04 \quad \text{and} \quad \beta = 0.1 \tag{A35}$$

In the above, both the $\alpha$ and $\beta$ values were halved from the values of Bohlen et al. (2012) to raise global mean $NO_3$ concentrations and lower the sedimentary to water column denitrification ratio to between 1 and 2. If $NO_3$ is not available, the remaining organic matter is remineralised using oxygen if the environment is sufficiently oxygenated. An additional limitation is set for sediments underlying hypoxic waters ($O_2 < 40$ mmol m$^{-3}$), where oxic remineralisation is weakened towards zero according to a hyperbolic tangent function ($0.5 + 0.5 \cdot \tanh(0.2 \cdot O_2 - 5)$). If oxygen is also limiting, the remaining organic matter is remineralised via sulfate reduction. As sulfate is not explicitly simulated, we assumed that sulfate is always available to account for the remaining organic matter.

Thus, sedimentary denitrification is heavily dependent on the rate of organic matter arriving at the sediments. However, a large amount of sedimentary remineralisation is not captured using only these parameterisations because the coarse resolution of the OGCM enables it to resolve only the largest continental shelves, such as the shallow Indonesian seas. Many small areas of raised bathymetry in pelagic environments are also unresolved by the OGCM. To address this insufficiency and increase the global rate of sedimentation and sedimentary denitrification, we coupled a sub-grid scale bathymetry to the course resolution OGCM following the methodology of Somes et al. (2013) using the ETOPO5 $\frac{1}{12}^{th}$ of a degree dataset. For each latitude by

longitude grid point, we calculated the fraction of area that would be represented by shallower levels in the OGCM if this finer resolution bathymetry were used. At each depth level above the deepest level, the fractional area represented by sediments on the sub-grid scale bathymetry can be used to remineralise all forms of exported matter ($C_{org}^{G}$, $C_{org}^{D}$ and $CaCO_3$) via sedimentary processes.

5     Also following the methodology of Somes et al. (2013), we included an option to amplify sedimentary denitrification in the upper 250 metres to account for narrow continental shelves that are not resolved by the OGCM. Narrow shelves experience strong rates of upwelling and productivity, and hence high rates of sedimentary denitrification (Gruber and Sarmiento, 1997). To amplify shallow rates of sedimentary denitrification, we included an optional acceleration factor ($\Gamma_{sed}$), set to 3.0 in the default parameterisation, dependent on the total fraction of shallower depths not covered by the sub-grid scale bathymetry:

$$\Delta NO_3(sed) = \Delta NO_3(sed) \cdot \left( \left(1 - F_{sgb}\right) \cdot \Gamma_{sed} + 1 \right) \tag{A36}$$

10    For those grids with a low fraction covered by the sub-grid scale bathymetry ($F_{sgb}$), the amplification of sedimentary denitrification is therefore greatest.

**Appendix B:  Parameterisation of the OBGCM ecosystem component**

Default parameters for the marine ecosystem component of CSIRO Mk3L-COAL are outlined in Tables A1, A2, and A3. The values presented in these tables are required as input when running the ocean model.

**Table A1.** Parameter values controlling export production in the ecosystem component of the CSIRO Mk3L–COAL ocean model. Default settings: *Michaelis-Menton == False, Optimal Uptake == True, fix == True and Vary CaCO3 == False.*

| Parameter | Action | Value | Active when |
|---|---|---|---|
| **General phytoplankton** ($^G$) | | | |
| $S_{E:P}^G$ | export to production ratio | 0.005 mmol P m$^{-3}$ day$^{-1}$ | Always |
| $\alpha$ | initial slope of production versus irradiance curve | 0.025 day$^{-1}$/(W m$^{-2}$) | Always |
| $PAR$ | fraction of shortwave radiation that is photosynthetically active | 0.5 | Always |
| $K_{PO_4}^G$ | half saturation coefficient for phosphate | 0.1 mmol P m$^{-3}$ | Michaelis-Menton == True |
| $K_{NO_3}^G$ | half saturation coefficient for nitrate | 0.75 mmol N m$^{-3}$ | Michaelis-Menton == True |
| $K_{Fe}^G$ | half saturation coefficient for iron | 0.1 $\mu$mol Fe m$^{-3}$ | Michaelis-Menton == True |
| $V/A$ | Maximum potential uptake over affinity for nutrient | 0.1 | Optimal Uptake == True |
| **Diazotrophs** ($^D$) | | | |
| $S_{E:P}^D$ | export to production ratio | 0.005 mmol P m$^{-3}$ day$^{-1}$ | fix == True |
| $K_{Fe}^D$ | half saturation coefficient for iron | 0.5 $\mu$mol Fe m$^{-3}$ | fix == True |
| $K_{Fe}^D$ | half saturation coefficient for phosphate | 0.1 mmol PO$_4$ m$^{-3}$ | fix == True |
| **Calcifiers** | | | |
| $R_{CaCO_3}$ | ratio of CaCO$_3$ produced per unit carbon of $P_{org}^G$ | 0.08 | Vary CaCO3 == False |
| $R_{CaCO_3}$ | ratio of CaCO$_3$ produced per unit carbon of $P_{org}^G$ | 0.022 | Vary CaCO3 == True |
| $\eta$ | exponent varying $R_{CaCO_3}$ due to calcite saturation | 0.53, 0.81 or 1.09 | Vary CaCO3 == True |

**Table A2.** Parameter values controlling remineralisation in the ecosystem component of the CSIRO Mk3L-COAL ocean model. Default settings: *ReminPico == True, fix == True, den == True,* and *sedfluxes == True.*

| Parameter | Action | Value | Active when |
|---|---|---|---|
| **General phytoplankton ($^G$)** | | | |
| $b$ | remineralisation profile exponent | -0.858 | ReminPico == False |
| $b$ | remineralisation profile exponent | -0.7 to -1.2 | ReminPico == True |
| $z_{rem}$ | depth at which remineralisation begins | 100 m | Always |
| **Diazotrophs ($^D$)** | | | |
| $z_{rem}$ | depth at which remineralisation begins | 25 m | fix == True |
| **Calcifiers** | | | |
| $z_{dis}$ | depth at which $e^{-1}$ CaCO$_3$ remains undissolved | 3500 m | Always |
| **Suboxic environments (affects $^G$ and $^D$)** | | | |
| $Den_{lim}^{O_2}$ | Dissolved oxygen concentration when denitrification begins | 7.5 mmol m$^{-3}$ | den == True |
| $Den_{lim}^{O_2}$ | Nitrate concentration when denitrification is limited | 30.0 mmol m$^{-3}$ | den == True |
| **Sediments (affects $^G$ and $^D$)** | | | |
| $\alpha$ | First constant in Bohlen Eq. (A34) | 0.04 | sedfluxes == True |
| $\beta$ | Second constant in Bohlen Eq. ( A34) | 0.1 | sedfluxes == True |
| $\Gamma_{sed}$ | Sedimentary remineralisation amplification factor | 3.0 | sedfluxes == True |

*Author contributions.* PJB designed the study, undertook model development, ran the experiments, analysed model output and wrote the manuscript. RJM designed the study, provided instruction on development, aided in analysis and edited the manuscript. ZC designed the study, aided in analysis and edited the manuscript. SJP aided in model development and edited the manuscript. NLB aided in interpretation of results and edited the manuscript.

5 *Competing interests.* The authors declare no competing interests

*Acknowledgements.* The Australian Research Council's Centre of Excellence for Climate System Science and the Tasmanian Partnership for Advanced Computing (TPAC) were instrumental for this research. This research was supported under the Australian Research Council's Special Research Initiative for the Antarctic Gateway Partnership (Project ID SR140300001). The authors wish to acknowledge the use of the Ferret program for the analysis undertaken in this work. Ferret is a product of NOAA's Pacific Marine Environmental Laboratory (Information
10 is available at http://ferret.pmel.noaa.gov/Ferret/). The matplotlib package (Hunter, 2007) and the cmocean package (Thyng et al., 2016) were used for producing the figures. We are indebted to Kristen Karsh, Daniel Sigman, Dario Marconi and Eric Raes for discussions that focussed this work. Special thanks to Christopher Somes for correspondence in some development steps and revisions that improved the manuscript. Finally, the lead author is indebted to an Australian Fulbright postgraduate scholarship, which supported him at the Princeton Geosciences department during the writing of this manuscript.

**Table A3.** Parameter values controlling stoichiometry in the ecosystem component of the CSIRO Mk3L–COAL ocean model. Default settings: *Vary Stoich == True,* *Vary frac13 == False* and *fix == True.*

| Parameter | Action | Value | Active when |
|---|---|---|---|
| **General phytoplankton ($^G$)** | | | |
| C:ALK:N:Fe:P | Nutrient stoichiometry of general phytoplankton | 106:-16:16:0.00032:1 | Vary Stoich == False |
| $O_2^{rem}$:$NO_3^{rem}$:P | Remineralisation requirements of $O_2$ and $NO_3$ | -138:-94.4:1 | Vary Stoich == False |
| C:P | Carbon stoichiometry of general phytoplankton | (50 to 170):1 | Vary Stoich == True |
| N:P | Nitrogen stoichiometry of general phytoplankton | (7 to 26):1 | Vary Stoich == True |
| $O_2^{rem}$:P | Remineralisation requirements of $O_2$ | (-220 to -60):1 | Vary Stoich == True |
| $NO_3^{rem}$:P | Remineralisation requirements of $NO_3$ | (-42 to -150):1 | Vary Stoich == True |
| $^{13}C$ $\epsilon_{bio}$ | Carbon isotope fractionation during biological assimilation | 21 ‰ | Vary frac13 == False |
| $^{13}C$ $\epsilon_{bio}$ | Carbon isotope fractionation during biological assimilation | 15 to 25 ‰ | Vary frac13 == True |
| **Diazotrophs ($^D$)** | | | |
| C:$^{13}$C:ALK:N:Fe:P | Nutrient stoichiometry of general phytoplankton | 331:327:-50:50:0.00064:1 | fix == True |
| $O_2^{rem}$:$NO_3^{rem}$:P | Remineralisation requirements of $O_2$ and $NO_3$ | -431:-294.8:1 | fix == True |
| **Calcifiers** | | | |
| C:$^{13}$C:ALK:P | Nutrient stoichiometry of calcifiers | 106:105.8:212:1 | Always |

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
