# Peer review of "Ocean carbon and nitrogen isotopes in CSIRO Mk3L-COAL version 1.0: A tool for palaeoceanographic research"

_Geoscientific Model Development, 2018_

## Referee Comment (RC1) · Anonymous Referee #1 · 21 Dec 2018

This paper describes the implementation of13C and 15N isotopes into the ocean model of the CSIRO model, shows the performance of the model to simulate pre-industrial conditions compared to data and other models, as well as explores the sensitivity of the simulated isotopes to different representations of the marine biological system. This paper is very well written, shows novel results and documents important model features (isotope capability), references the required literature, and has extremely nice figures. I have never had the pleasure to review such a nice a first submission.

This isn't the first time 13C and 15N have been added to an ocean model, but it is still worthwhile to publish this, to document the implementation and performance of

these isotopes for the CSIRO model. Furthermore, since the new model is compared to previous model results, this paper is useful beyond just a technical reference, but actually shows how different isotope-enabled models compare, which is of interest to many scientists. This comparison, which is great, is also related to the only real question I have: why is the Community Earth System Model (CESM), which also includes 13C (Jahn et al. 2015) and 15N (Yang and Gruber, 2016) isotopes, not included in the comparison with previous models? It would be great to see how the new developments in CSIRO compare to that model as well. But maybe those runs were not available for the same time period? If it's not possible to easily include results from the CESM in the paper, that's okay, but I would recommend to at least cite those two papers in the list of previous models that have these isotopes, in addition to the models currently listed (UVic, LOVECLIM, PICES).

Based on my reading of the manuscript, it clearly fits the criteria for publication in GMD.

References: Yang, S., and N. Gruber (2016), The anthropogenic perturbation of the marine nitrogen cycle by atmospheric deposition: Nitrogen cycle feedbacks and the 15N Haber‐Bosch effect, Global Biogeochem. Cycles, 30, 1418–1440, doi:10.1002/2016GB005421. Jahn, A., Lindsay, K., Giraud, X., Gruber, N., Otto-Bliesner, B. L., Liu, Z., and Brady, E. C.: Carbon isotopes in the ocean model of the Community Earth System Model (CESM1), Geosci. Model Dev., 8, 2419-2434, https://doi.org/10.5194/gmd-8-2419-2015, 2015.

Minor: Page 27, Line 1: The link goes off the page
* * *

---

## Referee Comment (RC2) · Somes (Referee) · 24 Dec 2018

Overview

This manuscript from Buchanan et al. describes and evaluates carbon and nitrogen isotopes in a computationally efficient Earth System Model designed for paleoceanography. The new isotopic components are described including its equations. It is validated against modern dissolved and core-top observations and shown to generally reproduce the observations. A mini model intercomparison is done to show COAL performs similarly to other Earth system models. A few additional experiments show the sensitivity to non-Redfield stoichiometry, iron limitation of diazotrophy, and calcite

saturation state.

Overall I think this is a satisfactory evaluation of the new isotopic components of the model. The description of the model is well done. I thought it was well written and there was a good balance of technical information including equations in the main text versus the Appendix. However, I do have comments that should be addressed before I would recommend publication.

Cheers,

Christopher Somes

GEOMAR Helmholtz Centre for Ocean Research Kiel

Comments

page 7, lines 9-13: Biological carbon fractionation

There should be more discussion justifying why you only account for a species effect and not aqueous $CO_2$ concentration (Popp et al., 1989;Rau et al., 1989) and/or phytoplankton growth rate (Laws et al., 1995). There are of course large uncertainties, but there seems to be some general relationship with aqueous $CO_2$ so I am surprised that is not included in a model designed for paleoceanography.

page 8: N2 fixation fractionation

Since N2 fixers have a lower del15N value than the atmospheric N2, this implies some fractionation, right? Does the del15N value go into diazotrophs biomass and then remineralize or go directly into NO3?

page 9, lines 5-20: NO3 utilization

Please show the model equation used for the calculation of utilization in the model (i.e. "u" in equation 15) since it is not straightforward exactly how this is calculated.

page10, Table 1: UVic model

Although the model is based on UVic, the University of Victoria group has not been involved in the C13 and N15 development. Please replace "UVic" with "UVic-MOBI" (Model of Ocean Biogeochemistry and Isotopes) and "University of Victoria" with "Oregon State University/GEOMAR Kiel".

page 10, line 22: "weak undercurrents that are important for reducing nutrient trapping at the equator"

Strong undercurrents and so-called nutrient trapping occur in the upper kilometer (mostly upper 400 meters), whereas your largest bias is between 1500-3000 meters, so something is missing here. I guess the main problem is that you switch off organic matter remineralization when oxygen runs out which allows the organic matter to sink and remineralize much too deep? If so, this should be pointed out here.

page 11, lines 3-7: ". . . far exceed reconstructions of Eide et al., (2017) . . . it is possible the upper ocean values of Eide et al. (2017) underestimate the preindustrial del13C-DIC field"

I think the robustness of the reconstruction deserves a discussion paragraph if you are going to raise this point. Perhaps there is reason to be somewhat skeptical of this reconstruction in the upper ocean. One important aspect I think they have not accounted for is the anthropogenic effect on biological uptake and remineralization.

My C13 model simulations predict this anthropogenic effect lowers d13C by ∼0.5 per mil in the Pacific at 700 meters (compare "Modern" versus "PreInd" differences at 15uM NO3 in Figure 3 of Glock et al. (2018)), which is due to phytoplankton incorporating the lighter anthropogenic CO2 and remineralizing at depth, whereas their reconstruction suggests basically negligible anthropogenic effect at these depths. Note this effect is required for my model's ability to reproduce the range of modern observations there (see Figure S5 in Glock et al., 2018) and becomes even more important as you approach the surface.

[Figure]

Do all of the models significantly overestimate these upper ocean values? It would be really interesting if you could also run a hindcast simulation forced by observed decreasing atmospheric del13CO2 and reproduce the modern observations. If so, I think you would have a legitimate argument that errors/uncertanties in the reconstruction may be significantly contributing to the large model-data misfit. I leave this up to you if it is feasible to accomplish, but I believe it is an important issue to discuss if this dataset is going to be the standard for model comparison.

That said, I still believe your decision not to include an aqueous CO2 dependency in your phytoplankton carbon fractionation is also likely contributing to your overestimated del13DIC, since that reduces phytoplankton fractionation in the warm open ocean gyres.

page 12: Figure 2

Something seems to be wrong with your color bar scale as it does not match the contours, which I assume are correct.

pages 13-15: Denitrification parameterizations

It is important to be more transparent about the artificial parameterizations to account for known model biases on both water column (i.e. NO3 reduction value) and sedimentary denitrification (i.e. amplification) in the main text. I have no problem including them, but I think it is fair to at least briefly note the effect they have on your simulations (e.g. how much the global rates change because of them).

It is not really a fair comparison to include models that include these artificial parameterization (COAL) to model's that don't (your chosen version of UVic-MOBI, PISCES). For example, our following paper with UVic-MOBI (Somes et al., 2017) with improved nitrogen cycle dynamics including sedimentary amplification better reproduces global mean del15NO3 similarly to COAL. It is not important which version of UVic-MOBI you decide to include, but these key denitrification parameterizations in COAL should be

stated in the main text given its importance for del15N.

I would argue that if water column denitrification cannot react naturally to climate-induced changes to oxygen and remineralization, it significantly limits the model's ability as a tool for paleoceanographic research from a nitrogen isotope perspective. This has led our group to implement physical parameterizations to better mimic equatorial undercurrents (Large et al., 2001;Getzlaff and Dietze, 2013), so we do not have to do rely on this artificial water column denitrification reduction parameterization anymore. This topic should be discussed.

page 14, line 9: del15N in PISCES

Please cite the paper that describes del15N in PISCES; I am unaware of any publication on del15N in PISCES.

pages 17-24: Section 5. Ecosystem effects

I liked the sensitivity experiments focusing on a few key parameters/processes. However, I think they would benefit from an extra table (or two) that summarizes their key results. There are so many numbers mentioned directly in the text, I found it difficult to "digest" them all in a comparative context.

page 18: Variable stoichiometry

Please cite the key studies here and refer to the specific Appendix section that describes this so readers can quickly find it.

page 37: Acknowledgements

Will your published code and model output be accessible to the public?

Bibliography

Getzlaff, J., and Dietze, H. (2013). Effects of increased isopycnal diffusivity mimicking the unresolved equatorial intermediate current system in an earth system climate

model. Geophysical Research Letters. doi: 10.1002/grl.50419

Glock, N., Erdem, Z., Wallmann, K., Somes, C.J., Liebetrau, V., Schonfeld, J., Gorb, S., and Eisenhauer, A. (2018). Coupling of oceanic carbon and nitrogen facilitates spatially resolved quantitative reconstruction of nitrate inventories. Nat Commun 9, 1217. doi: 10.1038/s41467-018-03647-5

Large, W.G., Danabasoglu, G., Mcwilliams, J.C., Gent, P.R., and Bryan, F.O. (2001). Equatorial Circulation of a Global Ocean Climate Model with Anisotropic Horizontal Viscosity. Journal of Physical Oceanography 31, 518-536. doi: doi:10.1175/1520-0485(2001)031<0518:ECOAGO>2.0.CO;2

Laws, E.A., Popp, B.N., Bidigare, R.R., Kennicutt, M.C., and Macko, S.A. (1995). Dependence of phytoplankton carbon isotopic composition on growth rate and [CO2)aq: Theoretical considerations and experimental results. Geochimica et Cosmochimica Acta 59, 1131-1138. doi: https://doi.org/10.1016/0016-7037(95)00030-4

Popp, B.N., Takigiku, R., Hayes, J.M., Louda, J.W., and Baker, E.W. (1989). The post-Paleozoic chronology and mechanism of 13 C depletion in primary marine organic matter. American Journal of Science 289, 436-454. doi: 10.2475/ajs.289.4.436

Rau, G.H., Takahashi, T., and Marais, D.J.D. (1989). Latitudinal variations in plankton $\delta$13C: implications for CO2 and productivity in past oceans. Nature 341, 516. doi: 10.1038/341516a0

Somes, C., Schmittner, A., Muglia, J., and Oschlies, A. (2017). A three-dimensional model of the marine nitrogen cycle during the Last Glacial Maximum constrained by sedimentary isotopes. Frontiers in Marine Science 4. doi: 10.3389/fmars.2017.00108

---

## Referee Comment (RC3) · Anonymous Referee #3 · 2 Jan 2019

Buchanan et al., describe the configuration of carbon and nitrogen isotopes in the CSIRO Mk3L-COAL Earth system model for the specific application to paleoceanographic records. The manuscript also presents a series of experiments demonstrating the impact of parameter values and model configuration. The introduction provides a really good background to isotopes and their use in paleoceanography, and the model itself is described in a good level of detail. I think the choice of experiments is particularly useful and really helps demonstrate the significance of various configurations of the model as well as highlighting potential future uses and scientific questions. As such, I think the manuscript will provide a really useful future resource for modellers as well as paleoceanographers. In general, I don't have any major comments but have

provided some minor comments and suggestions for the authors. Otherwise I recommend the manuscript for publication in Geoscientific Model Developments.

Specific comments:

Pg 4, lines 4 - 5: Does running with the offline OGCM restrict experiments to steady-state / timeslice experiments? What is the speed when the OGCM is online (relevant for transient paleo experiments)?

Pg 4, lines 25 - 27: I found the term "phytoplankton functional types" confusing as this usually refers to ecological models that have explicit plankton biomass state variables whereas this model parameterises the biological transformations of biogeochemical tracers (e.g., Hulse et al., 2017).

Figure 1: PGorg and PDorg have not been defined so were unclear until I had read more of the manuscript.

Pg 10, lines 1 - 2: There are other isotope enabled earth system models (e.g., Hulse et al, 2017, Understanding the causes and consequences of past marine carbon cycling variability through models, Earth Science Reviews, 171, pp. 349 - 382) but I guess these are those with comparable resolution or similar?

Pg 10, lines 5 - 6: I do not really understand what this sentence means: "...because many solutions were cumulatively run for many tens of thousands of years over the full course of development".

Pg 10, lines 20 - 23: Is there oxygen-dependent remineralisation in the model affecting this? If so, this could be stated more explicit here, perhaps linking to the relevant part of the appendices.

Pg 11, lines 4 - 6: It's also possible that the model is missing something. An alternative approach here might be to force the model with anthropogenic CO2 and explicitly account for the Suess effect?

Pg 11, lines 6 - 7: Please elaborate on the reason why it may be overestimated in the lower latitudes.

Tables 1 & 2: I find it difficult to really comprehend the comparisons in this table format. You could alternatively plot the data on Taylor diagrams (so keeping the table data on correlation on one axis and mean-normalised RMSE as the straight line distance) alongside Target diagrams to include the mean. See Jolliff et al., (2009) Summary diagrams for coupled hydrodynamic-ecosystem model skill assessment. Journal of Marine Systems. 76 (1 - 2), pp. 64 - 82

Pg 12, lines 3 - 4: "... suggests that the upper ocean values between 200 and 500 metres of (Eide et al., 2017) are too low." or alternatively there are structural errors common to all models?

Section 4.2.: Of the manuscript, I struggled with this section the most. Firstly, I was not familiar with the Schmittner paper itself and I had to go read it to find out what I needed. Secondly, i'm not sure what extra I have learnt here other than the mismatches in Fig 3 are related to mismatches between modelled DIC and observed DIC which is not really surprising. I think the section could be improved if it included a brief description of the Schmittner calibration and a brief discussion about the challenges of relating the measured foram isotopes and the model output if this is an intended use of the model in the future.

Pg 12, eqn 18: How variable are the depths of the Cibicides d13C observations? When binning the data to the model grid, do you weight the averages by depth? I'm curious about what error could be introduced if say you compared the d13C calculated using eqn 18 with a mid-depth of a model grid-box in the equation that is 100 m in depth for example, if the regridded observations fell predominantly in the upper part of the depth range.

Pg 18: It would help to briefly outline the reasons behind the trends in C:P and N:P when using the variable stoichiometry.

Pg 20 , lines 6 - 15: Is there any significance of these changes to potential paleo-applications?

Pg 21, line 4: "loss of alkalinity", I'm guessing this in the surface ocean not the global ocean inventory?

Pg 22, lines 1 - 3: The general statement that CaCO3 production doesn't affect the isotopes much is fine but a caveat should be added: you do not have a representation of CaCO3 sediments in the model and so cannot model any subsequent changes the alkalinity inventory due to burial/dissolution (e.g., Boudreau et al., 2018: The role of calcification in carbonate compensation, Nature Geoscience, 11 (12), pp. 894 - 900). These changes would be relevant over the timescales you are discussing and may drive further changes.

Pg 29, line 24: are the results in the manuscript run with the static or variable remineralisation scheme?

Pg. 43, lines 38 - 39: Should this be the companion paper: Simulations of radiocarbon in a coarse-resolution world ocean model: 1. Steady state prebomb distributions (https://doi.org/10.1029/JC094iC06p08217)?

---

## Author Comment (AC1) · 14 Feb 2019

Reviewer 1 provided some very generous comments of our work and we would like to thank them for those comments. The reviewer also had one primary suggestion that we agree would improve the manuscript.

Their concern about not including CESM simulations of carbon and nitrogen isotopes is valid. We have contacted the lead authors of two publications that contain model output of the nitrogen and carbon isotopes in this model.

We have subsequently heard back from Simon Yang, the author of a study using N

isotopes (Yang & Gruber, 2016, Global Biogeochemical Cycle), and have included this in the paper.

**[GMDD](https://gmd.copernicus.org)**

---

## Author Comment (AC2) · 14 Feb 2019

Response to Reviewer 2 (Christopher Somes)

Christopher Somes had some specific questions and comments before publication of the manuscript could be recommended. These relate to our treatment of biological fractionation in the carbon isotope routine and some issues with our interpretation/discussion of results of both d13C and d15N.

— Page 7, lines 9-13: Biological carbon fractionation. There should be more discussion justifying why you only account for a species effect and not aqueous $CO_2$ concentration

(Popp et al., 1989; Rau et al., 1989) and/or phytoplankton growth rate (Laws et al., 1995). There are of course large uncertainties, but there seems to be some general relationship with aqueous CO2 so I am surprised that this in not included in a model designed for palaeoceanography.

We have implemented this functionality and we are currently running experiments to quantify the effect of a "variable" fractionation factor (Laws 1995 relationship) versus fixed at 21 per mille. We expect these experiments to come to equilibrium state within a month.

— Page 8: N2 fixation fractionation. Since N2 fixers have a lower del15N value than the atmospheric N2, this implies some fractionation right? Does the del15N value go into diazotrophs biomass and then remineralize or go directly into NO3?

Yes N2 fixers do actually fractionate when fixing N2 to NH4 that is then incorporated into biomass and I suppose our wording here is misleading. We have corrected the sentence to illustrate that while N2 fixers do fractionate during their conversion of N2 gas (with a del15N of +0.7 per mil (Klots & Benson, 1963)) to NH4 that is incorporated into biomass (typically with a value of -1 per mille), we implicitly account for these transformations by specifying the end product.

"Because we simulate NO3 and 15NO3 as tracers, our calculations require solving for an implicit pool of 14NO3 during each reaction involving 15NO3. The introduction of NO3 at a fixed del15NNO3 of -1 ‰ due to remineralisation of N2 fixer biomass provides a simple example with which we can begin to describe our equations. Setting the isotopic value of newly fixed NO3 to -1 ‰ is simple because it removes any complications associated with fractionation. We note, however, that in reality the nitrogenase enzyme does fractionate during its conversion of aqueous N2 (+0.7 ‰ to ammonium, and that the biomass that is subsequently produced can vary substantially depending of the type of nitrogenase enzyme used (vanadium versus molybdenum based) (McRose et al., 2019). However, we choose to implicitly account for these transformations and con-
siderably simplify them by setting the del15N of N2 fixer biomass equal to -1 ‰ which reflects the more common Mo-nitrogenase during N2 fixation (Sigman and Casciotti, 2001). A del15NNO3 of -1 ‰ is equivalent to a 15N:14N ratio of 0.999 in our approach where 0 ‰ equals a 1:1 ratio of 15N:14N. If the amount of NO3 being added is known alongside its 15N:14N ratio, in this case 0.999 for N2 fixation, we are able to calculate how much 15NO3 is added. The derivation is as follows. We begin with two equations that describe the system."

— Page 9, lines 5-20: NO3 utilisation. Please show the model equation used for the calculation of utilisation in the model (i.e. "u" in equation 15) since it is not straightforward exactly how this is calculated.

We have added an additional equation and information in the paragraph to describe what this utilisation factor is and how we calculate it.

— Page 10, Table 1: UVic model. Although the model is based on UVic, the University of Victoria group has not been involved in the C13 and N15 development. Please replace "UVic" with "UVic-MOBI" (Model of Ocean Biogeochemistry and Isotopes) and "University of Victoria" with "Oregon State University/ GEOMAR Kiel".

Corrected.

— Page 10, line 22: "Weak undercurrents that are important for reducing nutrient trapping at the Equator". Strong undercurrents and so-called nutrient trapping occur in the upper kilometre (mostly upper 400 meters), whereas your largest bias is between 1500-3000 meters, so something is missing here. I guess the main problem is that you switch off organic matter remineralisation when oxygen runs out which allows the organic matter to sink and remineralise much too deep? If so, this should be pointed out here.

We have added a sentence that makes the reader aware of our treatment of organic matter remineralisation.

"Alternatively, the expansion oxygen minimum zones could be due to our conservative treatment of organic matter remineralisation (appendix A), where remineralisation will not occur when O2 and NO3 are limiting. Excess, unremineralised organic matter therefore falls deeper in the model in the oxygen-deficient zones."

Also, we are currently running a new experiment where this conservative remineralisation scheme is turned off to assess the effect.

— Page 11, lines 3-7: ". . . far exceed reconstructions of Eide et al., (2017) .. it is possible that the upper ocean values of Eide et al., (2017) underestimate the preindustrial del13C-DIC field". I think the robustness of the reconstruction deserves a discussion paragraph if you are going to raise this point. Perhaps there is reason to be somewhat sceptical of this reconstruction in the upper ocean. One important aspect I think they have not accounted for is the anthropogenic effect on biological uptake and remineralisation. My C13 model simulations predict this anthropogenic effect lowers d13C by 0.5 per mil in the Pacific at 700 meters (compare "Modern" versus "Preind" differences at 15 uM NO3 in Figure 3 of Glock et al., (2018)), which is due to phytoplankton incorporating the lighter anthropogenic CO2 and remineralising at depth, whereas their reconstruction suggests basically negligible anthropogenic effect at these depths. Note this effect is required for my model's ability to reproduce the range of modern observations there (see Figure S5 in Glock et al., 2018) and becomes even more important as approach the surface. Do all of the models significantly overestimate these upper ocean values? It would be really interesting if you could also run a hindcast simulation forced by observed decreasing atmospheric del13CO2 and reproduce the modern observations. If so, I think you would have a legitimate argument that errors/uncertanties in the reconstruction may be significantly contributing to the large model-data misfit. I leave this up to you if it is feasible to accomplish, but I believe it is an important issue to discuss if this dataset is going to be the standard for model comparison. That said, I still believe your decision not to include an aqueous CO2 dependency in your phytoplankton carbon fractionation is also likely contributing to your overestimated del13DIC,

since that reduces phytoplankton fractionation in the warm open ocean gyres.

First, we agree that the underestimation of d13C in the upper ocean in the Eide 2017 dataset is likely due to a neglect of biology introducing low d13C DIC via remineralisation.

Second, thank you for the reference to the Glock et al., 2018 paper. It certainly does seem that the 0.5 per mille offset near the surface (15 uM NO3) between your PI and Modern simulations fits with the offset between the models in this study and Eide reconstruction.

Third, while it is not feasible to run hindcast/historical simulations for this study, we think that the bulk of evidence from the four models shows that the upper ocean Eide reconstruction is likely biased low, owing to the neglect of the biological introduction of low d13C. Replicate figures of Figure 3 (previously figure 2) for each model are now included in a supplement.

The following alterations to this paragraph have been made: "All models performed most poorly in the Atlantic Ocean, with poor correlations, high variability and greater biases, and all models predicted upper ocean del13CDIC >= 2.0 ‰ (Supplementary Figures S1, S2 and S3) which further suggests that the upper ocean values between 200 and 500 metres of (Eide et al., 2017) may be too low. The underestimation of del13CDIC may be due to a neglect of biology introducing anthropogenic, isotopically depleted carbon to surface and subsurface layers in the Eide et al. (2017) reconstruction."

— Page 12: Figure 2 Something seems to be wrong with your color bar scale as it does not match the contours, which I assume are correct.

True! We have corrected the figure. We have also added the same figures but for the different models to the supp material.

— Pages 13-15: Denitrification parameterisations. It is important to be more transparent about the artificial parameterisations to account for known model biases on both water column (i.e. NO3 reduction value) and sedimentary denitrification (i.e. amplification) in the main text. I have no problem including them, but I think it is fair to at least briefly note the effect they have on your simulations (e.g. how much the global rates changes because of them). It is not really a fair comparison to include models that include these artificial parameterisations (COAL) to models that don't (your chosen version of UVic-MOBI, PISCES). For example, our following paper with UVic-MOBI (Somes et al., 2017) with improved nitrogen cycle dynamics including sedimentary amplification better reproduces global mean del15NO3 similarly to COAL. It is not important which version of UVic-MOBI you decide to include, but these key denitrification parameterisations in COAL should be stated in the main text given its importance for del15N. I would argue that if water column denitrification cannot react naturally to climate-induced changes to oxygen and remineralisation, it significantly limits the model's ability as a tool for palaeoceanographic research from a nitrogen isotope perspective. This has led our group to implement physical parameterisations to better mimic equatorial undercurrent (Large et al., 2001; Getzlaff & Dietze, 2013), so we do not have to rely on this artificial water column denitrification reduction parameterisation anymore. This topic should be discussed.

The points raised are important and we have included a discussion of them in the text. We have aimed to be more up front about what the limitations of the model are.

We have added the following: "An important caveat to the del15NNO3 routines of CSIRO Mk3L-COAL should be noted. CSIRO Mk3L-COAL underwent significant tuning of water column and sedimentary denitrification parameterisations in order to reproduce known values of del15NNO3 during development. One important parameter is the lower threshold of NO3 concentration at which point water column denitrification is shut off (section A2.3). In CSIRO Mk3L-COAL this is set at 30 mmol m3, which is an arbitrary limit that was implemented to prevent water column denitrification from reducing NO3 to zero in the large suboxic zones. Hence, a caveat of the current model

is an inability for water column and sedimentary denitrification to realistically adjust as suboxia changes. However, the parameterisation does allow for targeted experiments where the ratio of water column to sedimentary denitrification can be controlled if, for instance, it is unclear how water column and sedimentary denitrification respond to certain conditions. This is currently the case during the Last Glacial Maximum, where expansive suboxic zones in the Pacific (Hoogakker et al., 2018) were counterintuitively associated with lower water column denitrification (Ganeshram et al., 1995). We have, in this version, chosen to keep this parameterisation and note that future developments will involve an option to more realistically and dynamically simulate responses to variations in suboxia."

— Page 14, line 9: del15N in PISCES Please cite the paper that describes del15N in PISCES: I am unaware of any publication on del15N in PISCES.

There is currently no paper describing del15N in PISCES. The data was given to me by Laruent Bopp, who is currently working on a GMD paper for this purpose. I will include a citation of Bopp et al., (in prep) if this is agreeable to the editor/journal.

— Pages 17-24: Section 5. Ecosystem effects. I liked the sensitivity experiments focusing on a few key parameters/processes. However, I think they would benefit from an extra table (or two) that summarizes their key results. There are so many numbers mentioned directly in the text, I found it difficult to "digest" them all in a comparative context.

We have included a summary table of the major biogeochemical effects (table 5).

— Page 18: Variable stoichiometry. Please cite the key studies here and refer to the specific Appendix section that describes this so readers can quickly find it.

Completed. We have also added similar pointers in the other ecosystem experiment sections.

— Page 37: Acknowledgements. Will your published code and model output be accessible to the public.

Yes. The code is already accessible via the link in the Code Availability section. The data is being placed in an online repository for public access on the National Computational Infrastructure in Australia, which will be minted with its own doi.
* * *

---

## Author Comment (AC3) · 15 Feb 2019

Response to Reviewer 3

Reviewer 3 provided helpful suggestions and some very encouraging comments regarding the writing and choice of experiments. Although they had no major concerns with the work, they had minor suggestions that have been helpful to improve the manuscript.

— Page 4, lines 4-5: Does running with the offline OGCM restrict experiments to steady-state / timeslice experiments? What is the speed when the OGCM is online

(relevant for paleo experiments)?

Rewritten. We added the following in parentheses: "(compared to ∼10 years per day in fully coupled mode)."

— Page 4, lines 25-27: I found the term "phytoplankton functional types" confusing as this usually refers to ecological models that explicit plankton biomass state variables whereas this model parameterises the biological transformations of biogeochemical tracers (e.g. Hulse et al., 2017).

Rewritten. The sentence containing "phytoplankton functional types" has been replaced with a new sentence.

This is: "Briefly, the ecosystem model simulates the production, remineralisation and stoichiometry (elemental composition) of a general phytoplankton group, diazotrophs (N2 fixers) and calcifiers."

— Figure 1: PGorg and PDorg have not been defined so were unclear until I had read more of the manuscript.

Rewritten. Number 3 is now: "Biological uptake of nutrients and production of organic and inorganic matter. Particulate organic carbon (POC) is produced by the general phytoplankton group and N2 fixers (diazotrophs), while particulate inorganic carbon (PIC) as calcium carbonate (CaCO3) is produced by calcifiers. Export of POC by the general (G) phytoplankton group and N2 fixers (D) are herein referred to as CGorg and CDorg (see appendix A1), respectively."

— Page 10, lines 1-2: There are other isotope enabled earth system models (e.g. Hulse et al, 2017, Understanding the causes and consequences of past marine carbon cycling variability through models, Earth Science Reviews, 171, pp. 349-382) but I guess these are those with comparable resolution or similar?

We acknowledge that there are other isotope enabled models out there that include box models and Earth System Models of Intermediate Complexity, but we choose to

restrict our comparison to other ocean general circulation models in this instance.

We will clarify that we have chosen GCMs specifically and that our comparison is not exhaustive. "We make these model-data comparisons alongside other isotope-enabled ocean general circulation models, for which we chose a non-exhaustive selection (Table 1)."

— Page 10, lines 5-6: I do not really understand what this sentence: "...because many solutions were cumulatively run for many tens of thousands of years over the full course of development".

Rewritten. To reduce confusion as to what this means, we have altered the text to convey the important information.

"Each experiment was run towards steady-state under pre-industrial atmospheric conditions. All results presented in this paper therefore reflect tracers that have achieved an equilibrium solution. We present annual averages of the equilibrium state in the following analysis."

— Page 10, lines 20-23: Is there oxygen-dependent remineralisation in the model affecting this? If so, this could be stated more explicit here, perhaps linking to the relevant part of the appendices.

Oxygen-dependent remineralisation is included in the model. We chose to conserve oxygen, nitrate and organic matter in the treatment of remineralisation in all situations, with lots of O2 or no O2. So, when there is no O2, denitrification occurs, but some organic matter will go unremineralised and will fall into the grid cell below.

Conservative treatment of organic matter remineralisation in low O2 zones therefore causes a vertical expansion of the oxygen minimum zones. However, there are many reasons to suspect that the coarse resolution ocean model is not adequately resolving the complex tropical ocean currents, and this is the true cause of the unrealistic expansion of the OMZs. In contrast, there is no reason to suspect that the rates of export

production in the tropics are too large and driving too great oxygen demand. Moreover, the choice to conserve organic matter remineralisation is mechanistically important for paleoclimate simulations where different conditions evolve.

However, we acknowledge that this choice to conserve oxygen is causing a vertical expansion of the OMZs. "Alternatively, the expansion oxygen minimum zones could be due to our conservative treatment of organic matter remineralisation (appendix A), where remineralisation will not occur when O2 and NO3 are limiting. Excess, unremineralised organic matter therefore falls deeper in the model in the oxygen-deficient zones."

— Page 11, lines 4-6: It's also possible that the model is missing something. An alternative approach here might be to force the model with anthropogenic CO2 and explicitly account for the Suess effect?

We argue that this is outside the scope of this paper. However, future work will involve historical and future scenarios that will explicitly account for the Suess effect, and also paleoclimate experiments where atmospheric $\delta$13CO2 is different.

— Page 11, lines 6-7: Please elaborate on the reason why it may be an overestimate in the lower latitudes. Rewritten.

"It is also equally possible that our fixed biological fractionation of 21 ‰ may be an overestimate in highly productive tropical regions where high growth rates lower the fractionation factor towards 15 ‰ (Laws et al., 1995)."

However, given the concerns of both this reviewer and reviewer 2 (Christopher Somes), we have begun new experiments that include variable biological fractionation, where values between 15 and 25 ‰ are dynamically simulated according to growth rate and [CO2]aq. These experiments will take a few weeks / one month to complete, and so we ask for another month to integrate their results within the paper if the reviewers / editor think this would be a beneficial or necessary addition.

— Tables 1&2: I find it difficult to really comprehend the comparisons in this table format. You could alternatively plot the data on Taylor Diagrams (so keeping the table data on correlation on one axis and the mean-normalised RMSE as the straight line distance) alongside Target diagrams to include the mean. See Jolliff et al., (2009) Summary diagrams for coupled hydrodynamic-ecosystem model skill assessment. Journal of Marine Systems. 76 (1-2), pp. 64-82.

We have remade both the nitrogen and carbon isotope figures into Taylor Diagrams to better convey the model skill. See figures 2 and 5.

The original tables have been altered to only convey the global and regional means.

We have also included the CESM in the nitrogen isotope comparison. Although we know that carbon isotopes are available for this model, we have yet to hear back from authors of prior studies, and the data is not available online.

— Page 12, lines 3-4: "...suggests that the upper ocean values between 200 and 500 metres of Eide et al. (2017) are too low." Or alternatively there are structural errors common to all models?

We argue that the values of the Eide reconstruction are almost certainly too low. Observations and models both produce values in excess of 2 per mille in the upper ocean. We have added these figures to the supplementary material, and also added a new sentence to the discussion that addresses why this might be the case (see response to reviewer 2). Briefly, the introduction of depleted values via biology is likley biasing the Eide reconstruction too low, even though these authors made attempts to eliminate the suess effect.

— Section 4.2: Of the manuscript, I struggled with this section the most. Firstly, I was not familiar with the Schmittner paper itself and I had to go read it to find out what I needed. Secondly, I'm not sure what extra I have learned here other than the mismatches in Fig. 3 are related to mismatches between modelled DIC and observed DIC,

which is not really surprising. I think the section could be improved if it included a brief description of the Schmittner calibration and a brief discussion about the challenges of relating the measures forma isotopes and the model output if this is an intended use of the model in the future.

Rewritten. Hopefully the adjustments made to the following paragraph are sufficient, but if not, then please advise.

"We extended our assessment of modelled del13CDIC by comparing it to a compilation of benthic del13C values taken from the foraminiferal genus Cibicides (Schmittner et al., 2017), a genus on which much of the palaeoceanographic del13C records are based. For this comparison, we adjusted our predicted del13CDIC using the linear dependence on carbonate ion concentration and depth suggested by Schmittner et al. (2017): del13CCib = 0:45 + del13CDIC − 2.2x10-3 * CO3 − 6.6x10-5 * z This adjustment is necessary because the incorporation of DIC into foraminiferal tests is altered by the concentration of CO3 ions and pressure, such that a one to one comparison between del13CDIC and del13CCib introduces error. By adjusting our three-dimensional del13CDIC output using Eq. (19), we thus attain predicted del13CCib.We also computed measures of statistical fit for a traditional one to one comparison between del13CDIC and del13CCib to assess the benefit of the calibration."

— Page 12, eqn 18: How variable are the depths of the Cibicides d13C observations? When binning the data to the model grid, do you weight the averages by depth. I'm curious about what error could be introduced if say you compared the d13C calculated using eqn 18 with a mid-depth of the model grid-box in the equation that is 100 m in the depth for example, if the regridded observations fell predominantly in the upper part of the depth range.

The correction of modelled DI13C uses a depth dependent term of 6.6x10-5. Thus, at a depth of 1000 meters, the depth term becomes 0.066 per mille. At 3000-4000 metres it only just begins to be significant at 0.2-0.26 per mille. So first we argue that

the depth-dependent term is not the significant effect of the calibration throughout the upper ocean, compared with the CO3 term which is more of the order of 0.2 when CO3 is at 100 mmol/m3 and the constant of 0.45. Second, we argue that taking the bottom, top or mid-depth point of the ocean grid box as the depth used in the correction would have negligible effect on the fidelity of our model-data comparison. We say its negligible because using even the tallest boxes of 450 metres would generate a difference of 0.03 per mille in our model-data comparison.

— Page 18: It would help to briefly outline the reasons behind the trends in C:P and N:P when using the variable stoichiometry.

Rewritten. We have added the following text after the first sentence of this section:

"Organic matter had more carbon and nitrogen per unit phosphorus in regions with low PO4, such as the Atlantic Ocean (Fig 8a), which elevated O2 and NO3 demand during oxic and suboxic remineralisation, respectively. Lower ratios were produced in eutrophic regions such as the subarctic Pacific, Southern Ocean and tropical zones of upwelling. Overall, global mean C:P increased from the Redfieldian 106:1 to 117:1, causing an increase in carbon export from 7.6 to 8.0 Pg C yr-1."

— Page 20, lines 6-15: Is there any significance of these changes to potential pale-oapplications?

Yes, but we suggest that this is covered sufficiently in the current version. We neglect to invoke specific examples of changes in nitrogen and carbon isotopes from past climates because simulations under past climate conditions were not performed. We therefore leave it to the reader to think on our results and possibly identify where interesting effects may lie.

— Page 21, line 4: "loss of alkalinity", I'm guessing this in the surface ocean not the global ocean inventory? Yes. Surface alkalinity. Clarified.

— Page 22, lines 1-3: The general statement that CaCO3 production doesn't affect the

isotopes much is fine but a caveat should be added: you do not have a representation of CaCO3 sediments in the model and so cannot model any subsequent changes in the alkalinity inventory due to burial/dissolution (e.g. Boudreau et al., 2018: The role of calcification in carbonate compensation, Nature Geoscience, 11 (12) pp. 894-900). These changes would be relevant over the timescales you are discussing and may drive further changes.

Agreed. This is a good point. We have added a sentence that this a major caveat and will be addressed in future developments.

"However, we stress that version 1.0 of CSIRO Mk3L-COAL does not include CaCO3 burial or dissolution from the sediments according the calcite saturation state of overlying water (Boudreau, 2013). The neglect of ocean-sediment CaCO3 cycling means the neglect of an important aspect of the global carbon cycle active on millennial timescales Sigman et al. (2010). Changes in CaCO3 burial and dissolution could have a non-negligible effect on $\delta$13C through altering whole ocean alkalinity, which would eventually alter air-sea gas exchanges of CO2 and therein affect surface $\delta$13C. While we do not address these effects here, we aim to do so in upcoming versions of the model."

— Page 29, line 24: are the results of the manuscript run with the static or variable remineralisation scheme?

Rewritten. This has been clarified by adding (default) to the end of these sentences in the Appendix.

— Page 43, lines 38-39: Should this be the companion paper: Simulations of radiocarbonin a coarse-resolution world ocean model: 1. Steady state prebomb distributions (https://doi.org/10.1029/JC094iC06p08217)?

We interpret the reviewers suggestion as writing another paper describing the implementation of radiocarbon in the ocean model. This could be possible, but we have not attempted to do so as yet.

---

## Author Comment (AC4) · 20 Feb 2019

Further responses to two comments made by Reviewer 2 (1st and last):

— Page 7, lines 9-13: Biological carbon fractionation. There should be more discussion justifying why you only account for a species effect and not aqueous $CO_2$ concentration (Popp et al., 1989; Rau et al., 1989) and/or phytoplankton growth rate (Laws et al., 1995). There are of course large uncertainties, but there seems to be some general relationship with aqueous $CO_2$ so I am surprised that this in not included in a model designed for palaeoceanography. — Our first response was: We have implemented this functionality and we are currently running experiments to quantify the effect of a

"variable" fractionation factor (Laws 1995 relationship) versus fixed at 21 per mille. We expect these experiments to come to equilibrium state within a month.

We wish to also state that the implementation of this functionality is new, and that we could either discuss in the text about how this functionality will be explored in future or wait until these experiments are complete and include there effects.

— Page 37: Acknowledgements. Will your published code and model output be accessible to the public. —

Our first response was: Yes. The code is already accessible. The data is being placed in an online repository for public access on the National Computational Infrastructure in Australia, which will be minted with its own doi.

We also wish to state where exactly the code is available: https://www.tpac.org.au/csiro-mk3l-access-request/

---

## Author Response (AR2)

Dear Authors,

Thank you for your responses to your referees and for your revised manuscript.

After reviewing these, I am generally satisfied that your manuscript is now suitable for publication.

However, there are two points in your response to Referee 2 in which you mention performing additional work to address comments: "We expect these experiments to come to equilibrium state within a month and may be able to comment on them then"; and then, later, "Also, we are currently running a new experiment [that] could possibly be integrated within the manuscript in a few sentences".

For obvious reasons, the version of your manuscript that accompanies these responses does not currently address these points. As such, I am returning your manuscript to you for further revision that clarifies and finalises these points. Assuming that the additional work alluded to is complete (or will complete shortly), please revise your manuscript appropriately.

If you have any questions, please do not hesitate to get in contact.

With best regards,

Andrew Yool.

Dear Editor,

We have included new data based on the suggestions of reviewers 2 and 3 regarding the treatment of biological carbon fractionation and incorporated these results within the manuscript. Specifically, within the carbon isotope section. A new figure (4) and table (3) are introduced.

We have not included experiments suggested by reviewer 2 where remineralisation is made somewhat independent of oxygen, such that organic matter is remineralised even when oxygen is not sufficient. This suggestion was meant as a possible solution to reducing vertically expansive oxygen minimum zones, but it will require a greater effort of model development to incorporate the nitrogen cycle.

We have also taken the opportunity to clean up the manuscript by making minor edits in certain places outside of the carbon isotope section. We made a special effort is to make the description of the carbon isotope equations easier to understand and also the discussion of model-data comparison of carbon isotopes. Figures 2 and 4 (now figure 5) have been updated.

Finally, it should be noted that the lead author is now located at the Department of Earth, Ocean and Ecological Sciences, University of Liverpool, Liverpool, UK at pearse.buchanan@liverpool.ac.uk.

Thank you,

Pearse Buchanan, Richard Matear, Zanna Chase, Steven Phipps and Nathan Bindoff.

**Response to Reviewer 1**

Reviewer 1 provided some very generous comments of our work and we would like to thank them for those comments. The reviewer also had one primary suggestion that we agree would improve the manuscript.

Their concern about not including CESM simulations of carbon and nitrogen isotopes is valid. We have contacted the lead authors of two publications that contain model output of the nitrogen and carbon isotopes in this model.

We have subsequently heard back from Simon Yang, the author of a study using N isotopes (Yang & Gruber, 2016, Global Biogeochemical Cycle), and Alexandra Jahn, the author of a study using C isotopes (Jahn et al., 2015, Geoscientific Model Development), and have included these results in the paper.

**Response to Reviewer 2 (Christopher Somes)**

Christopher Somes had some specific questions and comments before publication of the manuscript could be recommended. These relate to (1) our treatment of biological fractionation in the carbon isotope routine, (2) some issues with our interpretation/discussion of results.

*Page 7, lines 9-13: Biological carbon fractionation.*

*There should be more discussion justifying why you only account for a species effect and not aqueous $CO_2$ concentration (Popp et al., 1989; Rau et al., 1989) and/or phytoplankton growth rate (Laws et al., 1995). There are of course large uncertainties, but there seems to be some general relationship with aqueous $CO_2$ so I am surprised that this in not included in a model designed for palaeoceanography.*

We have implemented this functionality and have quantified the effect of a "variable" fractionation factor (Laws 1995 relationship) versus fixed at 21 per mille.

The effect is significant in terms of absolute values of carbon isotopes, but does not have a significant effect on other measures of model skill.

Please see the altered discussion of carbon isotope model-data assessment.

Page 8: $N_2$ fixation fractionation.

Since $N_2$ fixers have a lower del15N value than the atmospheric $N_2$, this implies some fractionation right? Does the del15N value go into diazotrophs biomass and then remineralize or go directly into $NO_3$?

Yes $N_2$ fixers do actually fractionate when fixing $N_2$ to $NH_4$ that is then incorporated into biomass and I suppose our wording here is misleading. We have corrected the sentence to illustrate that while $N_2$ fixers do fractionate during their conversion of $N_2$ gas (with a del15N of +0.7 per mil (Klots & Benson, 1963)) to $NH_4$ that is incorporated into biomass (typically with a value of -1 per mille), we implicitly account for these transformations by specifying the end product.

**Because we simulate $NO_3$ and $_{15}NO_3$ as tracers, our calculations require solving for an implicit pool of $_{14}NO_3$ during each reaction involving $_{15}NO_3$. The introduction of $NO_3$ at a fixed $del_{15}N_{NO_3}$ of -1 ‰ due to remineralisation of $N_2$ fixer biomass provides a simple example with which we can begin to describe our equations. Setting the isotopic value of newly fixed $NO_3$ to -1 ‰ is simple because it removes any complications associated with fractionation. We note, however, that in reality the nitrogenase enzyme does fractionate during its conversion of aqueous $N_2$ (+0.7 ‰) to ammonium, and that the biomass that is subsequently produced can vary substantially depending of the type of nitrogenase enzyme used (vanadium versus molybdenum based) (McRose et al., 2019). However, we choose to implicitly account for these transformations and considerably simplify them by setting the $del_{15}N$ of $N_2$ fixer biomass equal to -1 ‰, which reflects the more common Mo-nitrogenase during $N_2$ fixation (Sigman and Casciotti, 2001).**

**A $del_{15}N_{NO_3}$ of -1 ‰ is equivalent to a $_{15}N:_{14}N$ ratio of 0.999 in our approach where 0 ‰ equals a 1:1 ratio of $_{15}N:_{14}N$. If the amount of $NO_3$ being added is known alongside its $_{15}N:_{14}N$ ratio, in this case**

**0.999 for N₂ fixation, we are able to calculate how much ₁₅NO₃ is added. The derivation is as follows. We begin with two equations that describe the system.**

Page 9, lines 5-20: NO$_3$ utilisation.

Please show the model equation used for the calculation of utilisation in the model (i.e. "u" in equation 15) since it is not straightforward exactly how this is calculated.

We have added an additional equation and information in the paragraph to describe what this utilisation factor is and how we calculate it.

Page 10, Table 1: UVic model.

Although the model is based on UVic, the University of Victoria group has not been involved in the C13 and N15 development. Please replace "UVic" with "UVic-MOBI" (Model of Ocean Biogeochemistry and Isotopes) and "University of Victoria" with "Oregon State University/ GEOMAR Kiel".

Corrected.

Page 10, line 22: "Weak undercurrents that are important for reducing nutrient trapping at the Equator".

Strong undercurrents and so-called nutrient trapping occur in the upper kilometre (mostly upper 400 meters), whereas your largest bias is between 1500-3000 meters, so something is missing here. I guess the main problem is that you switch off organic matter remineralisation when oxygen runs out which allows the organic matter to sink and remineralise much too deep? If so, this should be pointed out here.

We have added a sentence that makes the reader aware of our treatment of organic matter remineralisation.

**Alternatively, the expansion oxygen minimum zones could be due to our conservative treatment of organic matter remineralisation (appendix A), where remineralisation will not occur when O2 and NO3 are limiting. Excess, unremineralised organic matter therefore falls deeper in the model in the oxygen-deficient zones.**

More extensive model development will be required to address this issue, and is beyond the scope of this study. It does leave new pathways for development in subsequent versions. By acknowledging this treatment of remineralisation, we hope that the limitations of the model are more transparent.

Page 11, lines 3-7: "… far exceed reconstructions of Eide et al., (2017) .. it is possible that the upper ocean values of Eide et al., (2017) underestimate the preindustrial del13C-DIC field".

I think the robustness of the reconstruction deserves a discussion paragraph if you are going to raise this point. Perhaps there is reason to be somewhat sceptical of this reconstruction in the upper ocean. One important aspect I think they have not accounted for is the anthropogenic effect on biological uptake and remineralisation.

My C13 model simulations predict this anthropogenic effect lowers d13C by 0.5 per mil in the Pacific at 700 meters (compare "Modern" versus "Preind" differences at 15 uM $NO_3$ in Figure 3 of Glock et al., (2018)), which is due to phytoplankton incorporating the lighter anthropogenic $CO_2$ and remineralising at depth, whereas their reconstruction suggests basically negligible anthropogenic effect at these depths. Note this effect is required for my model's ability to reproduce the range of modern observations there (see Figure S5 in Glock et al., 2018) and becomes even more important as approach the surface.

Do all of the models significantly overestimate these upper ocean values? It would be really interesting if you could also run a hindcast simulation forced by observed decreasing atmospheric del13CO2 and reproduce the modern observations. If so, I think you would have a legitimate argument that errors/uncertanties in the reconstruction may be significantly contributing to the large model-data misfit. I leave this up to you if it is feasible to accomplish, but I believe it is an important issue to discuss if this dataset is going to be the standard for model comparison.
That said, I still believe your decision not to include an aqueous CO2 dependency in your phytoplankton carbon fractionation is also likely contributing to your overestimated del13DIC, since that reduces phytoplankton fractionation in the warm open ocean gyres.

First, we agree that the underestimation of d13C in the upper ocean in the Eide 2017 dataset is likely due to their neglect of biology introducing low d13C DIC via remineralisation.

Second, thank you for the reference to the Glock et al., 2018 paper. It certainly does seem that the 0.5 per mille offset near the surface (15 uM $NO_3$) between your PI and Modern simulations fits with the offset between the models in this study and Eide reconstruction.

Third, while it is not feasible to run hindcast/historical simulations for this study, we think that the bulk of evidence from the four models shows that the upper ocean Eide reconstruction is likely biased low, owing to the neglect of the biological introduction of low d13C. Replicate figures of Figure 3 (previously figure 2) for each model are now included in the supplement.

The following alterations to this paragraph have been made:

**…However, all models performed most poorly in the Atlantic Ocean, with poor correlations, high variability and greater biases.**

**Returning to the consistent positive bias in the upper ocean, most models (except iCESM-low) predicted upper ocean del13CDIC >= 2.0 ‰ (Supplementary Figures S1, S2, S3 and S4) similar to CSIRO Mk3L-COAL. As each model has a unique representation of the ecosystem and consequently a unique treatment of biological fractionation, the common prediction of high upper ocean del13C suggests that the upper ocean values between 200 and 500 metres of Eide et al. (2017) may be too low. The underestimation of del13CDIC may be due to a neglect of biology introducing anthropogenic, isotopically-depleted carbon to surface and subsurface layers via remineralisation (the biological Suess effect). This would in turn suggest that a higher global mean of 0.73 ‰**

**generated from a global compilation of foraminiferal del13C (Schmittner et al., 2017) is perhaps a more accurate representation of preindustrial del13C values.**

Page 12: Figure 2
Something seems to be wrong with your color bar scale as it does not match the contours, which I assume are correct.
True! We have corrected the figure.
We have also added the same figures but for the different models to the supp material.

Pages 13-15: Denitrification parameterisations.
It is important to be more transparent about the artificial parameterisations to account for known model biases on both water column (i.e. NO3 reduction value) and sedimentary denitrification (i.e. amplification) in the main text. I have no problem including them, but I think it is fair to at least briefly note the effect they have on your simulations (e.g. how much the global rates changes because of them).
It is not really a fair comparison to include models that include these artificial parameterisations (COAL) to models that don't (your chosen version of UVic-MOBI, PISCES). For example, our following paper with UVic-MOBI (Somes et al., 2017) with improved nitrogen cycle dynamics including sedimentary amplification better reproduces global mean del15NO3 similarly to COAL. It is not important which version of UVic-MOBI you decide to include, but these key denitrification parameterisations in COAL should be stated in the main text given its importance for del15N.
I would argue that if water column denitrification cannot react naturally to climate-induced changes to oxygen and remineralisation, it significantly limits the model's ability as a tool for palaeoceanographic research from a nitrogen isotope perspective. This has led our group to implement physical parameterisations to better mimic equatorial undercurrent (Large et al., 2001; Getzlaff & Dietze, 2013), so we do not have to rely on this artificial water column denitrification reduction parameterisation anymore. This topic should be discussed.

The points raised are important and we have included a discussion of them in the text. We have aimed to be more up front about what the limitations of the model are.

**An important caveat to the del15NNO3 routines of CSIRO Mk3L-COAL should be noted. CSIRO Mk3L-COAL underwent significant tuning of water column and sedimentary denitrification parameterisations in order to reproduce known values of del15NNO3 during development. One important parameter is the lower threshold of NO3 concentration at which point water column denitrification is shut off (section A2.3). In CSIRO Mk3L-COAL this is set at 30 mmol m3, which is an arbitrary limit that was implemented to prevent water column denitrification from reducing NO3 to zero in the large suboxic zones. Hence, a caveat of the current model is an inability for water column and sedimentary denitrification to realistically adjust as suboxia changes. However, the parameterisation does allow for targeted experiments where the ratio of water column to sedimentary denitrification can be controlled if, for instance, it is unclear how water column and sedimentary denitrification respond to certain conditions. This is currently the case during the Last Glacial Maximum, where expansive suboxic zones in the Pacific (Hoogakker et al., 2018) were counterintuitively associated with lower water column denitrification (Ganeshram et al., 1995). We have, in this version, chosen to keep this parameterisation and note that future developments**

**will involve an option to more realistically and dynamically simulate responses to variations in suboxia.**

Page 14, line 9: del15N in PISCES

Please cite the paper that describes del15N in PISCES: I am unaware of any publication on del15N in PISCES.

There is currently no paper describing del15N in PISCES. The data was given to me by Laruent Bopp, who is currently working on a GMD paper for this purpose. I will include a citation of Bopp et al., (*in prep*) if this is agreeable to the editor/journal.

Pages 17-24: Section 5. Ecosystem effects.

I liked the sensitivity experiments focusing on a few key parameters/processes. However, I think they would benefit from an extra table (or two) that summarizes their key results. There are so many numbers mentioned directly in the text, I found it difficult to "digest" them all in a comparative context.

We have included a summary table of the major biogeochemical effects (table 5).

Page 18: Variable stoichiometry.

Please cite the key studies here and refer to the specific Appendix section that describes this so readers can quickly find it.

Completed. We have also added similar pointers in the other ecosystem experiment sections.

Page 37: Acknowledgements.

Will your published code and model output be accessible to the public.

Yes. The code is already accessible at https://www.tpac.org.au/csiro-mk3l-access-request/. The data is being placed in an online repository for public access on the National Computational Infrastructure in Australia, which will be minted with its own doi.

**Response to Reviewer 3**

Reviewer 3 provided helpful suggestions and some very encouraging comments regarding the writing and choice of experiments. Although they had no major concerns with the work, they had minor suggestions that have been helpful to improve the manuscript.
* * *
*Page 4, lines 4-5: Does running with the offline OGCM restrict experiments to steady-state / timeslice experiments? What is the speed when the OGCM is online (relevant for paleo experiments)?*

Rewritten. We added the following in parentheses: "(compared to ~10 years per day in fully coupled mode)."
* * *
*Page 4, lines 25-27: I found the term "phytoplankton functional types" confusing as this usually refers to ecological models that explicit plankton biomass state variables whereas this model parameterises the biological transformations of biogeochemical tracers (e.g. Hulse et al., 2017).*

Rewritten. The sentence containing "phytoplankton functional types" has been replaced with a new sentence. This is:

"**Briefly, the ecosystem model simulates the production, remineralisation and stoichiometry (elemental composition) of a general phytoplankton group, diazotrophs (N$_2$ fixers) and calcifiers.**"
* * *
Figure 1: PGorg and PDorg have not been defined so were unclear until I had read more of the manuscript.

Rewritten. Number 3 is now: "
**Biological uptake of nutrients and production of organic and inorganic matter. Particulate organic carbon (POC) is produced by the general phytoplankton group and N2 fixers (diazotrophs), while particulate inorganic carbon (PIC) as calcium carbonate (CaCO3) is produced by calcifiers. Export of POC by the general (G) phytoplankton group and N2 fixers (D) are herein referred to as CGorg and CDorg (see appendix A1), respectively.**"
* * *
Page 10, lines 1-2: There are other isotope enabled earth system models (e.g. Hulse et al, 2017, Understanding the causes and consequences of past marine carbon cycling variability through models, Earth Science Reviews, 171, pp. 349-382) but I guess these are those with comparable resolution or similar?

We acknowledge that there are other isotope enabled models out there that include box models and Earth System Models of Intermediate Complexity, but we choose to restrict our comparison to other ocean general circulation models in this instance.

We stand by our current sentence as we make no claim to an exhaustive selection.

**"We make these model-data comparisons alongside other isotope-enabled ocean general circulation models (Table 1)."**
* * *
Page 10, lines 5-6: I do not really understand what this sentence: "…because many solutions were cumulatively run for many tens of thousands of years over the full course of development".

Rewritten. To reduce confusion as to what this means, we have altered the text to convey the important information.

**"Each experiment was run towards steady-state under pre-industrial atmospheric conditions over many thousands of years. All results presented in this paper therefore reflect tracers that have achieved an equilibrium solution. We present annual averages of the equilibrium state in the following analysis."**
* * *
Page 10, lines 20-23: Is there oxygen-dependent remineralisation in the model affecting this? If so, this could be stated more explicit here, perhaps linking to the relevant part of the appendices.

Oxygen-dependent remineralisation is included in the model. We chose to conserve oxygen, nitrate and organic matter in the treatment of remineralisation in all situations, with lots of $O_2$ or no $O_2$. So, when there is no $O_2$, denitrification occurs, but some organic matter will go unremineralised and will fall into the grid cell below.

Conservative treatment of organic matter remineralisation in low $O_2$ zones therefore causes a vertical expansion of the oxygen minimum zones. However, there are many reasons to suspect that the coarse resolution ocean model is not adequately resolving the complex tropical ocean currents, and this is the true cause of the unrealistic expansion of the OMZs. In contrast, there is no reason to suspect that the rates of export production in the tropics are too large and driving too great oxygen demand. Moreover, the choice to conserve organic matter remineralisation is mechanistically important for paleoclimate simulations where different conditions evolve.

However, we acknowledge that this choice to conserve oxygen is causing a vertical expansion of the OMZs.

**"Alternatively, the expansion oxygen minimum zones could be due to our conservative treatment of organic matter remineralisation (appendix A), where remineralisation will not occur when O2 and NO3 are limiting. Excess, unremineralised organic matter therefore falls deeper in the model in the oxygen-deficient zones."**
* * *
Page 11, lines 4-6: It's also possible that the model is missing something. An alternative approach here might be to force the model with anthropogenic $CO_2$ and explicitly account for the Suess effect?

We argue that this is outside the scope of this paper. However, future work will involve historical and future scenarios that will explicitly account for the Suess effect, and also paleoclimate experiments where atmospheric $\delta^{13}CO_2$ is different.
* * *
Page 11, lines 6-7: Please elaborate on the reason why it may be an overestimate in the lower latitudes.

Rewritten.

**"It is also equally possible that our fixed biological fractionation of 21 ‰ may be an overestimate in highly productive tropical regions where high growth rates lower the fractionation factor towards 15 ‰ (Laws et al., 1995)."**

~~However, given the concerns of both this reviewer and reviewer 2 (Christopher Somes), we have begun new experiments that include variable biological fractionation, where values between 15 and 25 ‰ are dynamically simulated according to growth rate and [CO₂]ₐq. These experiments will take one month to complete, and so it would be possible to integrate their results within the paper if the reviewers / editor think this would be a beneficial or necessary addition.~~

These experiments are complete and have motivated alteration of our discussion of the carbon isotope model-data comparison. The new paragraph now states:

**It is also equally possible that a fixed biological fractionation (ε13Cbio) of 21 ‰ may have driven unrealistic enrichment in the simulated field. High growth rates, such as occurs in the tropical regions, are thought to lower the strength of fractionation during carbon fixation (Laws et al., 1995). To explore the possibility of model-data mismatch caused by our choice to fix ε13Cbio at 21 ‰, we implemented biological fractionation that is dependent on phytoplankton growth rate and aqueous CO2 concentration (Eq. 6). We found the implementation of a variable ε13Cbio reduced high values in the upper part of the low latitude ocean, but that this reduction was small (Fig. 4). The overwhelming effect was an increase in del13CDIC throughout the interior, itself caused by weaker fractionation in the tropical ocean. Global mean del13CDIC subsequently increased by 0.25 ‰. Meanwhile, model skill was unaffected (see CSIRO Mk3L-COAL (vary-ε13Cbio) in Fig. 2). Neither fixed nor variable biological fractionation could reproduce the low upper ocean values of the data.**
* * *
Tables 1&2: I find it difficult to really comprehend the comparisons in this table format. You could alternatively plot the data on Taylor Diagrams (so keeping the table data on correlation on one axis and the mean-normalised RMSE as the straight line distance) alongside Target diagrams to include the mean. See Jolliff et al., (2009) Summary diagrams for coupled hydrodynamic-ecosystem model skill assessment. Journal of Marine Systems. 76 (1-2), pp. 64-82.

We have remade both the nitrogen and carbon isotope figures into Taylor Diagrams to better convey the model skill. See figures 2 and 5.

The original tables have been altered to only convey the global and regional means.

We have also included the CESM in both comparisons.
* * *
Page 12, lines 3-4: "…suggests that the upper ocean values between 200 and 500 metres of Eide et al. (2017) are too low." Or alternatively there are structural errors common to all models?

We argue that the values of the Eide reconstruction are almost certainly too low. Observations and models both produce values in excess of 2 per mille in the upper ocean. We have added these figures to the supplementary material.
* * *
Section 4.2: Of the manuscript, I struggled with this section the most. Firstly, I was not familiar with the Schmittner paper itself and I had to go read it to find out what I needed. Secondly, I'm not sure what extra I have learned here other than the mismatches in Fig. 3 are related to mismatches between modelled DIC and observed DIC, which is not really surprising. I think the section could be improved if it included a brief description of the Schmittner calibration and a brief discussion about the challenges of relating the measures forma isotopes and the model output if this is an intended use of the model in the future.

Rewritten.

"We extended our assessment of modelled del13CDIC by comparing it to a compilation of benthic del13C values taken from the foraminiferal genus Cibicides (Schmittner et al., 2017), a genus on which much of the palaeoceanographic del13C records are based. For this comparison, we adjusted our predicted del13CDIC using the linear dependence on carbonate ion concentration and depth suggested by Schmittner et al. (2017):
del13CCib = 0:45 + del13CDIC − 2.2x10$^{-3}$ * CO3 − 6.6x10$^{-5}$ * z
This adjustment is necessary because the incorporation of DIC into foraminiferal tests is altered by the concentration of CO3 ions and pressure, such that a one to one comparison between del13CDIC and del13CCib introduces error. By adjusting our three-dimensional del13CDIC output using Eq. (19), we thus attain predicted del13CCib. We also computed measures of statistical fit for a traditional one to one comparison between del13CDIC and del13CCib to assess the benefit of the calibration."

This has subsequently been edited.

**We extended our assessment of modelled del13CDIC by comparing it to a compilation of benthic del13C measured within the calcite of foraminifera from the genus Cibicides (Schmittner et al., 2017), a genus on which much of the palaeoceanographic del13C records are based. For this comparison, we adjusted our predicted del13CDIC to predicted del13CCib using the linear dependence on carbonate ion concentration and depth suggested by Schmittner et al. (2017):**
**del13CCib = 0:45 + del13CDIC − 2.2x10$^{-3}$ * CO3 − 6.6x10$^{-5}$ * z**

**This adjustment accounts for slight fractionation during incorporation of DIC into foraminiferal calcite and is found to be partly explained by the concentration of CO3 ions and pressure. A one to one comparison between del13CDIC and del13CCib hence introduces some degree of error since this fractionation is not accounted for. Because we are interested in applying simulated del13CDIC to a palaeoceanographic context, we must first be able to convert our simulated del13CDIC to del13CCib in an effort to make better comparisons, particularly as the distribution of CO3 is subject to change. By adjusting our three-dimensional del13CDIC output using Eq. (21), we attain predicted del13CCib (see inset entitled "Calibration" in Fig. 5). For good measure, we also computed measures of statistical fit for a traditional one to one comparison between del13CDIC and del13CCib to assess the benefit of the calibration.**
* * *
Page 12, eqn 18: How variable are the depths of the Cibicides d13C observations? When binning the data to the model grid, do you weight the averages by depth. I'm curious about what error could be introduced if say you compared the d13C calculated using eqn 18 with a mid-depth of the model grid-box in the equation that is 100 m in the depth for example, if the regridded observations fell predominantly in the upper part of the depth range.

The correction of modelled $DI^{13}C$ uses a depth dependent term of $6.6x10^{-5}$. Thus, at a depth of 1000 meters, the depth term becomes 0.066 per mille. At 3000-4000 metres it only just begins to be significant at 0.2-0.26 per mille. So first we argue that the depth-dependent term is not the significant effect of the calibration throughout the upper ocean, compared with the CO3 term which is more of the order of 0.2 when CO3 is at 100 mmol/m3 and the constant of 0.45. Second, we argue that taking the bottom, top or mid-depth point of the ocean grid box as the depth used in the correction would have negligible effect on the fidelity of our model-data comparison. We say its negligible because using even the tallest boxes of 450 metres would generate a difference of 0.03 per mille in our model-data comparison.
* * *
Page 18: It would help to briefly outline the reasons behind the trends in C:P and N:P when using the variable stoichiometry.

Rewritten. We have added the following text after the first sentence of this section:

**"Organic matter had more carbon and nitrogen per unit phosphorus in regions with low $PO_4$, such as the Atlantic Ocean (Fig 8a), which elevated $O_2$ and $NO_3$ demand during oxic and suboxic remineralisation, respectively. Lower ratios were produced in eutrophic regions such as the subarctic Pacific, Southern Ocean and tropical zones of upwelling. Overall, global mean C:P increased from the Redfieldian 106:1 to 117:1, causing an increase in carbon export from 7.6 to 8.0 Pg C $yr^{-1}$."**
* * *
Page 20, lines 6-15: Is there any significance of these changes to potential paleoapplications?

Yes, but we suggest that this is covered sufficiently in the current version. We neglect to invoke specific examples of changes in nitrogen and carbon isotopes from past climates because simulations under past climate conditions were not performed. We therefore leave it to the reader to think on our results and possibly identify where interesting effects may lie.
* * *
Page 21, line 4: "loss of alkalinity", I'm guessing this in the surface ocean not the global ocean inventory?

Yes. Surface alkalinity. Clarified.
* * *
Page 22, lines 1-3: The general statement that CaCO3 production doesn't affect the isotopes much is fine but a caveat should be added: you do not have a representation of CaCO3 sediments in the model and so cannot model any subsequent changes in the alkalinity inventory due to burial/dissolution (e.g. Boudreau et al., 2018: The role of calcification in carbonate compensation, Nature Geoscience, 11 (12) pp. 894-900). These changes would be relevant over the timescales you are discussing and may drive further changes.

Agreed. This is a good point.

We have added a sentence that this a major caveat and will be addressed in future developments.

**"However, we stress that version 1.0 of CSIRO Mk3L-COAL does not include CaCO3 burial or dissolution from the sediments according the calcite saturation state of overlying water (Boudreau, 2013). The neglect of ocean-sediment CaCO3 cycling means the neglect of an important aspect of the global carbon cycle active on millennial timescales Sigman et al. (2010). Changes in CaCO3 burial and dissolution could have a non-negligible effect on $\delta^{13}$C through altering whole ocean alkalinity, which would eventually alter air-sea gas exchanges of CO2 and therein affect surface $\delta^{13}$C. While we do not address these effects here, we aim to do so in upcoming versions of the model."**
* * *
Page 29, line 24: are the results of the manuscript run with the static or variable remineralisation scheme?

Rewritten. This has been clarified by adding **(default)** to the end of these sentences in the Appendix.
* * *
Page 43, lines 38-39: Should this be the companion paper: Simulations of radiocarbonin a coarse-resolution world ocean model: 1. Steady state prebomb distributions (https://doi.org/10.1029/JC094iC06p08217)?

We interpret the reviewers suggestion as writing another paper describing the implementation of radiocarbon in the ocean model. This could be possible, but we have not attempted to do so as yet.

[revised manuscript text omitted]